# Somatostatin triggers local cAMP and Ca²⁺ signaling in primary cilia to modulate pancreatic β-cell function

Ceren Incedal Nilsson [1], Özge Dumral [1], Gonzalo Sanchez [1], Beichen Xie [2,3], Andreas Müller [4,5,6], Michele Solimena [4,5,6], Huixia Ren [2,3] & Olof Idevall-Hagren [1✉]

## Abstract

**Somatostatin, released from δ-cells within pancreatic islets of Langerhans, is one of the most important negative regulators of islet hormone secretion. We find that islet δ-cells are positioned near, and release somatostatin onto, primary cilia of the other islet cell types, including insulin-secreting β-cells. Somatostatin activates ciliary somatostatin receptors, resulting in rapid lowering of the ciliary cAMP concentration which in turn promotes more sustained nuclear translocation of the cilia-dependent transcription factor GLI2 through a mechanism that operates in parallel with the canonical Hedgehog pathway and depends on ciliary Ca²⁺ signaling. We also find that primary cilia length is reduced in islets from human donors with type-2 diabetes, which is associated with a reduction in interactions between δ-cells and cilia. Our findings show that islet cell primary cilia constitute an important target of somatostatin action, which endows somatostatin with the ability to regulate islet cell function beyond acute suppression of hormone release.**

**Keywords** δ-cell; Hedgehog; Protein Kinase A; Type-2 Diabetes
**Subject Categories** Cell Adhesion, Polarity & Cytoskeleton; Signal Transduction

## Introduction

The islets of Langerhans are small cell clusters scattered throughout the pancreas that release blood glucose-regulating hormones. They are primarily composed of insulin-secreting β-cells and glucagon-secreting α-cells, but also of less abundant cell types such as the somatostatin-secreting δ-cells. Somatostatin secretion, like that of the other islet hormones, is glucose-regulated and suppresses the release of both insulin and glucagon (Hauge-Evans et al, 2009).

Somatostatin is therefore considered an important local regulator of islet function and, indirectly, of blood glucose homeostasis. Somatostatin signals via five somatostatin receptors (SSTR1-5) which are G-protein-coupled and typically induce lowering of cAMP that in turn inhibits hormone secretion (Patel et al, 2011). SSTR2 is considered the predominant somatostatin receptor in human islet β-cells (Braun, 2014; Kailey et al, 2012), while SSTR3 appears to be the major isoform in mouse β-cells, at least at the mRNA level (DiGruccio et al, 2016). In addition to its well-characterized acute effects on hormone secretion, somatostatin also contributes to the long-lasting regulation of islet cell function through transcriptional changes (Damsteegt et al, 2019; Zhou et al, 2014), but the mechanisms are poorly understood. SSTR3 is expressed in both mouse and human islet cells and has been shown to contribute to the inhibitory actions of somatostatin (Adamson et al, 2023; Braun, 2014). This receptor contains a signal sequence in one of the intracellular loops that grants it access to the primary cilium, a small rod-like organelle that protrudes from the surface of most mammalian cells, including islet cells (Hughes et al, 2020; Iwanaga et al, 2011). In the ciliary membrane, it joins a subset of other GPCRs, such as neuropeptide Y receptor 2 (NPYR2), melanin-concentrating receptor 1 (MCHR1), GABA_B1 receptor (GABABR), free fatty acid receptor 4 (FFAR4), and the GPCR-like receptor Smoothened (SMO) (Barbeito et al, 2021; Corbit et al, 2005; Idevall-Hagren et al, 2024; Sanchez et al, 2023; Wu et al, 2021). The presence of receptors in the primary cilium enables these structures to function as antennas that sense changes in the local cell environment, and downstream signaling is often restricted to the cilium, which therefore forms an isolated signaling compartment (Delling et al, 2013; Hansen et al, 2022; Sanchez et al, 2023; Truong et al, 2021). cAMP is perhaps the best-characterized ciliary messenger, since it is an essential component of the Hedgehog (Hh) signaling pathway (Hansen et al, 2022; Truong et al, 2021; Wu et al, 2021). The Hh pathway is oriented around the primary cilia, where Hh binds its ciliary receptor Patched and initiate a process involving Patched exit from the cilium and SMO entry into the cilium, leading to reduced intraciliary cAMP through inhibition of adenylate cyclases (ACs).

[1]Department of Medical Cell Biology, Uppsala University, BMC Box 571, 75123 Uppsala, Sweden. [2]Center for Quantitative Biology, Peking University, 100871 Beijing, China. [3]Peking-Tsinghua Center for Life Sciences, Peking University, 100871 Beijing, China. [4]Molecular Diabetology, University Hospital and Faculty of Medicine Carl Gustav Carus, TU Dresden, Dresden, Germany. [5]Paul Langerhans Institute Dresden (PLID) of the Helmholtz Center Munich at the University Hospital Carl Gustav Carus and Faculty of Medicine of the TU Dresden, Dresden, Germany. [6]German Center for Diabetes Research (DZD e.V.), Neuherberg, Germany. ✉E-mail: olof.idevall@mcb.uu.se

The reduction of cAMP levels reduces protein kinase A (PKA) activity that promotes processing and activation of GLI transcription factors, which exit the cilium for long-term regulation of cell function (Bachmann et al, 2016; Moore et al, 2016; Mukhopadhyay et al, 2013). Many ciliary GPCRs, such as SSTR3, are Gαi-coupled, and their activation lowers cAMP levels and may thus tune Hh signaling (Truong et al, 2021). The Hh pathway is important during organ development in the embryo, but it also remains active in many cells during adulthood (Lau and Hebrok, 2010; Petrova and Joyner, 2014). Hh signaling is required for the establishment of functional islets of Langerhans as disruption of the Hh pathway leads to reduced insulin production and secretion (Apelqvist et al, 1997; Cervantes et al, 2010; Landsman et al, 2011; Nakayama et al, 2008). Hh is also involved in maintaining functionality of these organs in adult animals, as inhibition of this pathway lowers both insulin production and secretion (Thomas et al, 2000). These effects are likely also caused by GLI-mediated transcriptional changes in gene expression (Thomas et al, 2001). Importantly, similar changes have been seen following disruption of the primary cilium, indicating that ciliary signaling is important for maintaining functional β-cells (Hughes et al, 2020; Volta et al, 2019). Recent studies have shown that islet cell primary cilia signaling is required for prompting a normal secretory response following a glucose challenge, and that changes in cilia structure and function may participate in the dysregulation of β-cells observed in type-2 diabetes (Chabosseau et al, 2023; Hughes et al, 2020; Kluth et al, 2019; Walker et al, 2023). In this study, we demonstrate that the primary cilium functions as a specialized cAMP compartment that can make physical and chemical contact with somatostatin-secreting δ-cells. Primary cilia being enriched with SSTR3 can detect local somatostatin secretion and tune a non-canonical pathway by lowering the ciliary cAMP levels and induce the nuclear translocation of GLI2 transcription factors.

# Results

## δ-cells make contact with β-cell primary cilia

The δ-cell is a relatively rare cell-type in the islet, but thanks to an irregular morphology with long, filopodia-like protrusions, a single δ-cell can directly interact with many islet cells (Arrojo e Drigo et al, 2019). To determine whether islet δ-cells directly interacted with primary cilia, we performed 3D segmentation of FIB-SEM images of mouse pancreas (Li et al, 2023). As recently reported (Li et al, 2023; Müller et al, 2024), we found that most β-cells have a single primary cilium that protrudes from a deep ciliary pocket into the islet interstitium (Fig. 1A). Islet δ-cells, identified based on their irregular morphology and characteristic secretory granules, were often found in close proximity to β-cell primary cilia, and we observed several examples where a primary cilium that emerged from a β-cell immediately encountered a δ-cell protrusion and then traced this protrusion for an extended distance, sometimes for several μm (Fig. 1B–E). Consistent with previous observations, we also found that the δ-cell protrusion contained somatostatin secretory granules (Arrojo e Drigo et al, 2019; Müller et al, 2024). These results show that there is a structural basis for direct communication between islet δ-cells and the primary cilia of β-cells, and that this communication may involve the release of somatostatin.

## The primary cilia of islet cells are enriched for SSTR3

Somatostatin potently attenuates glucose-stimulated insulin secretion through activation of Gαi-coupled somatostatin receptors and lowering of cAMP, and SSTR3 has been proposed to be the dominating receptor isoform in mouse β-cells. Primary cilia are targets of somatostatin action in neurons, acting via cilia-localized SSTR3 (Tereshko et al, 2021), and islet cells also express SSTR3 in their cilia (Adamson et al, 2023; Iwanaga et al, 2011). Immuno-fluorescence staining of mouse and human islets showed primarily cytosolic localization of SSTR2 (Appendix Fig. S1), while SSTR3 localized to the cilium of clonal MIN6 β-cells, mouse and human islet cells (Fig. 2A,B), consistent with previous studies. Next, we investigated SSTR3 dynamics because it is known that plasma membrane-localized somatostatin receptors rapidly desensitize by internalization (Roth et al, 1997). Immunostaining of MIN6 cells following short (15 min) and long (18 h) exposure to 100 nM somatostatin revealed very stable localization of SSTR3 to the primary cilium. Acute exposure even led to a slight enrichment of the receptor in the cilium, while 18 h stimulation was without effect on both the fraction of cilia positive for SSTR3 and on receptor density in the cilia, although it did cause slight cilia shortening (Fig. 2C–E). Importantly, SSTR3 localization did not change even by prolonged agonist exposure in the cilia of mouse islet cells, and the number of cells showing SSTR3 in their cilia was also unaffected by the presence of somatostatin (Fig. 2F–I). These results indicate that SSTRs function differently in primary cilia compared to the plasma membrane, and that the cilia-localized SSTRs would be well-suited for the detection of tonic somatostatin signals.

## Shortening of islet cell cilia in human type-2 diabetes leads to impaired connectivity with δ-cells

Recent work has indicated correlations between changes in primary cilia function and type-2 diabetes (Kluth et al, 2019; Walker et al, 2023). We obtained isolated human pancreatic islets from five donors with, and seven donors without, type-2 diabetes (Table 1). These islets were immunostained against insulin and acetylated tubulin and imaged by confocal microscopy. We found that primary cilia length was reduced in both insulin-positive and insulin-negative cells from type-2 diabetic islets, while cilia numbers were unchanged (Figs. 3A–C and EV1A,B). Similar to what we previously described in mouse islets (Sanchez et al, 2023), we observed cilia with rod-like shapes and cilia with swollen tips. Both morphologies were present at similar frequencies in type-2 diabetic and non-diabetic islets, and both were shorter in islets from type-2 diabetic donors (Fig. 3D–F). While cilia length negatively correlated with type-2 diabetes diagnosis, there was no correlation with BMI of the donors and only weak negative correlation with donor age (Fig. 3G,H). To mimic a diabetogenic environment in vitro, human islet from non-diabetic donors were cultured for 7 days in the presence of 0.5 mM palmitic acid. This treatment caused a reduction in cilia length similar to what was observed in islets from type-2 diabetic donors (Fig. 3I). Next, we immunolabeled human islets against somatostatin (to label δ-cells) and acetylated tubulin. We found that there was a reduction in δ-cell numbers in islets from donors with type-2 diabetes (Fig. 3K), which was accompanied by an increased distance between primary cilia and δ-cells (Fig. 3L,M). Immunolabeling against SSTR3 did not reveal any obvious

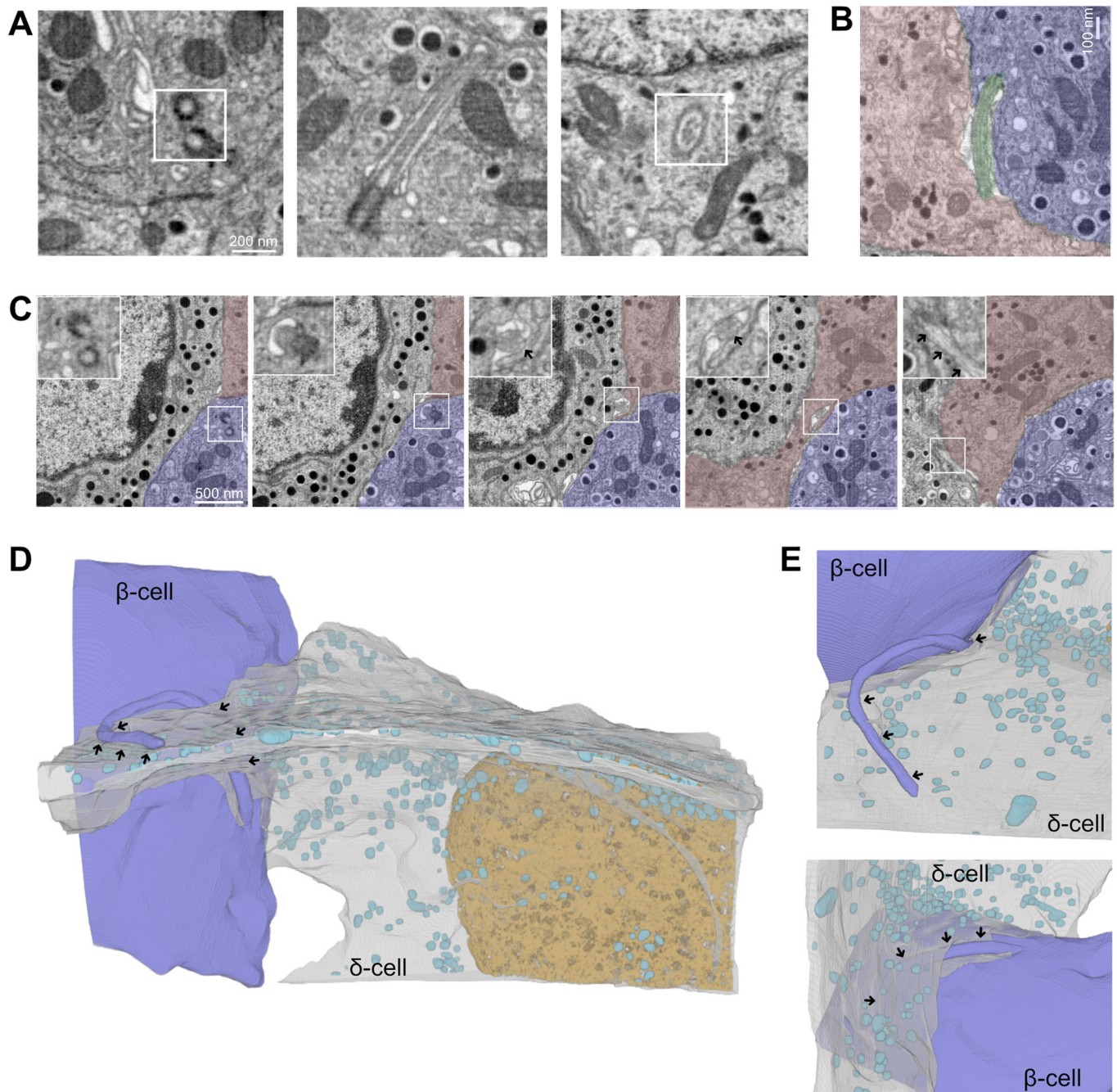

**Figure 1. δ-cell primary cilia contacts within mouse islets revealed by FIB-SEM.**

(A) Electron microscopy images of primary cilia in mouse β-cells. Picture to the left shows a pair of centrioles at the base of the cilium, middle panel shows a longitudinal section of a primary cilium and the right panel shows a transverse section. (B) An example of a β-cell (blue) primary cilium (green) in close proximity to a δ-cell (red). (C) FIB-SEM images from different axial positions of a mouse islet showing a β-cell (blue) primary cilium (boxed area) that encounters a δ-cell (red). Arrows show location of the primary cilium. (D, E) Volumetric reconstruction of FIB-SEM data from a mouse islet showing examples of close interactions between β-cell primary cilia and somatostatin-containing δ-cell protrusions. Arrows indicate contact points between δ-cells (gray) and β-cell primary cilia (blue). Cyan structures are somatostatin-containing granules. Source data are available online for this figure.

difference between islets from non-diabetic and type-2 diabetic organ donors (Fig. 3N). Together, these results show that type-2 diabetes is associated with shortening of primary cilia in both β- and non-β-cells and in reduced interactions between primary cilia and δ-cells.

## Somatostatin secreted from islet δ-cells locally activates ciliary somatostatin receptors

Although our ultrastructural examination clearly showed the existence of contacts between islet cell primary cilia and δ-cells,

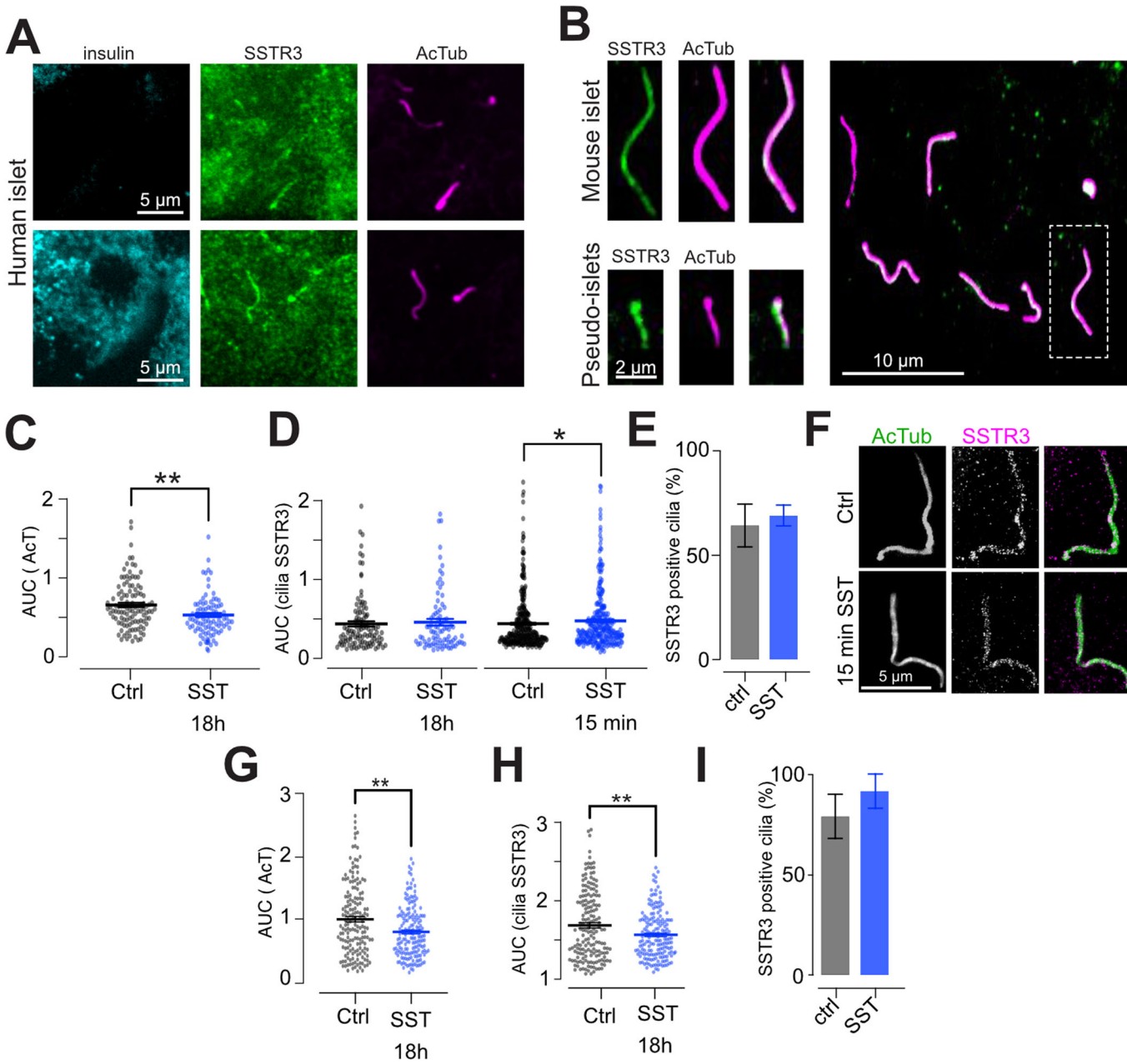

**Figure 2.   Primary cilia of islet cells are enriched for SSTR3.**

(**A**) Confocal microscopy images showing ciliary localization of SSTR3 (green) in both insulin-positive and -negative cells in a human islet. (**B**) Confocal microscopy images from a mouse islet (top) and MIN6 pseudoislet (bottom) showing the localization of SSTR3 to primary cilia (visualized by anti-acetylated tubulin immunostaining and shown in magenta). (**C**) Quantifications of the effect of 18 h somatostatin treatment (100 nM) on cilia length (determined as AUC for normalized acetylated tubulin immunoreactivity) in MIN6 cells (means ± SEM; nctrl = 111, nSST = 87 cilia from three different preparations, change in cilia length $P = 0.003$ assessed by unpaired Kolmogorov–Smirnov test). (**D**) Quantifications of the effect of 18 h (**G**) and 15 min (**H**) somatostatin treatment on cilia SSTR3 immunoreactivity in MIN6 pseudoislets (means ± SEM; for 18 h SST treatments; $n_{ctrl} = 111$, $n_{SST} = 87$ cilia from three different preparations, change in SSTR3 not significant; for 15 min SST treatment $n_{ctrl} = 340$, $n_{SST} = 295$ cilia from three different preparations, $P = 0.0146$. Significance is assessed by unpaired Kolmogorov–Smirnov test). (**E**) Quantification of fraction of SSTR3-positive cilia in MIN6 cells exposed to 100 nM somatostatin for 18 h (means ± SEM; $n_{ctrl} = 111$, $n_{SST} = 87$ cilia from three different preparations). (**F**) STED microscopy images showing the distribution of SSTR3 (magenta) in mouse islet cells under resting condition and after 15 min SST stimulation. (**G–I**) Quantifications of the effect of 18 h somatostatin treatment on cilia length (**G**), cilia SSTR3 immunoreactivity (**H**) and fraction of SSTR3-positive cilia (**I**) in mouse islet cells (means ± SEM; $n_{ctrl} = 170$, $n_{SST} = 171$ cilia from two different animals, $P_{AcT} = 0.0055$, $P_{SSTR3} = 0.0030$, assessed by unpaired Kolmogorov–Smirnov test). Source data are available online for this figure.

**Table 1.** Donor characteristics.

|  | **Non-diabetic** | **Type-2 diabetic** |
|---|---|---|
| Gender | 6 male/0 female | 4 male/1 female |
| Age | 60 ± 6 years (50–69) | 73 ± 6 years (68–80) |
| BMI | 28 ± 4 kg/m$^2$ (22–33) | 29 ± 2 kg/mol (28–32) |
| HbA1c | 37 ± 3 mmol/mol (33–41) | 44 ± 6 mmol/mol (37–48) |
| Cold ischemic time | 11 h 20 min (7 h 59 min–16 h 50 min) | 12 h 40 min (7 h 10 min–20h 57 min) |

Characteristics for the donors of the islets used in the study.

this methodology does not permit easy examinations of large numbers of islet cell cilia-δ-cell interactions. In order to substantiate this claim, we immunostained mouse islets against somatostatin, to identify δ-cells, and against acetylated tubulin to visualize primary cilia, and imaged the samples on a confocal microscope. Consistent with the FIB-SEM data, we found that δ-cells have a non-symmetric morphology with long, filopodia-like protrusions that extend throughout the islet interstitium (Fig. 4A,C). In addition, we observed that δ-cells were found near primary cilia from other islet cell types, and that each δ-cell was in contact with 3.2 cilia on average, compared to 1.8 cilia for non-δ-cells (Fig. 4B). Very similar observations were made when the δ-cells were instead labeled by transgenic expression of tdtomato using the somatostatin promoter, and many of the δ-cell-proximal cilia were positive for somatostatin receptor type 3 (SSTR3) (Fig. 4C–E). Similar proximities were also seen in human islets, but they were less pronounced due to an overall lower cilia density (Fig. 4F; see also Fig. 3). To test if somatostatin is released onto primary cilia, we made use of a recently developed sensor for detection of extracellular somatostatin based on circularly permutated GFP and SSTR5 (SST1.0; (Wang et al, 2023)). Mouse islets were infected with an adenovirus carrying SST1.0 and the islets were examined by confocal microscopy. The sensor localized to the plasma membrane, as expected, but also to small protrusions that resembled primary cilia (Fig. 4G). Immunostaining against the ciliary axoneme confirmed the localization of SST1.0 to the ciliary membrane (Fig. 4H), and sequence analysis as well as immunostainings in mouse and human islets confirmed the ciliary localization of SSTR5 (Fig. EV2A–E). We next expressed SST1.0 in mouse islets with transgenic expression of tdtomato in δ-cells and exposed the islets to 10 mM glucose. This triggered increases in SST1.0 fluorescence in both primary cilia and plasma membrane within distinct islet regions (Fig. 4I). The SST1.0 response amplitude was not different between primary cilia and cilia-adjacent plasma membrane when matched for distance to δ-cells (average distance between primary cilium and δ-cell was 4.3 ± 0.9 μm and between plasma membrane and δ-cell 3 ± 0.7 μm; NS), but the response frequency was 1.8-fold higher in primary cilia (Fig. 4K,L). Importantly, the addition of 1 μM exogenous somatostatin resulted in identical SST1.0 fluorescence changes in both plasma membrane and primary cilia, showing that the sensitivity of SST1.0 is not location-dependent (Fig. 4J). Quantifications further showed that the responses in both plasma membrane and primary cilia inversely correlated with the distance to δ-cells, indicating that the sensor reports local somatostatin release (Fig. 4M). Together, these results show that primary cilia are

exposed to endogenously released somatostatin, and also indicate that somatostatin secretion is at least partially directed towards primary cilia.

## Direct measurement of ciliary cAMP

Somatostatin receptors typically couple to inhibitory G-proteins and cAMP lowering, and this also seems to be the case for ciliary SSTRs (Braun, 2014; Patel et al, 2011). To better understand the role of cAMP in the primary cilium of β-cells, we developed several cilia-targeted cAMP sensors. Despite the existence of previously described cilia-targeted cAMP reporters (Moore et al, 2016; Mukherjee et al, 2016), we aimed for a version with enhanced brightness and targeting accuracy. First, we fused the ratiometric Epac-based high-affinity Fluorescence Resonance Energy Transfer (FRET) sensor EpacS$^{H187}$ (Klarenbeek et al, 2015) to the C-terminus of the cilia receptor Smoothened (Smo-EpacS$^{H187}$) (Fig. 5A). Although primarily enriched in the cilium, the presence of small amounts of the sensor in the plasma membrane enabled simultaneous recordings of cAMP concentration changes in both compartments. TIRF microscopy recordings of Smo-EpacS$^{H187}$ FRET ratio changes in MIN6 cells showed an increase of cAMP in both cilia and cytosol after addition of 10 μM of the adenylate cyclase (AC) activator forskolin. Donor (CFP) and acceptor (YFP) fluorophores exhibited opposite changes in intensity from the 442 and 540 nm emission wavelengths in both compartments, indicative of efficient FRET (Fig. 5B). It has been shown that the resting cAMP concentration is higher in the cilium than in the cytosol, at least in part due to high resting AC activity (Moore et al, 2016). However, we did not detect any difference in the resting FRET ratio between cilia and cytosol in MIN6 cells expressing either Smo-EpacS$^{H187}$ or a version where the cAMP sensor was instead targeted to the cilium using the 5HT$_6$ receptor (Fig. 5B). Similarly, no difference in resting cytosolic or ciliary cAMP concentrations was seen using a cAMP sensor based on the cilioplasmic protein Arl13b and the low-affinity cAMP sensor EpacS$^{H188}$ (Arl13b-188) (Appendix Fig. S2; Fig. 5B). In addition, we also developed a single wavelength cilia cAMP sensor based on Arl13b and the high-affinity cAMP sensor RFlincA (Ohta et al, 2018). Stimulation with either the phosphodiesterase (PDE) inhibitor IBMX or with forskolin resulted in an increase in RFlincA fluorescence in both cilia and cytosol, indicating increased cAMP concentration (Fig. 5C). However, the brightness of RFlincA was very low at resting conditions, which complicated experiments, and the FRET sensors outperformed the red sensor also with regards to signal-to-noise. Using Smo-EpacS$^{H187}$, we next determined the cAMP entry and exit kinetics into and out of primary cilia. Forskolin induced an immediate increase in cytosolic cAMP which was mirrored by an increase in the cilium, indicating either unrestricted diffusion between the two compartments or similar rates of cAMP synthesis (Fig. 5D,E). Washout of forskolin resulted in a lowering of the cAMP concentration that was slower than the increase, but also here the kinetics were similar in both cytosol and cilia, indicating again unrestricted diffusion or similar rates of cAMP turnover (Fig. 5D,F).

## The primary cilium of islet cells lacks a cAMP barrier

The spatiotemporal precision of signal propagation and the selectivity of signaling outcomes are fundamental in intracellular signaling. To determine how cAMP signaling, which is key to both cytosolic and ciliary function, differs between these two compartments, we exposed

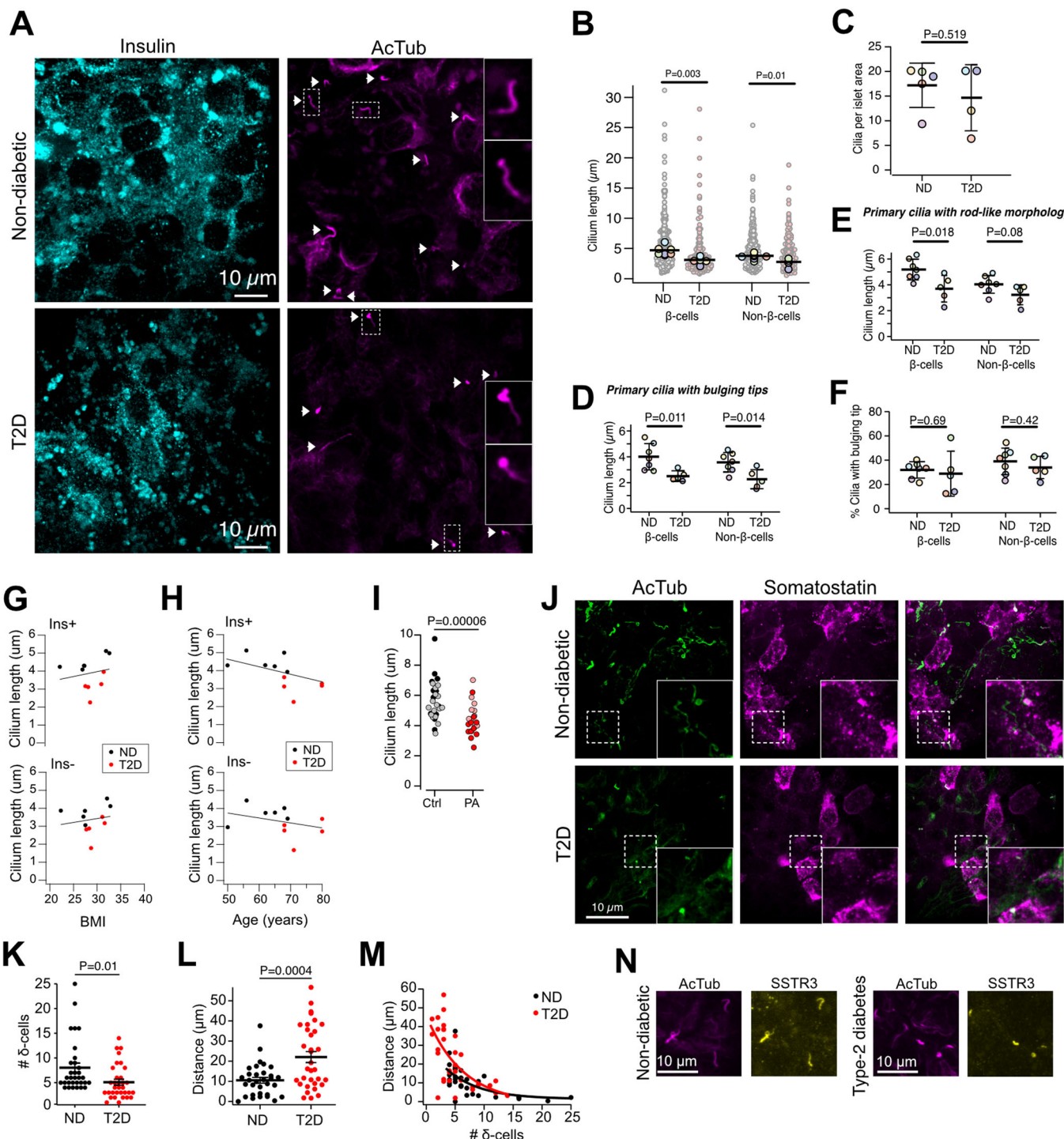

Smo-EpacS[H187]-expressing MIN6 cells to cAMP-elevating stimuli. To trigger cAMP increases selectively in the cytosol, we stimulated the cells with 100 nM GLP-1. Notably, the GLP-1 receptor was not cilia-localized, and agonist stimulation did not traffic GLP-1R to the primary cilium of islet cells (Fig. 5G; Appendix Fig. S2). GLP-1 induced a rise of cytosolic cAMP that was followed by a rise of ciliary cAMP with a delay of around 10 s (Fig. 5H,I). The addition of forskolin to activate ACs resulted in a simultaneous rise of cAMP in

cytosol and cilium, while inhibition of PDEs with IBMX caused a rise of cAMP in the cilium that preceded that of the cytosol (Fig. 5H,I). These results show that cAMP diffuses freely between the cytosol and cilium, consistent with previous reports (Marley et al, 2013; Truong et al, 2021). There was no difference in the Smo-EpacS[H187] response amplitudes between cilia and cytosol for any of the stimuli, indicating that the magnitude of the cAMP gradient between cytosol and cilium is without effect on ciliary cAMP entry (Fig. 5J).

**Figure 3. Reduced cilia length in type-2 diabetes.**

(A) Confocal microscopy images of human islets from a non-diabetic and a type-2 diabetic (T2D) organ donor immunostained against insulin (cyan) and acetylated tubulin (magenta) (white arrowheads point to cilia). (B) Quantifications of cilia length in insulin-positive and negative cells of non-diabetic and type-2 diabetic islets (means ± SD; ND: $N = 7$ donors; 10–20 islets per donor; 946 cilia. T2D: $N = 5$ donors; 10–20 islets per donor; 505 cilia; Student's two-tailed unpaired $t$ test). (C) Number of cilia per area in islets from non-diabetic and type-2 diabetic islets (means ± SD; ND: $n = 7$ donors; T2D: $n = 5$ donors; Mann–Whitney $U$ test). (D) Length of cilia with a swollen tip morphology in human islets from non-diabetic and typ-2 diabetic donors (means ± SD; ND: $n = 6$ donors; T2D: $n = 5$ donors; Mann–Whitney $U$ test). (E) Length of cilia with rod-like morphology in human islets from non-diabetic and typ-2 diabetic donors (means ± SD; ND: $n = 7$ donors; T2D: $n = 5$ donors; Mann–Whitney $U$ test). (F) Fraction of cilia with a swollen tip morphology in human islets from non-diabetic and typ-2 diabetic donors (means ± SD; ND: $n = 7$ donors; T2D: $n = 5$ donors; Mann–Whitney $U$ test). (G) Correlation between cilium length and BMI in insulin-positive (Ins + , top) and insulin-negative (Ins-, bottom) cells in human islets from non-diabetic (black) and type-2 diabetic (red) donors (ND: $n = 6$ donors; T2D: $n = 5$ donors). (H) Correlation between cilium length and donor age in insulin-positive (Ins + , top) and insulin-negative (Ins-, bottom) cells in human islets from non-diabetic (black) and type-2 diabetic (red) donors (ND: $n = 6$ donors; T2D: $n = 5$ donors). (I) Primary cilia length in human islets cultured for 7 days in the absence (Ctrl) or presence (PA) of 0.5 mM palmitic acid (means ± SEM; Ctrl: 30 islets; PA: 22 islets; Student's unpaired $t$ test; 2 donors). (J) Confocal microscopy images of human islet from non-diabetic and type-2 diabetic donors immunostained against acetylated tubulin (green) and somatostatin (magenta). (K) Number of δ-cells per islet area (means ± SEM; ND: 31 islets, 3 donors; T2D: 32 islets, 2 donors; Student's unpaired two-tailed $t$ test). (L) Shortest distance between primary cilia and δ-cells (means ± SEM; ND: 31 islets, 3 donors; T2D: 32 islets, 2 donors; Student's unpaired two-tailed $t$ test). (M) Correlation between number of δ-cells and shortest distance between δ-cells and primary cilia (ND: 31 islets, 3 donors; T2D: 32 islets, 2 donors). (N) Confocal microscopy images of human islets from a non-diabetic and a type-2 diabetic donor immunostained against acetylated tubulin (magenta) and SSTR3 (yellow). Source data are available online for this figure.

## Somatostatin lowers ciliary cAMP via SSTR3

Having demonstrated that cAMP can freely diffuse from the cytosol to the primary cilium, we next asked whether primary cilia can suppress cAMP formation independent of cytosolic cAMP. We expressed the cAMP sensor Arl13b-EpacS[H188], which is strongly enriched in primary cilia but also present in small amounts in the cytosol, in mouse islets and MIN6 pseudoislets and stimulated cAMP production by the addition of 1 μM forskolin. This caused a rise of cAMP in both cytosol and cilia, and the subsequent addition of 100 nM somatostatin lowered cAMP in both compartments (Fig. 6A,B). There was no significant difference in the response to forskolin between the cytosol and cilia, and somatostatin lowered cAMP back to basal levels in both compartments (Fig. 6B). Similar results were obtained in mouse islets and MIN6 pseudoislets. FFAR4 is a Gαs-coupled receptor that was recently shown to localize to MIN6 cell primary cilia and to promote cilia-specific increases in cAMP (Wu et al, 2021). To test if somatostatin could also counteract receptor-triggered cAMP formation in the cilium, we expressed the cilia-localized cAMP sensor in MIN6 pseudoislets and exposed them to the FFAR4-specific agonist TUG-891. This resulted in an increase in both cilia and cytosolic cAMP that was reversed by the subsequent addition of 100 nM somatostatin in the continued presence of TUG-891 (Fig. 6C). Although we did not observe cilia-specific effects of FFAR4 activation, these results still demonstrate that somatostatin can suppress receptor-mediated ciliary cAMP signaling. Next, we reduced the expression of SSTR3 in MIN6 pseudoislets by shRNA-mediated knockdown (Fig. 6D). To confirm the knockdown, we immunostained control and SSTR3 KD pseudoislets against acetylated tubulin, as a primary cilia marker, and SSTR3 and imaged the samples on a spinning-disk confocal microscope. Consistent with SSTR3 being primarily a ciliary receptor, we observed a significant reduction in SSTR3 immunoreactivity in the primary cilia of SSTR3 KD pseudoislets, but no change in the cytosolic SSTR3 immunoreactivity, which likely reflects non-specific labeling (Figs. 6E and EV3A–C). Next, we exposed control and SSTR3 KD pseudoislets expressing Arl13b-EpacS[H188] to 1 μM forskolin to stimulate cAMP production, followed by the addition of 100 nM somatostatin. Forskolin-induced cAMP increases in both cytosol and primary cilium of control and SSTR3 KD cells. While the subsequent addition of somatostatin lowered cAMP back to pre-stimulatory levels in the cytosol and cilium of control cells and the

cytosol of SSTR3 KD cells, it failed to do so in the cilia of SSTR3 KD cells (Fig. 6F,H). Similar results were obtained in MIN6 pseudoislets where SSTR3 expression was transiently reduced by siRNA-mediated knockdown (Fig. 6G). These results show that somatostatin lowers ciliary cAMP via activation of SSTR3.

## Somatostatin-induced lowering of ciliary cAMP is accompanied by ciliary Ca²⁺ signaling

Both plasma membrane- and cilia-localized SSTRs are Gαi-coupled and lower cAMP, but it is not known if activation of these receptors also involves additional signaling pathways. We recently found that the GABA[B1]-receptor, another Gαi-coupled receptor, localizes to primary cilia and couple to $Ca^{2+}$ influx through a non-canonical mechanism (Sanchez et al, 2023). To test if cilia-localized SSTRs engage in similar signaling, we transduced mouse islets and MIN6 pseudoislets with the cilia-targeted $Ca^{2+}$ sensor 5HT[6]-GGECO1 (Sanchez et al, 2023). The addition of 100 nM somatostatin initiated, after a delay of 5–10 min, $Ca^{2+}$ responses in the primary cilia that were not accompanied by changes in cytosolic $Ca^{2+}$ (Fig. 7A–D). siRNA-mediated knockdown of SSTR3 in MIN6 pseudoislets significantly reduced the response to somatostatin, indicating that increased ciliary $Ca^{2+}$ was initiated downstream of SSTR3 (Fig. 7E). The response to somatostatin stimulation was also completely prevented by overnight treatment with pertussis toxin to inhibit Gαi-dependent signaling (Figs. 7F and EV4A). These results indicate a connection between cAMP lowering and the initiation of $Ca^{2+}$ signaling in the primary cilium. To test this more directly, we exposed mouse islets expressing 5HT[6]-GGECO1 to 100 nM somatostatin, followed by the addition of 100 μM of the FFAR4 agonist TUG-891 or 100 nM of the Gαs-coupled PAK1 receptor agonist PACAP (pituitary adenylate cyclase-activating peptide). Consistent with an inverse relationship between cAMP and $Ca^{2+}$, we found that the somatostatin-induced $Ca^{2+}$ signaling was suppressed when cAMP was elevated (Figs. 7G and EV4B,C). Similar suppression was also seen when cytosolic cAMP levels were elevated by blue light-induced activation of bPAC (Fig. EV4D) (Stierl et al, 2011). Inhibition of protein kinase A (PKA), the major cAMP effector protein, with 10 μM H89 also induced, with some delay, $Ca^{2+}$ signaling in primary cilia of both mouse islet cell and MIN6 pseudoislets (Figs. 7H and EV4E). Together, these results show that cAMP lowering in the cilium is coupled to initiation of $Ca^{2+}$ signaling. Next,

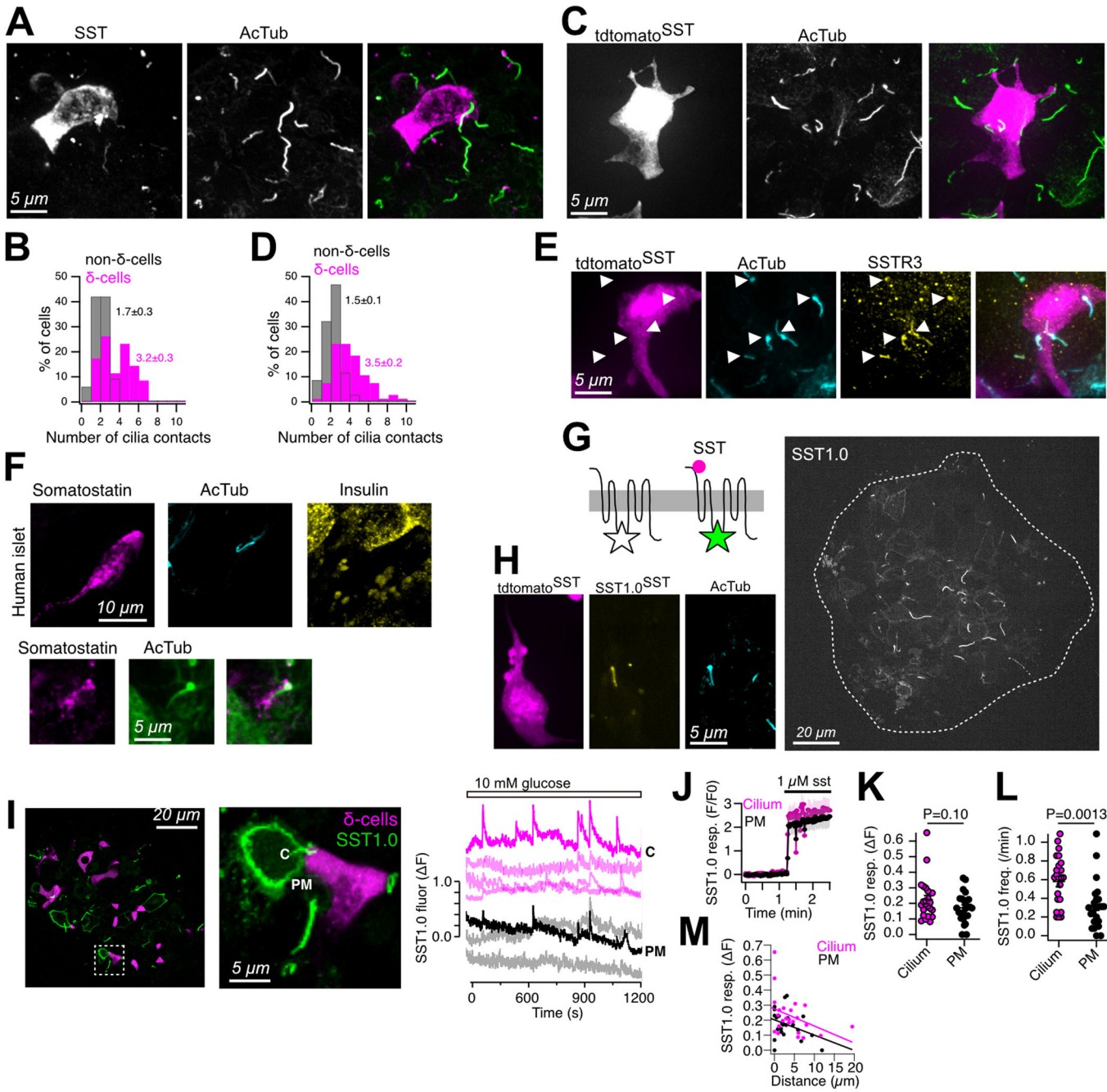

we wanted to test if endogenous somatostatin secretion could trigger ciliary $Ca^{2+}$ signaling and therefore stimulated mouse islets expressing $5HT_6$-GGECO1 with 10 nM ghrelin, a hormone that triggers $Ca^{2+}$-dependent somatostatin release (Adriaenssens et al, 2016; DiGruccio et al, 2016). The addition of somatostatin triggered a Ca2+ response in 65% of all islet cells, while ghrelin triggered a response in around 40% of these cilia (Fig. 7I,J). The responses to ghrelin were confined to cilia located in distinct regions of the islet, likely indicating that these cilia were near δ-cells (see Fig. 4). In some cases, we observed cells that responded to ghrelin with increases in cytosolic $Ca^{2+}$ (likely δ-cells) that were in the immediate vicinity of cilia that also showed response to ghrelin (Fig. 7K,L). These results show that somatostatin released

from δ-cells within intact mouse islets initiates ciliary $Ca^{2+}$ signaling in a cAMP-dependent manner.

## Somatostatin controls gene expression through a pathway involving the primary cilium, SSTR3, and GLI2

The Hedgehog signaling pathway is organized around the primary cilium and operates through the lowering of ciliary cAMP (Mukhopadhyay et al, 2013). Since we found that somatostatin is secreted onto cilia and lowers the ciliary cAMP concentration, we next investigated if somatostatin could cross-talk with the Hedgehog pathway. The latter stages of this pathway involve activation of

**Figure 4.  Somatostatin from islet δ-cells is released onto primary cilia.**

(A) Confocal microscopy image of a mouse islet immunostained for somatostatin and acetylated tubulin. (B) Quantification of the number of cilia in close proximity to somatostatin-positive δ-cells and non-δ-cells in mouse islets (means ± SEM; $n = 35$; $N = 7$; $P = 1.0E-5$, Student's paired $t$ test). (C) Confocal microscopy images of mouse islets with transgenic expression of tdtomato in somatostatin-positive δ-cells and immunostained for acetylated tubulin to visualize cilia. (D) Quantification of the number of cilia in close proximity to tdtomato-positive or negative cells in mouse islets (means±SEM; $n = 64$; $N = 10$; $P = 1.1E-12$, Student's paired $t$ test). (E) Confocal microscopy image of a mouse islet with transgenic expression of tdtomato in δ-cells (magenta) immunostained for acetylated tubulin (cyan) and SSTR3 (yellow). Arrowheads show SSTR3-positive cilia. (F) Confocal microscopy images from two human islets immunostained against somatostatin (magenta), acetylated tubulin (cyan/green) and insulin (yellow). (G) Schematic cartoon of SST1.0 (left) and a confocal microscopy image (right) of a mouse islet expressing SST1.0. (H) Confocal microscopy image from a mouse islet with transgenic expression of both tdtomato (magenta) and SST1.0 (yellow) in δ-cells and immunostained against acetylated tubulin (cyan). (I) Confocal micrograph of a mouse islet with transgenic expression of tdtomato in δ-cells (magenta) and infected with adenovirus to express SST1.0 (green). Boxed region is magnified to the right. Traces are from cilia and cilia-adjacent plasma membrane regions and show SST1.0 fluorescence changes in response to 10 mM glucose. The regions corresponding to traces "C" and "PM" are indicated in the confocal micrograph. (J) SST1.0 fluorescence change in a mouse islet stimulated with 1 μM somatostatin. Traces are means ± SEM for 19 cilia and 17 cilia-adjacent plasma membrane regions. (K) SST1.0 fluorescence change in the cilium (magenta) and cilium-adjacent plasma membrane (black) within mouse islets exposed to 10 mM glucose (means ± SEM; $N = 4$ islets; $n = 25$ cells; two-tailed Student's paired $t$ test). (L) Frequency of SST1.0 fluorescence changes in the cilium (magenta) and cilium-adjacent plasma membrane (black) within mouse islets exposed to 10 mM glucose (means ± SEM; $N = 4$ islets; $n = 25$ cells; two-tailed Student's paired $t$ test). (M) Correlation between SST1.0 response amplitude in the plasma membrane (black) and cilium (magenta) and distance to nearest δ-cell. Notice that response amplitude is reduced with increased distance to δ-cells. Source data are available online for this figure.

the GLI2 transcription factor, which then leaves the cilium to enter the nucleus (Niewiadomski et al, 2014). We therefore expressed GLI2-Halo$^{JFX650}$ in mouse islets, treated or not with 100 nM SMO agonist (SAG) for 18 h. Under resting conditions, GLI2-Halo$^{JFX650}$ was primarily cytosolic with some enrichment in the cilium, and the addition of SAG resulted in nuclear translocation of the transcription factor, consistent with Hedgehog pathway activation (Fig. 8A,B). Next, we treated the mouse islets cells with 100 nM somatostatin for 18 h and determined the extent of nuclear accumulation of GLI2-Halo$^{JFX650}$. We found that somatostatin was as efficient in inducing GLI2 nuclear entry as SAG alone, indicating that somatostatin signaling cross-talk with the Hedgehog pathway (Fig. 8A,B). Similar results were obtained in MIN6 cells expressing GLI2-GFP, where we also found that the combined effects of SAG and somatostatin were not additive (Fig. EV5A). Exposure to SAG or somatostatin was associated with a slight shortening of cilia (Fig. 8C). SAG-dependent activation of GLI2 involves exit of the Hedgehog receptor Patched from the cilium. Consistently, we observed a reduction in Patched immunoreactivity in the cilium after acute SAG stimulation, whereas Patched levels were unaffected by somatostatin stimulation (Fig. 8D). Next, we stably reduced the expression of SSTR3 in MIN6 cells expressing GLI2-GFP and exposed them to somatostatin for 18 h. Whereas somatostatin induced nuclear translocation of GLI2 in control cells, it failed to do so in cells with reduced expression of SSTR3 (Fig. 8E). These results indicate that somatostatin exert long-term effects on islet cells through ciliary signaling. Importantly, 18 h treatment with somatostatin did not result in receptor desensitization in the cilium, since acute addition of 100 nM somatostatin was still able to initiate ciliary Ca$^{2+}$ signaling (Fig. 8F–H). Hedgehog pathway activation has been shown to stimulate cilia Ca$^{2+}$ influx through a cAMP-dependent mechanism, and blocking this influx attenuated GLI2 processing and downstream transcription (Moore et al, 2016). To test if the somatostatin-induced GLI2 nuclear translocation depended on ciliary Ca$^{2+}$ increases, we expressed a cilia-localized Ca$^{2+}$ chelator (Arl13b-parvalbumin-mCherry) in MIN6 cells and exposed them to somatostatin for 18 h. There was a slightly increased fraction of cells with nuclear GLI2 under resting conditions, but this fraction did not increase further in the presence of somatostatin, indicating that the response is Ca$^{2+}$-dependent (Fig. 8I). Quantitative RT-PCR showed that SAG

reduced the expression of SSTR3 and GLI3 and increased the expression of GLI2, while somatostatin had no effect on GLI2 expression but increased GLI3 expression in a SSTR3-dependent manner (Fig. 8J–L). These results show that ciliary Hedgehog and somatostatin signaling to some extent involve distinct transcriptional changes. The Hedgehog pathway is important for cell specification and maintenance, both during development and in adulthood. We therefore determined how prolonged exposure to SAG or somatostatin influenced β-cell Ca$^{2+}$ and cAMP signaling. MIN6 pseudoislets expressing the cytosolic Ca$^{2+}$ indicator R-GECO1 were treated for 18 h with somatostatin, SAG or a combination of both followed by real-time recordings of glucose-induced Ca$^{2+}$ concentration changes. Control pseudoislets exhibited typical, regular slow Ca$^{2+}$ oscillations in response to glucose. Similar responses were seen in SAG-treated islets while somatostatin-treated islets exhibited more rapid, irregular Ca$^{2+}$ changes that were not obviously altered by simultaneous SAG treatment but were partially normalized following SSTR3 knockdown (Fig. EV5C–F). Next, the cytosolic cAMP FRET sensor EpacSH187 was expressed in MIN6 cells that were subsequently exposed to SAG, somatostatin or a combination of both for 18 h. cAMP production was subsequently stimulated with 100 nM GLP-1. In untreated cells, GLP-1 induced a pronounced increase in cAMP, and this response was strongly attenuated in cells treated with somatostatin, SAG or a combination of both (Fig. 8M). Experiments in knockdown cells showed that this effect of somatostatin required ciliary SSTR3 (Fig. 8N,O). Together, these results show that somatostatin can act as a cilia-dependent transcriptional regulator and that this mechanism operates in parallel to somatostatin-mediated suppression of insulin secretion to maintain functional β-cells (Fig. 8P).

# Discussion

In this study, we find that islet δ-cells are positioned near β-cell primary cilia, and that somatostatin secretion directly activates ciliary somatostatin receptors. This activation leads to lowering of the ciliary cAMP concentration and to inhibition of PKA, resulting in Ca$^{2+}$ influx that is restricted to the primary cilium. Reduced ciliary cAMP together with increased ciliary Ca$^{2+}$ promotes GLI2

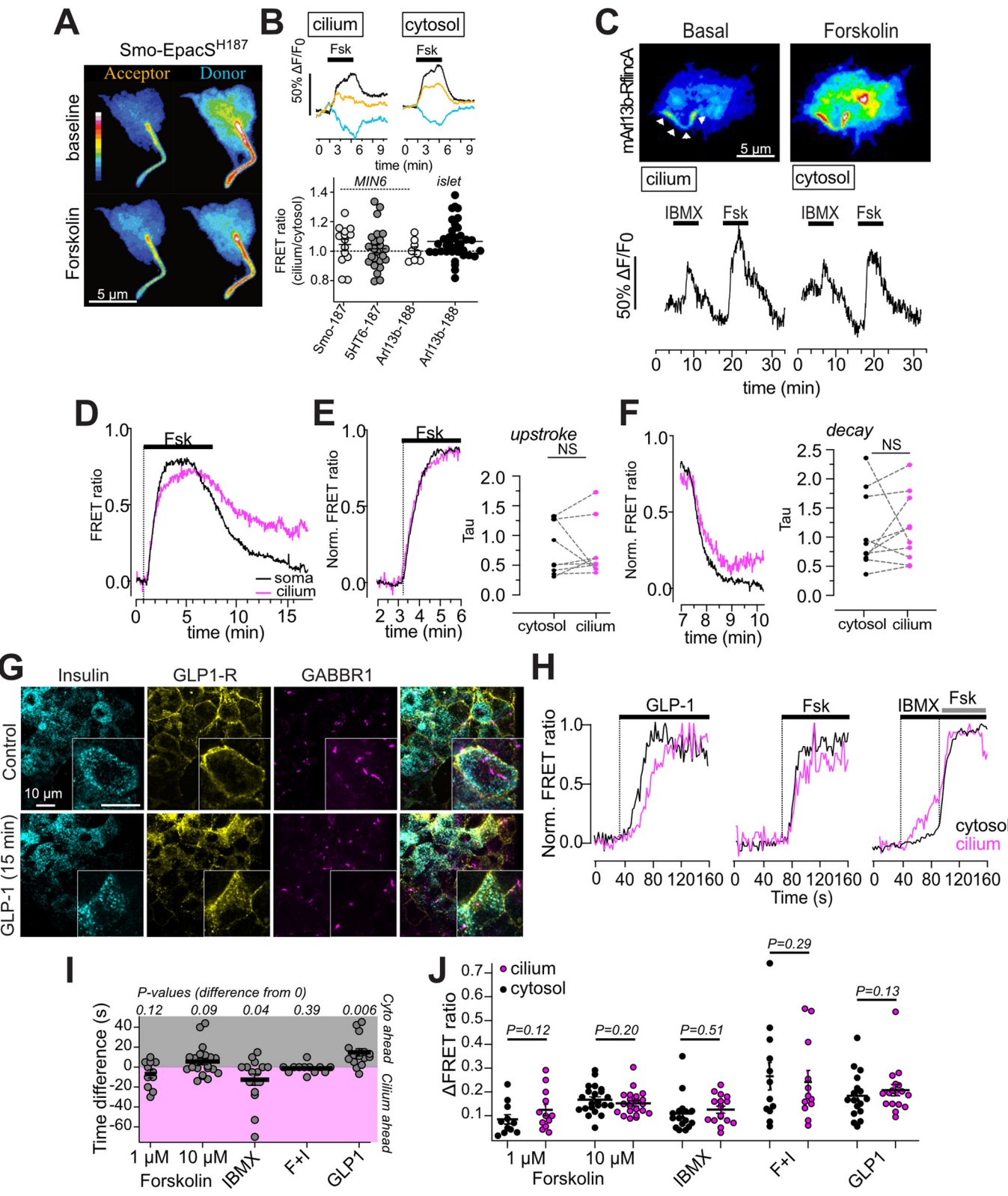

**Figure 5.   Direct measurements of ciliary cAMP.**

(A) TIRF microscopy images depicting a MIN6 cell expressing EpacS$^{H187}$, targeted to the cilium by Smoothened. Images on top are from resting conditions, and images at the bottom are in the presence of 10 μM forskolin. (B) Representative recording of EpacSH187 FRET ratio (CFP/YFP) changes in the cilium and cytosol in response to 10 μM forskolin. Below are shown the ratio (means ± SEM) between resting ciliary and cytosolic CFP/YFP FRET ratio in Smo-EpacS$^{H187}$ ($n = 14$ cells from four experiments), 5HT$_6$-EpacS$^{H187}$ ($n = 24$ cells from eight experiments) and mArl13b-EpacS$^{H188}$ ($n = 8$ cells in two experiments) expressing MIN6 cells and mArl13b-EpacS$^{H188}$ ($n = 39$ cells in five independent experiments) expressing mouse islet cells. (C) Pseudo-colored TIRF images of an islet cell showing an increase in ciliary and cytosolic cAMP in response to 10 μM forskolin and 100 μM IBMX reported with mArl13b-RflincA. Fluorescence changes over time are shown below. (D) TIRF microscopy recordings of Smo-EpacS$^{H187}$ FRET ratio in the cilium and cytosol of MIN6 cells following addition and washout of forskolin. Data presented as means for ten cells. (E) Amplitude-normalized traces from (D) highlighting the kinetics during forskolin-evoked cAMP increases in the cilia and cytosol Shown to the right are calculations of the time-constant for the forskolin-induced rise of cAMP. (F) Amplitude-normalized traces from (D) highlighting the kinetics during cAMP lowering in the cilia and cytosol following forskolin washout. Shown to the right are calculations of the time-constant for the forskolin-induced rise of cAMP. Statistical significance was assessed with the two-tailed paired Student´s $t$ test (no difference). (G) Representative confocal microscopy images of mouse islets immunostained against the GLP-1 receptor (yellow), insulin (cyan) and GABBR1 (cilia base; magenta) and treated or not with 100 nM GLP-1 for 15 min. (H) Representative TIRF microscopy recordings of Smo-EpacS$^{H187}$ FRET ratio changes in the cytosol (black) and cilium (magenta) of MIN6 cells exposed to 100 nM GLP-1, 10 μM forskolin, and 100 μM IBMX. (I) Difference in half-maximal cAMP increases in the cytosol and cilium in response to forskolin (1 and 10 μM), IBMX (50 μM), a combination of forskolin and IBMX (F + I) or GLP-1 (100 nM) (means ± SEM; $n = 11, 22, 16, 12,$ and 17 cells from three to six experiments). Statistical significance was assessed with one-sample $t$ test. (J) Means of ±SEM for the Smo-EpacS$^{H187}$ FRET ratio change in the cytosol (black) and cilia (magenta) in response to Forskolin, IBMX, forskolin+IBMX or GLP-1 (two-tailed paired Student´s $t$ test). Source data are available online for this figure.

activation and leads to its entry into the nucleus. Together, these findings show a structural basis for direct communication between δ-cells and primary cilia and demonstrate that somatostatin acts as a transcriptional regulator by engaging ciliary signaling pathways.

The primary cilium is a well-known target of somatostatin, and SSTR3 is a ubiquitously expressed ciliary receptor (Barbeito et al, 2021; Guadiana et al, 2016; Iwanaga et al, 2011). Despite this, relatively little is known about the role of primary cilia in the action of somatostatin. Somatostatin is produced and released from many locations, including the pancreatic islets of Langerhans, the gastrointestinal tract, and neurons in the central nervous system. Islet-derived somatostatin is cleared from tissues within minutes through the action of peptidases (Patel et al, 2011), and it is therefore considered primarily a local paracrine regulator. In the pancreatic islets, it suppresses both insulin and glucagon secretion by lowering the cytosolic cAMP concentration, but recent studies have also shown that primary cilia are involved in somatostatin-regulation of hormone secretion (Adamson et al, 2023; Hughes et al, 2020). β-cells with experimental deletion of primary cilia fail to respond to exogenous somatostatin with suppression of insulin secretion, and both inhibition and transient knockdown of SSTR3 partially recapitulate this phenotype. However, suppression of SSTR3 activity does not augment glucose-stimulated insulin secretion, indicating that this mechanism is not responsible for suppression of insulin secretion by endogenously released soma-tostatin, which instead involves plasma membrane-localized SSTR2 (Adamson et al, 2023; Kailey et al, 2012). It therefore seems likely that ciliary somatostatin signaling represents an alternative path-way of somatostatin action in the islet. We now show that there is indeed a structural basis for direct communication between β-cell primary cilia and δ-cells, likely facilitated by the extended shape of δ-cells that enables them to interact with other islet structures across large volumes (Arrojo e Drigo et al 2019). We did not provide evidence supporting the existence of synapses between δ-cells and cilia, similar to what has been observed between certain neurons and cilia, including β-cell cilia (Müller et al, 2024; Sheu et al, 2022; Wu et al, 2024). However, using a recently developed sensor for somatostatin based on SSTR5, we were able to show that glucose stimulation of endogenous somatostatin secretion led to activation of somatostatin receptors both on the cilia membrane and the plasma membrane. In fact, the sensor response was most

pronounced in the cilium, likely reflecting the proximity between the β-cell cilium and δ-cell.

SSTRs are Gαi-coupled receptors that lower cAMP (Dickerson et al, 2022; Patel et al, 2011). Although many recent studies have shown that cAMP can be locally generated and degraded in the primary cilium (Paolocci and Zaccolo, 2023; Truong et al, 2021), it is also known that it can freely diffuse between the cytosol and cilium (Jiang et al, 2019). We now find that cAMP, generated downstream of plasma membrane-localized GLP-1 receptors, readily enters the cilium with a lag-time of a few seconds. Pharmacological activation of ACs caused the simultaneous rise of cAMP in cilia and cytosol and identical lowering kinetics following washout of the drug, while inhibition of cAMP degradation caused a more rapid rise of cAMP in the cilium, perhaps reflecting higher PDE activity (Hansen et al, 2022). These results show that cilia can generate cAMP signals, but that these are closely linked to cytosolic changes. Nevertheless, we observed conditions where ciliary cAMP was regulated independently of cytosolic cAMP. For example, we found that somatostatin efficiently lowered both cytosolic and ciliary cAMP, but only the effect in the cilium was dependent on SSTR3. This suggests that activation of ciliary ACs can maintain elevated concentrations of cAMP in the cilium even in the face of accelerated degradation in the cytosol. In pancreatic β-cells, where cAMP increases are pivotal for normal insulin secretion, it is reasonable to think that primary cilia can sense environmental cues, such as somatostatin, and use these signals to counteract the cytosolic cAMP increases by local cAMP degradation. This specificity in ciliary cAMP signaling may be further facilitated by the activation of receptors exclusively found on the cilia membrane or by the targeted release of ligands toward the cilium, as was recently shown to occur at serotonergic synapses in the hippocampus (Sheu et al, 2022).

Lowering of ciliary cAMP is an integral part of the Hedgehog pathway, which alleviates PKA-mediated inhibitory phosphorylation of GLI transcription factors, thereby enabling their processing, ciliary exit, and nuclear entry. We now find that somatostatin, through activation of ciliary SSTR3, induces the exit of GLI2 from the cilium and promotes its nuclear entry. The effect of somatostatin was similar in magnitude to that of a Hedgehog pathway activator (SAG) and the effects of SAG and somatostatin were not additive. Similar observa-tions were recently made in epithelial cells overexpressing SSTR3

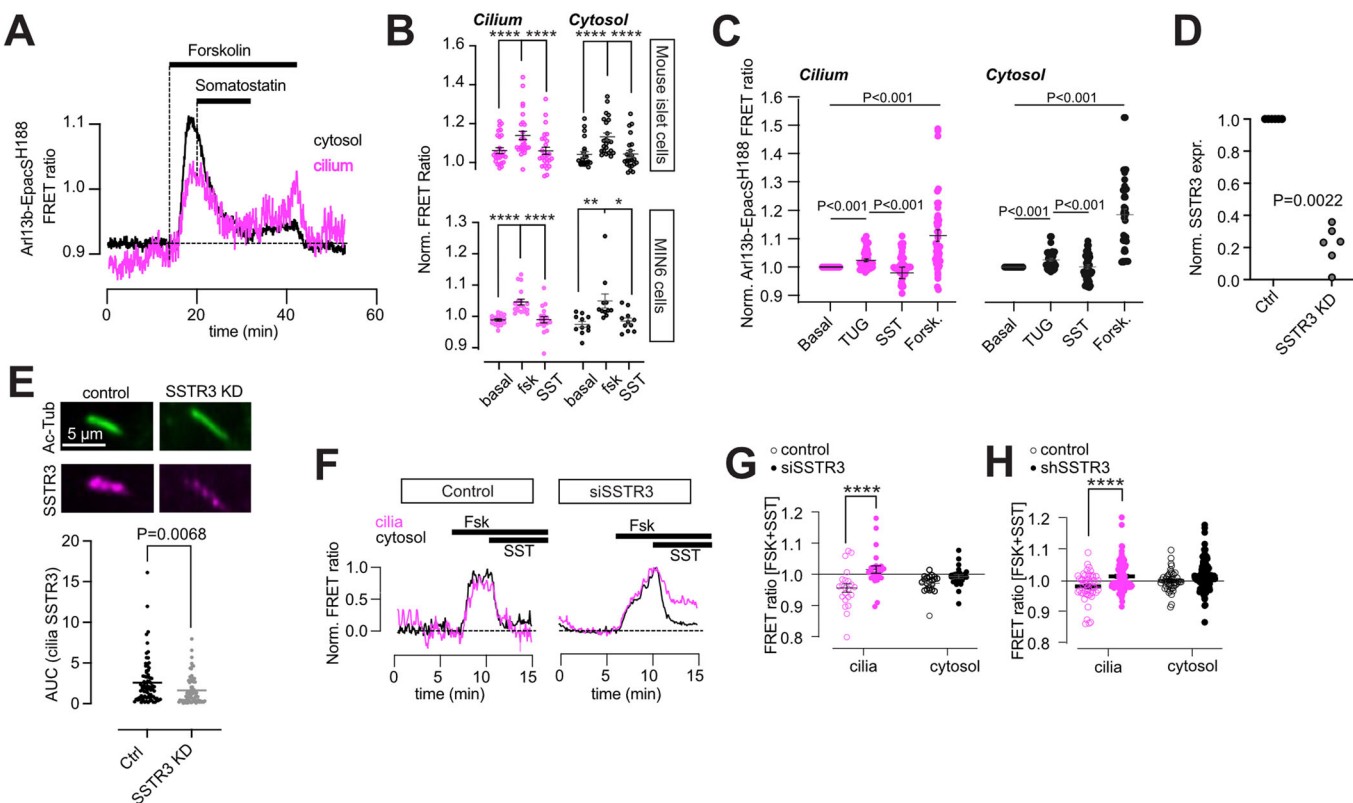

**Figure 6. Somatostatin lowers ciliary cAMP via SSTR3.**

(A) Representative TIRF microscopy recordings of cytosolic (black) and ciliary (magenta) FRET ratio changes in an islet cell expressing mArl13b-EpacS$^{H188}$ following exposure to 1 μM forskolin and 100 nM somatostatin. (B) Means of ±SEM for the mArl13b-EpacS$^{H188}$ FRET ratio change in mouse islet of cilia (magenta) and cytosol (black) in response to 1 μM forskolin, followed by the addition of 100 nM somatostatin. $n_{cytosol}$ = 23 cells, $n_{cilia}$ = 27 cilia, from three different animals. cilia basal-forskolin $P$ = 0.00000024, cilia forskolin-SST $P$ = 0.00000107, cytosol basal-forskolin $P$ = 0.00000342, cytosol forskolin-SST $P$ = 0.00000446; Sidak's multiple comparison test. Bottom panel shows means ± SEM for the mArl13b-EpacS$^{H188}$ FRET ratio change in MIN6 pseudoislet cilia (magenta) and cytosol (black) in response to 1 μM forskolin, followed by the addition of 100 nM somatostatin. $n_{cytosol}$ = 11 cells, $n_{cilia}$ = 18 cilia, from three different preparations. cilia basal-forskolin $P$ = 0.000012, cilia forskolin-SST $P$ = 0.000012, cytosol basal-forskolin $P$ = 0.009468, cytosol forskolin-SST $P$ = 0.013005, assessed with two-way ANOVA, Sidak's multiple comparison test. (C) Means of ±SEM for the mArl13b-EpacS$^{H188}$ FRET ratio change in MIN6 pseudoislet cilia (magenta) and cytosol (black) in response to TUG-891 (100 μM) followed by the addition of 100 nM somatostatin, and after addition of 10 μM forskolin. $n_{cytosol}$ = 33 cells, $n_{cilia}$ = 41 cilia, from three different preparations. Cilia; basal-TUG $P$ = 0.0000002, TUG-SST $P$ = 0.0000000000005, basal-forskolin $P$ = 0.00002, cytosol; basal-TUG $P$ = 0.0000009, TUG-SST $P$ = 0.00000002, basal-forskolin $P$ = 0.000007. Statistics was assessed with two-way ANOVA, Tukey's multiple comparison test. (D) Quantitative RT-PCR determination of the relative reduction in SSTR3 mRNA following shRNA-mediated knockdown in MIN6 cells (six different experiments; Mann–Whitney U test). (E) Confocal microscopy images of primary cilia from MIN6 pseudoislets transfected with control (left) or SSTR3 (right) siRNA and immunostained against acetylated tubulin (green) and SSTR3 (magenta). The SSTR3 immunoreactivity in the cilium is quantified below (means ± SEM; $n_{siCtrl}$ = 83, $n_{siSSTR3}$ = 86 cilia from three separate experiments; Mann–Whitney U test). (F) TIRF microscopy recordings of Arl13b-EpacS$^{H188}$ FRET ratio in control (left) and SSTR3 KD (right) MIN6 pseudoislets exposed to 1 μM forskolin and 100 nM somatostatin. Magenta color indicates recordings from the cilium and black from the cell body (means ± SEM for 53 cells for control, 46 cells for knockdown). (G) Quantifications of the Arl13b-EpacS$^{H188}$ FRET ratio from siSSTR3 transfected MIN6 pseudoislets in the presence of forskolin and somatostatin (normalized to pre-stimulatory level) (means ± SEM; 25 cilia-cytosol pairs for control and 22 cilia-cytosol pairs for knockdown. ****$P$ = 0.000064; two-way ANOVA, Uncorrected Fisher´s LSD.). (H) Quantifications of the Arl13b-EpacS$^{H188}$ FRET ratio following shRNA-mediated knockdown of SSTR3 in MIN6 pseudoislets exposed to forskolin and somatostatin (normalized to pre-stimulatory level) (means ± SEM; WT$_{cilia}$ = 50, WTcytosol = 44, KD$_{cilia}$ = 89, KD$_{cytosol}$ = 106. ****$P$ = 0.000058; two-way ANOVA, Uncorrected Fisher´s LSD). Source data are available online for this figure.

(Truong et al, 2021), and other studies have shown that the response to Hedgehog can be modulated by the simultaneous activation of ciliary Gαs and Gαi receptors (Pusapati et al, 2018). Somatostatin-induced GLI2 nuclear entry also depended on Ca$^{2+}$, since expression of a cilia-localized Ca$^{2+}$ chelator completely prevented somatostatin-induced GLI2 nuclear entry. Hedgehog stimulation has also been shown to result in ciliary Ca$^{2+}$ increases, but whether this increase is required for canonical Hedgehog signaling appears to be cell-type or context-dependent (Delling et al, 2013; Belgacem and Borodinsky, 2011; Teperino et al, 2012; Moore et al, 2016). The mechanism likely involves PKA, since acute inhibition of this enzyme also triggered ciliary Ca$^{2+}$

signaling, and Ca$^{2+}$ in turn may inhibit AC5/6, as has been shown in other cell types (Moore et al, 2016), in a positive feedback loop. Consistently, both Ca$^{2+}$ increases and cAMP lowering have been shown to promote cilia shortening, but whether the two messengers operate through a common pathway is not known (Besschetnova et al, 2010). GLI-dependent transcription is important during pancreas development and controls the expression of numerous genes that define endocrine lineage and β-cell identity, including PAX6 and NKX6.1. Dysregulation of these genes in adulthood lead to loss of β-cell function due to dedifferentiation and apoptosis, but exactly how the expression of these genes is maintained is not well-understood

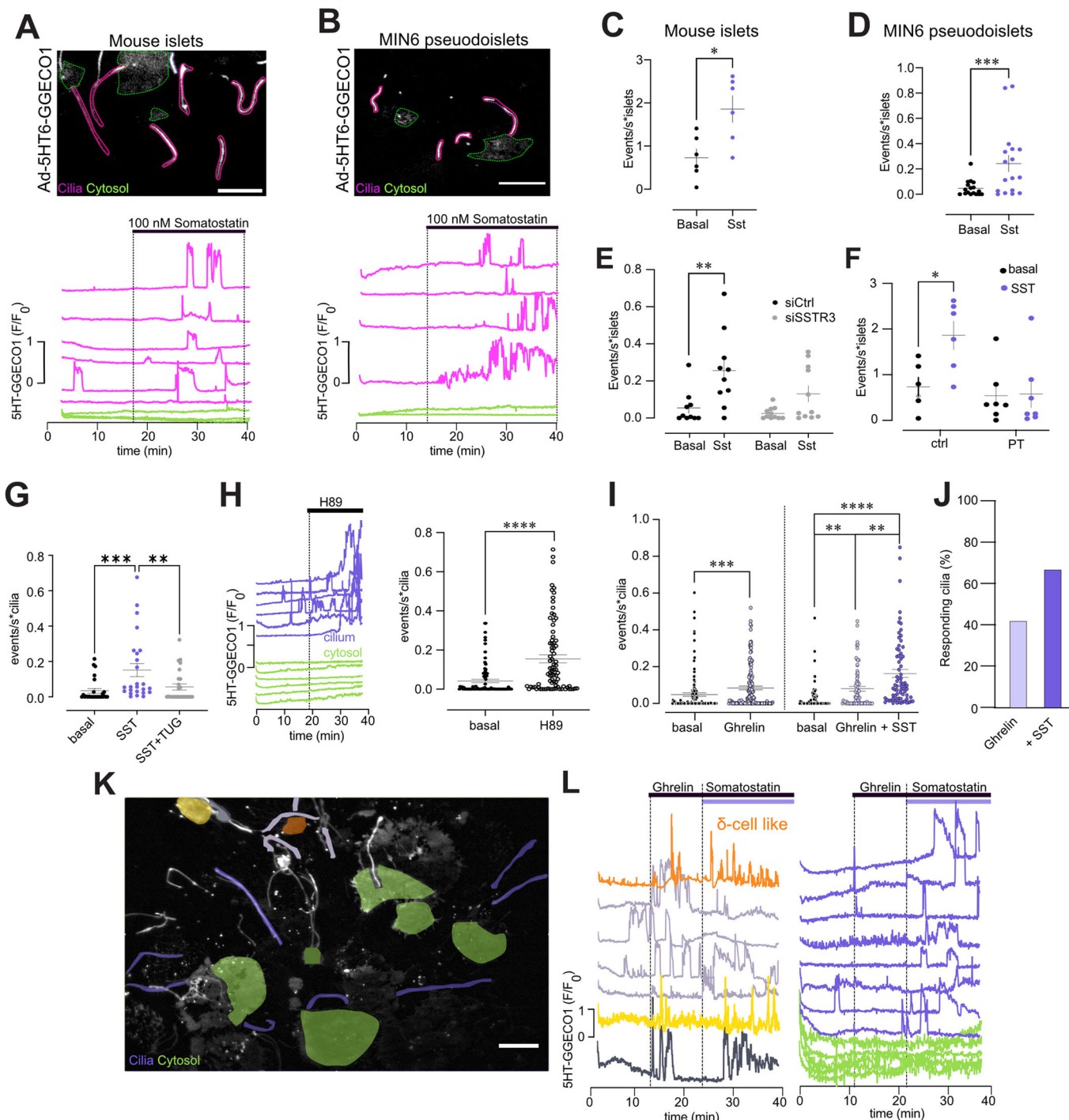

(Cervantes et al, 2010; Lau and Hebrok, 2010). Somatostatin has been shown to control gene expression in β-cells downstream of SSTRs (Zhou et al, 2014). Our results indicate that somatostatin may exert some of these effects trough control of cilia-dependent transcription. Somatostatin secretion is glucose-dependent, but the dose response is less distinct than for the other islet hormones and secretion is observed throughout the physiological glucose range (Hauge-Evans et al, 2009). This means that somatostatin may exert a tonic effect on ciliary receptors that enable GLI-dependent transcription and maintenance

of a highly differentiated phenotype, probably facilitated by the lack of desensitization of SSTR3 observed in the present study. Understanding how somatostatin controls β-cell function beyond the acute effects on hormone secretion represents an important future research direction.

Type-2 diabetes is characterized by functional impairment of β-cells, and recent studies indicate that dysregulation of cilia genes may contribute to both disease development and progression (Kluth et al, 2019; Walker et al, 2023). We now show that islets from type-2 diabetic organ donors are characterized by shorter cilia. Regulation of cilia

**Figure 7. SSTR3 activation initiates ciliary Ca²⁺ signaling.**

(A) TIRF microscopy image of a mouse islet expressing 5HT$_6$-GGECO1. Traces below show the GGECO1 fluorescence change in cilia (magenta) and cytosol (green) in response to 100 nM somatostatin. (B) TIRF microscopy image of a MIN6 pseudoislet expressing 5HT$_6$-GGECO1. Traces below show the GGECO1 fluorescence change in cilia (magenta) and cytosol (green) in response to 100 nM somatostatin. (C) Event count of all cilia Ca²⁺ changes in mouse islet cells under resting conditions and following addition of 100 nM somatostatin (means ± SEM; $n = 6$ islets, 72 cilia; $P = 0.0312$ Wilcoxon test). (D) Event count of all cilia Ca²⁺ flashes in MIN6 pseudoislet cells under resting conditions and following addition of 100 nM somatostatin (means ± SEM; $n = 17$ islets, 95 cilia; $P = 0.0008$ Wilcoxon test). (E) Event count of ciliary Ca²⁺ changes of MIN6 pseudoislets following the addition of 100 nM somatostatin in control cells (black) and SSTR3 KD cells (magenta) (means ± SEM; $n_{ctrl} = 10$ islets and 40 cilia, $n_{KD} = 11$ islets and 30 cilia from three different preparations, $P_{sictrl} = 0.003$, no change in siSSTR3, assessed by Sidak´s multiple comparison test). (F) Event count of all ciliary Ca²⁺ changes in mouse islet cells that were either cultured under control condition or in the presence of pertussis toxin for 18 h and exposed to 100 nM somatostatin (means ± SEM; $n_{ctrl} = 6$, 72 cilia, $n_{PT} = 7$, 65 cilia; islets from four different preparations, SST response in control $P = 0.0212$, SST response in Pertussis-toxin treated islets not significant, assessed by Sidak´s multiple comparison). (G) Event count of ciliary Ca²⁺ changes in mouse islet cells in response to 100 nM somatostatin, followed by the addition of 100 µM TUG-891. Means ± SEM for 25 cilia, from three different animals. ***$P = 0.0004$, **$P = 0.0021$; two-way ANOVA, Dunn's multiple comparison test. (H) Traces show the 5HT$_6$-GGECO1 fluorescence change in cilia (purple) and cytosol (green) of mouse islet cells. Event count of ciliary Ca²⁺ changes from mouse islet cells in response to 10 µM H89 are shown to the right (means ± SEM; 8 islets; 82 cilia; 3 different preparations. ****$P = 0.000000001$, Wilcoxon-matched pair $t$ test). (I) Event count of ciliary Ca²⁺ changes from mouse islets in response to 100 nM ghrelin (means ± SEM; 13 islets; 131 cilia from seven different preparations. ***$P = 0.0008$, Wilcoxon-matched pair $t$ test). To the right is shown event count of ciliary Ca²⁺ flashes from mouse islet in response to 100 nM ghrelin, followed by 100 nM SST (means ± SEM; 8 islets; 72 cilia from five different preparation. Dunn's multiple comparisons test; basal-ghrelin $P = 0.0035$, basal-SST $P = 0.0000000005$, ghrelin-SST $P = 0.0053$). (J) Percentage of cilia in mouse islets that exhibit Ca²⁺ changes in response to SST ($n = 72$) or Ghrelin ($n = 46$). (K) TIRF microscopy image of a mouse islet expressing 5HT$_6$-GGECO1. Highlighted areas are: yellow and orange (δ-cells), green (β-cells), light purple (cilia close to δ-cells), purple (cilia far from δ-cells). (L) Ca²⁺ concentration changes within the shaded areas in (K) in response to 100 nM ghrelin and 100 nM somatostatin. Scale bars: 5 µm. Source data are available online for this figure.

length is a dynamic process important for controlling the strength of ciliary signaling (Keeling et al, 2016; Song et al, 2018), and shortening of cilia has been observed under various pathologic conditions, including experimental diabetes (Alhassen et al, 2022; Chinipardaz et al, 2023) and been shown to contribute to the development of obesity (Oya et al, 2024). From our current experiments, we cannot conclude whether cilia shortening is a cause or a consequence of type-2 diabetes, but nevertheless, this shortening is likely to influence ciliary signaling by reducing both signal strength and the cilium's ability to interact with islet structures, including δ-cells. The latter is likely exacerbated by the reduction in δ-cell numbers that we observe in isolated islets from donors with type-2 diabetes. Although similar reductions in δ-cell numbers have been observed in some studies, others report normal δ-cell numbers in type-2 diabetes (Cohrs et al, 2020; Kothegala et al, 2023). Somatostatin secretion has also been shown to be reduced in both diabetic ob/ob and high fat diet-fed mice (Kellard et al, 2020; van der Meulen et al, 2015), and a recent study showed reduced numbers of ghrelin-secreting cells and reduced circulating levels of ghrelin in human type-2 diabetes (Lindqvist et al, 2020), which may explain the impaired somatostatin release.

This study highlights the role of islet cell primary cilia as sensors of the islet environment and shows that somatostatin is one environmental signal that can impact β-cell function through ciliary signaling. We believe that the cilium integrates many different signals through cilia-localized receptors and converts these to second messenger signals that are relayed to the cell body to maintain β-cell function. Primary cilia may therefore play a

crucial role in type-2 diabetes development as a failing link between intrinsic regulation and changes within the islet microenvironment.

# Methods

## Plasmids, adenoviruses, and reagents

The following plasmids were used; Gli2-GFP (Addgene plasmid #37672), 5HT$_6$-G-GECO (Addgene plasmid #47499), EpacSH187 and EpacSH188 (gift from Kees Jalink, Netherlands Cancer Institute, Amsterdam, Netherlands (Klarenbeek et al, 2015), RflincA kind gift from Kazuki Horikawa, The Institute of Scientific and Industrial Research, Osaka University, Japan (Ohta et al, 2018), Smo-mCherry (Addgene plasmid #55134), pGEM-Heh_bPAC-myc (Addgene plasmid #28134 gift from Peter Hegemann, Humboldt University, Berlin Germany) (Stierl et al, 2011). mArl13b-187 and mArl13b-188 were generated by PCR amplification of pCSDest2-arl13b-PVALB-cherry (gift from Zhaoxia Sun, Yale University; (Li et al, 2016)) using the following primers: HindIII-mArl13b-Fwd (5´-GCACAAGCTTGC-CACCATGTTCAATCTGATGGCGAAC-3´) and mArl13-NotI-rv (5´-CAGCGGGCCGCCATTGCCCCGGAAATGACATCATG-3´) followed by digestion of PCR product with HindIII and NotI andligation into upstream of EpacSH187 and EpacSH188 respectively. Smo-RflincA was generated by PCR amplification of pcDNA4HMB-RFlincA using

**Reagents and tools table**

| Reagent/resource | Reference or source | Identifier or catalog number |
|---|---|---|
| **Experimental models** | | |
| MIN6 cells (*M. musculus*) | Yamamura Lab | Miyazaki et al, 1990 |
| C57BL/6J | Scandbur, Denmark | N/A |
| **Recombinant DNA** | | |
| pCELFmGFP-Gli2 | Addgene | #37672 |

| Reagent/resource | Reference or source | Identifier or catalog number |
|---|---|---|
| 5HT6-G-GECO1.0 | Addgene | #47499 |
| EpacSH187 | Kees Jalink | Klarenbeek et al, 2015 |
| EpacSH188 | Kees Jalink | Klarenbeek et al, 2015 |
| pcDNA4HMB-RFlincA | Kazuki Horikawa | Ohta et al, 2018 |
| mCherry-Smoothened1-N18 | Addgene | #55134 |
| pGEM-Heh_bPAC-myc | Addgene | #28134 |
| pCSDest2-arl13b-PVALB-cherry | Zhaoxia Sun | Yuan et al, 2015 |
| mArl13b-187 | In this study | Table 1 |
| mArl13b-188 | In this study | Table 1 |
| Smo-RflincA | In this study | Table 1 |
| Smo-187 | In this study | Table 1 |
| 5HT6-187 | In this study | Table 1 |
| CMV-R-GECO1 | Addgene | 32444 |
| Ad-CMV-R-GECO1 | Vector Biolabs | Custom-made |
| Ad-CMV-5HT6-GGECO | Vector Biolabs | 20230117T#19 |
| Ad-CMV-Halo-Gli2 | VectorBuilder | VB240611-1532shj |
| Ad-CMV-mArl13b-188 | VectorBuilder | Custom-made |
| Ad-CMV-EpacS [H187] | Vector Biolabs | Custom-made |
| Ad- CMV-bPAC-myc | Vector Biolabs | Custom-made |
| **Antibodies** | | |
| Mouse anti-acetylated tubulin (1:500) | Sigma | T7451 |
| Rabbit anti-acetyl-α-tubulin (K40) (1:500) | BioNordika | 5335 |
| Rabbit anti-SSTR3 (1:200) | Elabscience | E-AB-16077 |
| Mouse anti-Arl13b (1:300) | Abcam | ab136648 |
| Rabbit anti-Somatostatin (1:500) | Dako | A0566 |
| Guinea pig anti-insulin (1:500) | Dako | A0564 |
| Mouse anti-SSTR5 (1:200) | Abcam | 66772-1-Ig |
| Rabbit anti-SSTR2 (1:200) | Abcam | ab134152 |
| Rabbit anti-Pericentrin (1:200) | Abcam | ab4448 |
| Rabbit anti-GABBR1 (1:200) | Alomone labs | AGB001AN102 |
| Mouse anti-GLP-1R (1:10) | Developmental Studies Hybridoma Bank | Mab 7F38-s |
| Chicken anti-GFP (1:500) | Abcam | AB16901 |
| Goat Anti-mouse Alexa Fluor 568 (1:500) | Invitrogen | A11004 |
| Goat Anti-mouse Alexa Fluor 488 (1:500) | Invitrogen | A28175 |
| Donkey Anti-mouse Alexa Fluor 647 (1:500) | Invitrogen | A31571 |
| Donkey Anti-rabbit Alexa Fluor 488 (1:500) | Invitrogen | A32790 |
| Goat Anti-rabbit Alexa Fluor 568 (1:500) | Invitrogen | A11011 |
| Donkey Anti-rabbit Alexa Fluor 647 (1:500) | Invitrogen | A32795 |
| Goat anti-Guinea Pig Alexa Fluor 488 (1:500) | Invitrogen | A11073 |
| Goat anti-Guinea Pig Alexa Fluor 568 (1:500) | Invitrogen | A11075 |
| Goat anti-Guinea Pig Alexa Fluor 647 (1:500) | Invitrogen | A21450 |
| Goat anti-Chicken Fluor 488 (1:500) | Invitrogen | A-11039 |
| Anti-mouse Abberior Star 580 (1:300) | Abberior GmbH | ST580 |
| Anti-rabbit Abberior Star Red (1:300) | Abberior GmbH | STRED |

| Reagent/resource | Reference or source | Identifier or catalog number |
| --- | --- | --- |
| **Oligonucleotides and other sequence-based reagents** | | |
| ON-TARGETplus Mouse Gabbr1 -smartpool | Dharmacon | L-057519-02-0005 |
| ON-TARGETplus Non-targeting Pool<br>5′-UGGUUUACAUGUCGACUAA-3′<br>5′-UGGUUUACAUGUUGUGUGA-3′<br>5′-UGGUUUACAUGUUUUCUGA-3′<br>5′-UGGUUUACAUGUUUUCCUA-3′ | Dharmacon | D-001810-10-05 |
| ON-TARGETplus Mouse SSTR3 -smartpool<br>5′-AGAUCAGGGCCCAAGUUAA-3′<br>5′-GCUGUGUGGUCUCGGCCAA-3′<br>5′-GUAAAGGUGCGGUCGACCA-3′<br>5′-GGGCUGUUGUCAGGGAGUA-3′ | Dharmacon | L-043208-03-0005 |
| pLV[shRNA]- mCherry:T2A:<br>Puro-U6>mSstr3[shRNA#1]<br>5′-GCACACTGAGCCATCTGTAA<br>GCTCGAGCTTACAGATGGCTCAGTGTGC-3′ | VectorBuilder | VB900139-1248bdp<br>Table 1 |
| pLV[shRNA]-control virus<br>5′-CCTAAGGTTAAGTCGCCCTCGCTCGAGCGA<br>GGGCGACTTAACCTTAGG-3′ | VectorBuilder | VB010000-0009mxc<br>Table 1 |
| SSTR3 qPCR primers<br>(target: NM_001356961.1)<br>Fwd: 5′-GAGGGCTTCCAT<br>TTCCCAGG-3′<br>Rev: 5′-  GTTGCCACTACT<br>CCCACCTC-3′ | In this study | |
| Gli2 qPCR primers<br>(target: NM_001081125.1)<br>Fwd: 5′-CAGCAGGACAGC<br>TACCAACA-3′<br>Rev: 5′-TTGCCCAGAACG<br>TACTCCAC-3′ | In this study | |
| Gli3 qPCR primers<br>(target: NM_008130.3)<br>Fwd: 5′-GCCCTCGACGTCT<br>AGTGATG-3′<br>Rev: 5′-TCCCACGGTAAGG<br>GAGAGAG-3′ | In this study | |
| Cloning primers:<br>mArl13b-fwd: GCACAAGCTTGCCACCATGTTCAATCTGATGGCGAAC<br>mArl13b-rev: CAGCGGGCCGCCATTGCCCCGGAAATGACATCATG<br>Age1-RFlincA-fwd:<br>GCACCGGTAGCCACCATGGGTAGGAGGCGACGAGGTGCTATCAGCG<br>RFlincA-rev: GCGCGGCCGCCACTGTGCTGGATATCTG<br>Smo-HindIII-fwd: CCCAAGCTTGCCACCatggccgctggc<br>Smo-Not1-rev: CTCTgcggccgcCATGGTGGCGACCGGTGG<br>Smo-fwd: TATATAAAGCTTATGGTTCCAGAGCCCGGCCCTGT<br>Smo-rev: TATATAGCGGCCGCGGATCCTCCTCCTGCGCTACC | Target: mouse Arl13b in pCSDest2-arl13b-PVALB-cherry. Ligated to EpacSH188/EpacSH187 using HindIII and Not1 to make mArl13b-EpacSH187/188.<br>Target: pcDNA4HMB-RFlincA. Ligated to Smo-N18-mCherry to make Smo-RFlincA.<br>Target: Smo in Smo-N18-mCherry. Ligated to EpacSH187 using HindIII and Not1 to make Smo-EpacSH187.<br>Target: EpacSH187 in Smo-EpacSH187. Ligated to 5HT6-GGECO1 by replacing GGECO1 to make 5HT6-EpacSH187. | |
| **Chemicals, enzymes, and other reagents** | | |
| DPBS | Gibco | 14190250 |
| Paraformaldehyde 16% | Thermo Scientific Chemicals | 043368.9M |
| Triton™ X-100 Surface-Amps™ Detergent Solution | Thermo Scientific | 28314 |

| Reagent/resource | Reference or source | Identifier or catalog number |
| --- | --- | --- |
| ProLong Gold Antifade Mountant | Thermo Scientific | P36934 |
| Bovine Serum Albumin Fraction V | Sigma-Aldrich | A8806 |
| Agarose I | Thermo Scientific | 17850 |
| Lipofectamine 2000 | Invitrogen | 11668027 |
| Lipofectamine RNAiMAX | Invitrogen | 13778150 |
| OptiMEM I Reduced Serum Medium | Gibco | 31985070 |
| Dulbecco's modified Eagle's medium (DMEM) | Gibco | 61965059 |
| RPMI 1640 Medium | Gibco | 21875034 |
| RPMI 1640 Medium, no glucose | Gibco | 11879020 |
| CMRL Medium, no glutamine | Gibco | 21530027 |
| Fetal Bovine Serum | Gibco | A5256801 |
| GlutaMAX Supplement | Gibco | 35050038 |
| Penicillin-Streptomycin | Gibco | 15140122 |
| Trypsin-EDTA, phenol red | Gibco | 25300054 |
| Palmitic acid | Sigma-Aldrich | P0500-10G |
| 2-Mercaptoethanol | Gibco | 31350010 |
| JFX650 Halotag Ligand | Lavis Lab (Janalia Research Campus) | N/A |
| Hoechst 33342 | Thermo Scientific | 62249 |
| 5X siRNA Buffer | Dharmacon | B-002000-UB-100 |
| Molecular Grade Rnase-free water | Dharmacon | B-003000-WB-100 |
| Somatostatin | Sigma-Aldrich | S9129 |
| GLP-1 (7-36) | Sigma-Aldrich | G8147 |
| Forskolin | Sigma-Aldrich | F3917-10MG |
| 3-isobutyl-1-methylxanthine (IBMX) | Sigma-Aldrich | I5879 |
| TUG-891 | Tocris | TUG-891 |
| Pertusis Toxin | Tocris | 3097 |
| H89 dihydrochloride | Tocris | 2910 |
| Ghrelin | Cayman chemicals | 15072 |
| PACAP 6-38 | Tocris | 3236 |
| SAG dihydrochloride | Tocris | 6390 |
| Poly-L-lysine hydrochloride | Sigma-Aldrich | P2658 |
| HindIII-HF | New Englands Biolabs | R3104S |
| NotI-HF | New Englands Biolabs | R3189L |
| AgeI-HF | New Englands Biolabs | R3552S |
| Phusion High Fidelity DNA polymerase | New Englands Biolabs | M0530L |
| **Software** | | |
| Prism10 | GraphPad Software | N/A |
| MetaFluor | Molecular Devices | N/A |
| MetaMorph | Molecular Devices | N/A |
| ImageJ | Fiji | N/A |
| **Other** | | |
| Cover glass, Menzel Gläser | WVR | 631-1346 |
| Microscope slides | WVR | 630-1985 |
| NucleoSpin Plasmid | Macherey-Nagel | 740588.50 |

| Reagent/resource | Reference or source | Identifier or catalog number |
|---|---|---|
| NucleoBond Xtra Midi | Macherey-Nagel | 740410.50 |
| NucleoSpin™ RNA Plus XS | Macherey-Nagel | 740990.50 |
| NucleoSpin™ Gel and PCR Clean-up XS Columns | Macherey-Nagel | 17122806 |
| AriaDx Real-time PCR System | Agilent | K8930AA |
| FIB-SEM dataset of a mouse pancreas | OpenOrganelle | https://janelia.figshare.com/articles/dataset/Near-isotropic_reconstructed_volume_electron_microscopy_FIB-SEM_of_P7_mouse_pancreas_jrc_mus-pancreas-4_/23411843 |
| iTaq Universal SYBR Green One-Step Kit | Bio-Rad | 172-5151 |
| Rapid DNA Dephos & Ligation Kit | Roche | RDLIG-RO |
| DH5α Competent Cells | Invitrogen | 18265017 |

primer set of AgeI-RflincA-fwd (5´- GCACCGGTAGCCACCATG GGTAGGAGGCGACGAGGTGCTATCAGCG-3´) and RflincA-rv- (5´-GCGCGGCCGCCACTGTGCTGGATATCTG-3´) followed by digestion of amplified PCR product with AgeI and NotI and ligation C-terminal to Smo in Smo-mCherry (Addgene plasmid 55134, kind gift from Michael Davidson). mArl13b-RflincA was generated by PCR amplification of pCSDest2-arl13b-PVALB-cherry using primers targeted for NheI-mArl13b-fw (5´-GTAGCTAGCGCCACCATGTT-CAGTCTGATGGCC-3´) and mArl13b-AgeI-rv (5´-GCACCGGTCC CTTTGAGATCGTGTCCTG-3´) followed by digestion of amplified mArl13b with NheI and AgeI and ligation N-terminal to Rflinca. mArl13b-188, 5HT6-GGECO (Sanchez et al, 2023), bPAC and Halo-Gli2 were packed in E5 serotype adenovirus under the control of cytomegalovirus (CMV) promoter and produced by Vector Biolabs (Malvern, PA) or VectorBuilder. All salts and chemicals were from Sigma-Aldrich/Merck if not stated otherwise. GLP-1, forskolin, IBMX, were from Tocris. The following primary antibodies were used in the study; acetylated tubulin (T745, host: mouse, 1:500; Sigma-Aldrich), Acetyl-α-Tubulin (K40) (5335, host: rabbit, 1:500; BioNordika), SSTR3 (E-AB-16077, host: rabbit, 1:200; Elabscience), Arl13b (ab136648, host: mouse, 1:300; Abcam), SST (A0566, host: rabbit, 1:500, Dako), Insulin (A0564, host: guinea pig, 1:500; Dako), SSTR5 (66772-1-Ig, host: mouse, 1:200, Proteintech), SSTR2 (ab134152, host: rabbit, 1:200, Abcam), Pericentrin (ab4448, Abcam, host: rabbit, 1:200), GABBR1 (AGB001AN102, Alomone labs, host: rabbit, 1:200), GLP-1R (Mab 7F38-s, Developmental Studies Hybridoma Bank, host: mouse, 1:10) and GFP (AB16901, Abcam, host: chicken, 1:500). Secondary antibodies for confocal anti-mouse 568 nm (A11004; 1:500; Invitrogen), anti-mouse 488 nm (A28175; 1:500; Invitrogen), anti-mouse 647 nm (A31571; 1:500; Invitrogen), anti-rabbit 488 nm (A32790; 1:500; Invitrogen), anti-rabbit 568 nm (A11011; 1:500; Invitrogen), anti-rabbit 647 nm (A32795; 1:500; Invitrogen), anti-guinea pig 488 nm (A11073; 1:500; Invitrogen), anti-guinea 568 nm (A11075; 1:500; Invitrogen), anti-guinea pig 647 nm (A21450; 1:500; Invitrogen) and anti-chicken 488 nm (A-11039, 1:500, Invitrogen). The following secondary antibodies were used for super-resolution microscopy; anti-mouse Abberior Star 580 and anti-rabbit Abberior Star Red (Abberior GmbH; 1:300).

## MIN6 cell culture and transfection with siRNAs and plasmids

The mouse β-cell line MIN6 (passages 18–30; not recently authenticated; (MIYAZAKI et al, 1990)) was maintained in high-glucose (4.5 g/L) Dulbecco's modified Eagle's medium (DMEM) supplemented with 15% Fetal bovine serum (FBS), 2 mmol/l GlutaMAX, 100 U/ml penicillin, 100 μg/ml streptomycin, 50 μmol/l 2-mercaptoethanol at 37 °C in a humidified atmosphere containing 5% $CO_2$. All cell culture-related reagents were from Life Technologies. Small interfering RNA (siRNA) knockdown experiments were performed by double transfection using 1 million MIN6 cells. First, cells were mixed with 50 nM siSSTR3 (Dharmacon), 4 μl Lipofectamine 2000 (Life Technologies) and 200 μl OptiMEM-l medium (Life Technologies) and seeded onto non-adherent plastic Petri dish (Sarstedt) to form pseudoislets. After 3 h, a 2nd transfection was performed using 50 nM siSSTR3, 5 μl Lipofecta-mine RNAiMAX (Life Technologies) and 250 μl OptiMEM-l medium supplemented with 5% FBS that was added to the pseudoislets, which were thereafter incubated overnight. The transfection was stopped by adding 2 ml complete culture medium. For plasmid transfections, 0.1–0.25 ug DNA was diluted in 25 μl of OptiMEM and mixed with 1 μl of Lipofectamine 2000 diluted in 25 μl OptiMEM. Cells and lipid-plasmid mix was plated onto the center of a 25-mm glass coverslip. After 4 h, the transfection was stopped with the addition of complete culture medium. In some cases, cells were incubated with 100 nM smoothened agonist (SAG), 100 nM somatostatin (SST) or both before immunofluorescence experiments next day of transfection.

The siRNAs were used: ON-TARGETplus Mouse SSTR3 (20607) siRNA (L-043208-03-0005), Target Sequences: AGAUCAGGGCCCAA GUUAA, GCUGUGUGGUCUCGGCCAA, GUAAAGGUGCGGUCG ACCA, GGGCUGUUGUCAGGGAGUA and ON-TARGETplus non-targeting siRNA (D-001810-10-05), Target Sequences: UGGUUUACAU GUCGACUAA, UGGUUUACAUGUUGUGUGA, UGGUUUACAUG UUUUCUGA, UGGUUUACAUGUUUUCCUA. The shRNAs were used ordered from VectorBuilder. pLV(shRNA)-mCherry: T2A:Puro-U6 > mSSTR3[GCACACTGAGCCATCTGTAAG] and control shRNA was packed into pLV(shRNA)-EGFP:T2A:Puro-U6>Scramble_shRNA [CCTAAGGTTAAGTCCCCTCG]. The concentration of both lenti-viruses exceeds $10^8$ TU/ml.

## MIN6 pseudoislet formation and culture

In all, 3–6 million/ml MIN6 cells were seeded into non-adherent 60 mm Petri dishes with 5 ml of DMEM culture medium and incubated at 37 °C in a humidified atmosphere containing 5% $CO_2$ for 3–7 days to aggregate and form pseudoislets.

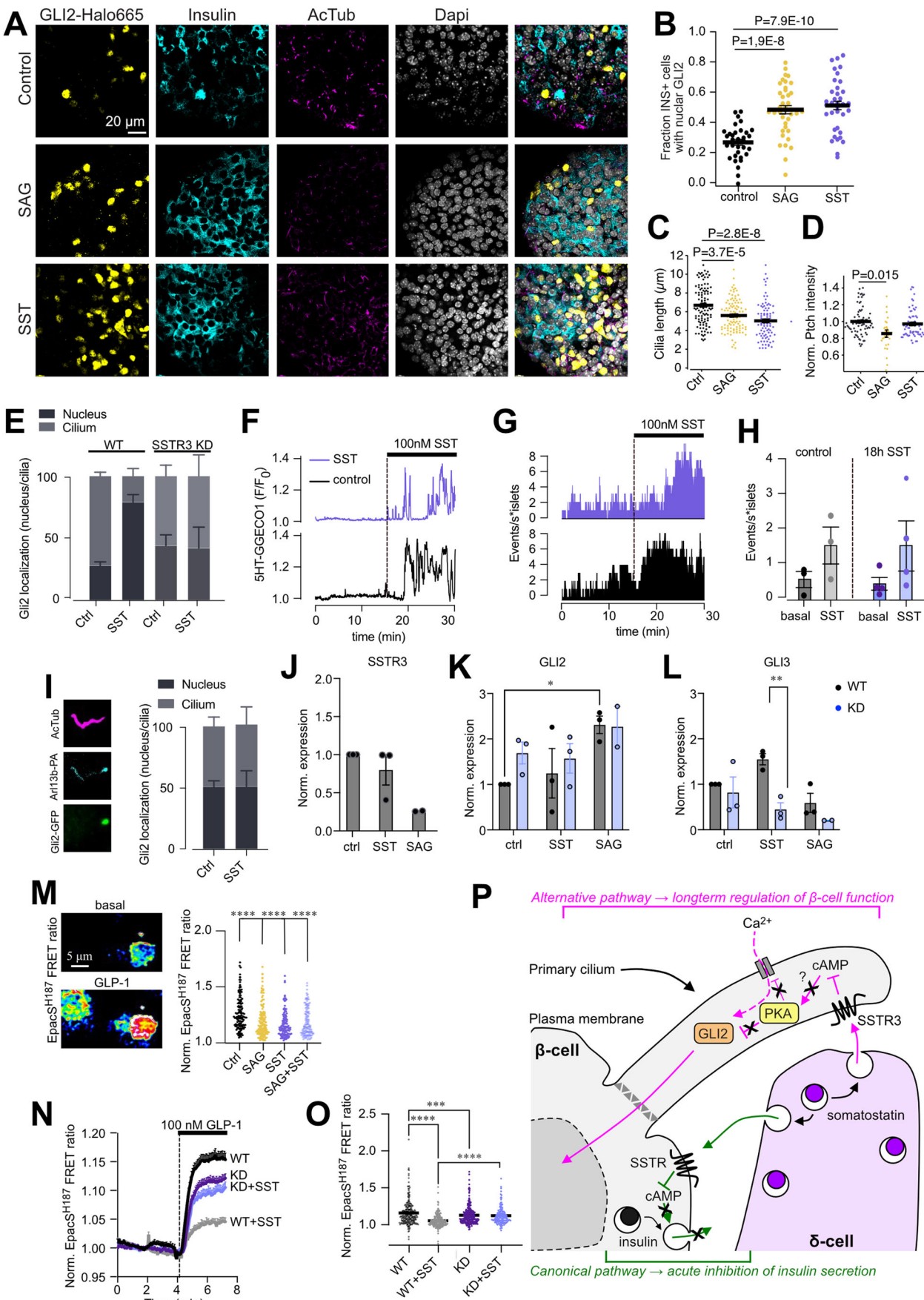

**Figure 8. Somatostatin induces GLI2 nuclear entry.**

(A) Confocal microscopy images of mouse islets expressing Gli2-Halo[JFX650] (yellow) and immunostained against acetylated tubulin (magenta) and insulin (cyan). Eighteen hours of treatment with 100 nM SAG or 100 nM SST induced translocation of Gli2 to the nucleus. (B) Fraction of β-cell nuclei positive for Gli2- Halo[JFX650] in control mouse islets and in islets treated for 18 h with 100 nM SAG or 100 nM somatostatin (means ± SEM; $n_{control} = 34$, $n_{SAG} = 40$, $n_{SST} = 39$ islets from two experiments; Student's unpaired two-tailed $t$ test). (C) Quantifications of cilia length (acetylated tubulin staining) from control (black), 18 h SST-treated (yellow) and 18 h SAG-treated (blue) mouse islet cells (means ± SEM; $n_{ctrl} = 111$, $n_{SST} = 87$, and $n_{SAG} = 105$ cilia from two different preparations; Student's unpaired two-tailed $t$ test). (D) Quantifications of ciliary Ptch1 intensity from control (black), SST-treated (100 nM, 15 min; yellow) and SAG-treated (100 nM, 15 min; blue) mouse islet cells (means ± SEM; $n = 79$, 27 and 77 cilia; Student's unpaired two-tailed $t$ test). (E) Means ± SEM for the nucleus/cilia Gli2 ratio change in shSSTR3 transfected MIN6 cells following 18 h exposure to 100 nM somatostatin ($P_{ctrl} = 0.0418$, $P_{shSSTR3} > 0.999$, assessed with Sidak´s multiple comparison test). (F) TIRF microscopy recordings of cilia $Ca^{2+}$ from mouse islets expressing 5HT$_6$-GGECO1 and stimulated with 100 nM somatostatin. The islets were either cultured under control condition (black) or in the presence of somatostatin (purple) for 18 h followed by a 60 min recovery period. (G) Histogram showing ciliary $Ca^{2+}$ changes following 18 h incubation with (blue) or without (black) somatostatin followed by 30 min in the absence of somatostatin and acute re-application of 100 nM somatostatin ($n_{ctrl} = 16$ cilia, $n_{SST} = 27$ cilia; ciliary $Ca^{2+}$ changes response to SST is quantified; $P_{Ctrl} = 0.0016$, $P_{SST} = 0.00011$; Wilcoxon test). (H) Quantification of the effect of the effect of long-term somatostatin treatment on somatostatin-induced cilia $Ca^{2+}$ changes (means ± SEM; $n_{ctrl} = 3$ islets, $n_{SST} = 4$ islets). (I) Confocal image of a MIN6 cell expressing Gli2-GFP (green) and Arl13b-Parvalbumin (cyan) and immunostained against acetylated tubulin (magenta). Means ± SEM for the nucleus/cilia Gli2 ratio change in MIN6 cells expressing Arl13b-Parvalbumin and exposed to somatostatin for 18 h ($n_{ctrl} = 132$, $n_{SST} = 132$ cells from 4 independent experiments, Sidak´s multiple comparison test). (J) Changes in SSTR3 expression in MIN6 cells exposed to 100 nM somatostatin or 100 nM SAG for 18 h (means ± SEM; $n = 2$–3 experiments). (K) Changes in GLI2 expression in control or SSTR3 knockdown MIN6 cells exposed to 100 nM somatostatin or 100 nM SAG for 18 h (means ± SEM; $n = 3$–4 experiments; *$P = 0.0376$; Tukey´s multiple comparison test). (L) Changes in GLI3 expression in control or SSTR3 knockdown MIN6 cells exposed to 100 nM somatostatin or 100 nM SAG for 18 h (means ± SEM; $n = 3$–4 experiments; **$P = 0.0047$; Tukey´s multiple comparison test). (M) Pseudo-colored TIRF microscopy images showing the increase cytosolic cAMP in response to 100 nM GLP-1 reported with EpacS[H187]. Means ± SEM for the FRET ratio changes from MIN6 pseudo-islets treated for 18 h with SAG (yellow), SST (purple) and SAG + SST (blue) are shown to the right (control-SAG $P = 0.000008531$, control-SST $P = 0.000000004$, control-SAG + SST $P = 0.000002794$; assessed by Dunn´s multiple comparison test; $n_{ctrl} = 143$, $n_{SAG} = 161$, $n_{SST} = 132$ and $n_{SAG+SST} = 131$ cells from three independent experiments). (N, O) Traces (means ± SEM) and scatter plot from EpacS[H187]-expressing control or SSTR3 knockdown MIN6 pseudoislets exposed or not to somatostatin for 18 h, followed by acute stimulation with 100 nM GLP-1 (WT-WT_SST $P < 0.000000000000001$, WT_SST-KD_SST $P = 0.0000002$, WT-KD $P = 0.00070$; Sidak´s multiple comparison test; $n_{WT} = 240$, $n_{WT-SST} = 200$, $n_{shSSTR3} = 259$, $n_{shSSTR3-SST} = 148$ from three different preparations). (P) A model of somatostatin signaling in the β-cell primary cilium. Source data are available online for this figure.

## Mouse islet isolation and culture

Female adult C57BL/6J mice (> 8 months; Scandbur, Denmark) housed in 12 h light/dark cycles with ad libitum access to food were euthanized by $CO_2$ and culled through decollation. The pancreas was collected, cut into smaller pieces and placed on ice before digestion with 1.7 mg/ml collagenase P on a shaker at 37 °C. The digestion was stopped by adding 2.5 μl/ml BSA and islets were picked under a stereomicroscope. Isolated islets were cultured in RPMI 1640 medium (Gibco) with 5.5 mM glucose supplemented with 100 U/ml penicillin, 100 μg/ml streptomycin and 10% FBS at 37 °C in a humidified atmosphere containing 5% $CO_2$. All experiments were approved by the Uppsala Animal Ethics Committee and conducted in compliance with national ethical regulations (Dnr 5.8.18-18546/2021). Experiments on transgenic mice expressing tdtomato under the somatostatin promoter were conducted at Peking University and approved by the Ethics Committee of Peking University (protocol code # AAIS-TangC-1). Mice were housed with a 12-h on/12-h off light cycle, 20–24 °C ambient temperature and 40–70% humidity. Mice with δ-specific expression of tdtomato were generated by crossbreeding SST-Cre mice (Jackson Laboratory) and tdtomato[fl/fl] mice (Biocytogen Pharmaceuticals Co., Ltd.). Islets were isolated from 4 moth old SST-tdtomato mice, followed by overnight culture in 8 mM glucose medium before imaging.

## Human islet culture

The Nordic Network for Clinical Islet Transplantation at the Academic hospital in Uppsala provided the isolated human islets from normoglycemic cadaveric organ donors (Dnr. 2006/348). The human islets were picked under stereomicroscope by hand and cultured in CMRL Medium (Gibco) with 5.5 mM glucose supplemented with 10% FBS, 100 U/ml penicillin and 100 μg/ml streptomycin at 37 °C and containing 5% $CO_2$. All experiments were approved by the Uppsala Human Ethics Committee. Donor characteristics are presented in Table 1. Palmitic acid was dissolved in 50% ethanol to make a 100 mmol/L stock solution. This was diluted in CMRL Medium with 0.5% fatty acid-free bovine serum albumin and incubated at 37 °C for 60 min to get a final concentration of 0.5 mmol/L.

## Viral transduction of islets and pseudoislets

Islets were infected with 2.5 μl high titration virus (> 10$^{12}$–10$^{13}$ vp/ml; Vector Biolabs) in 200 μl of culture medium that were transferred into Petri dish and incubated for 3 h at 37 °C in a humidified atmosphere containing 5% $CO_2$. After incubation, 3 ml of culture medium was added, and the islets were incubated for 24–48 h to allow for expression of the fusion proteins. For shRNA lentivirus-mediated knockdown experiments, islets were infected with 2.5 μl high titer virus (> 10$^6$ vp/ml; VectorBuilder) in 100 μl of culture medium and incubated for 5 h at 37 °C in a humidified atmosphere containing 5% $CO_2$. After incubation, 3 ml of culture medium was added, and the islets were incubated for 48–72 h to allow for knockdown of SSTR3.

## Immunofluorescence

Culture medium was removed, and cells or islets were washed with PBS followed by fixation with Paraformaldehyde, 4% in PBS (Thermo Fisher Scientific) for 20 min and permeabilized with 0.2% Triton X-100 in PBS for 5 min. The samples were blocked with 2% BSA in PBS for 1 h at room temperature and before addition of primary antibodies in blocking solution. After 2 h incubation at RT, samples were washed with PBS and incubated with secondary antibodies for 1 h. After secondary antibody incubation, cells were washed with PBS, and mounted using ProLong Gold Antifade Mountant (Thermo Fisher Scientific) on a glass coverslip. A small amount of 1% Agar in PBS was applied between glass slide and coverslip to embed the islets.

## Confocal microscopy

Pictures of immunolabeled samples were acquired using a spinning-disk confocal microscope unit (Yokogawa CSU-10) mounted on a Nikon Eclipse Ti2 body. The microscope was fitted with a CFI Apochromat TIRF 100×, 1.49-NA oil immersion objective from Nikon, and 491 and 561-nm diode-pumped solid-state (DPSS) lasers and a 640-nm diode laser (all from Cobolt/ Hübner Photonics) were used for excitation. The lasers were combined with dichroic mirrors, homogenized using a spatial filter, and aligned into a fiberoptic cable. Electronic shutters (Smart-Shutter, Sutter Instruments) selected the excitation light source, and emission light was separated using filters in a filter wheel controlled by a Lambda 10-3 unit (Sutter Instruments). GFP/ Alexa488 (530/50, Semrock), mCherry/Alexa561 (600/52; Semrock) and Alexa647 (650LP; Semrock) filters were used for emission light separation. Images were captured using a back-illuminated electron-multiplying charge-coupled device (EMCCD) camera (DU-888; Andor Technology) using MetaFluor software (Molecular Devices). For live cell imaging, 25-mm glass coverslips with cells or islets were used as exchangeable bottoms in an open Sykes-Moore perfusion chamber. The experiments were conducted at a temperature of 37 °C using an experimental buffer consisting of 125 mM NaCl, 4.9 mM KCl, 1.2 mM $MgCl_2$, 1.3 mM $CaCl_2$, 25 mM Hepes, 3 mM D-glucose, and 0.1% BSA at a pH of 7.40. Temperature was controlled by a custom-made microscope stage heater and objective heater. Experiments using islet isolated from transgenic mice expressing tdtomato in δ-cells were performed on a Zeiss 980 laser scanning microscope equipped with a 40×/1.25 NA objective with excitation at 488-nm and 561-nm. Islets were stimulated with 10 mM glucose for 30 min, followed by a brief pulse of 1 μM SST, a concentration that saturates SST1.0. Experiments on mouse islets expressing Halo-Gli2 were conducted using an inverted Leica Stellaris 5 microscope with 63×/1.4 oil immersion objective. The microscope was equipped with a 405-nm laser and a white light laser spanning a wavelength range of 485–790 nm and detection was done using three HyD detectors.

## TIRF microscopy

Unless stated otherwise, all experiments were conducted at a temperature of 37 °C using an experimental buffer consisting of 125 mM NaCl, 4.9 mM KCl, 1.2 mM $MgCl_2$, 1.3 mM $CaCl_2$, 25 mM HEPES, 3 mM D-Glucose, and 0.1% BSA at a pH of 7.40. Temperature was maintained by a custom-made stage heater and an objective heater. siRNA knockdown experiments were carried out using a Nikon Ti-E microscope equipped with an iLAS2 TIRF illuminator (Gataca Systems). The imaging was performed using a 100×, 1.49-NA Apo-TIRF objective (Nikon). For FRET-based detection of cAMP, EpacS[H188]-based sensors were imaged using 445-nm excitation light (Coherent DPSS; 80 mW). The emission was detected at 483/32 and 542/27 nm using interference filters from Semrock. A back-illuminated EMCCD camera (DU-897; Andor Technology) controlled by MetaMorph (Molecular Devices) was used to detect fluorescence. The remaining TIRF microscopy imaging was conducted on a custom-built setup based around an inverted Eclipse Ti-E microscope (Nikon) equipped with a TIRF illuminator and a 60×/1.45 NA plan-Apo objective (Nikon). The

system utilized diode-pumped solid-state lasers (Cobolt, Hübner photonics, Solna, Sweden) to provide excitation light for mTurquoise2 (445 nm), mCitrine (515 nm) and GFP/GGECO1 (491 nm). Emission light was separated by interference filters (483/32 for mTq2, 542/27 for mCitrine and 527/27 for GFP; all from Semrock, Rochester, NY). To record the translocation of $C_\alpha$ of PKA, islets were excited using a solid-state 561 nm DPSS laser (Cobolt) and emission light was collected through 587-LP (Semrock). A motorized filter wheel (Lambda 10-3, Sutter Instruments) equipped with interference filters (Semrock) was used to select laser lines, and a shutter (Sutter Instruments, Novato, CA) was used to block the beam between image captures. The fluorescence was captured by a CCD camera (Orca-ER, Hamamatsu) that was controlled by MetaFluor software (Molecular Devices).

## STED microscopy

Super-resolution microscopy was carried out using a Stedycon STED instrument (Abberior Instruments) mounted on an Olympus BX63F motorized microscope. The objective lens had a numerical aperture of 1.56 and a magnification of 100× (Olympus). The STED depletion laser was set to a wavelength of 775 nm, and the emitted light was filtered through 575–625 nm and 650–700 nm filters, which corresponded to Star 580 and Star Red fluorophores, respectively.

## Image analysis

The Fiji version of the ImageJ (Schindelin et al, 2012) software was used to analyze all images. Changes in fluorescence intensity over time was determined from segmented region drawn to cover the cilium or part of cytoplasm, and values were background corrected and normalized to the pre-stimulatory level. For FRET-based cAMP measurement, ratiometric data was obtained in a similar way by determining the change in CFP/YFP emission in similar regions. To analyze live cell $Ca^{2+}$ and cAMP imaging data (TIRF), ROIs containing the cilium were drawn, and mean values of pixel intensities were extracted for each timeframe (sampling frequency was 0.2–2 Hz). For analysis of SST2.0, average intensity projection of the original time-lapse recording was used to identify areas of interest. Here, the bases and tips of cilia were marked as C1 and C2 respectively. Areas adjacent to the cilia were selected as R1, R2, R3, and R4. R5 and R6 were the areas relatively far from cilia. The average fluorescent intensities were extracted from the ROIs elucidated above along the time series. To eliminate the effect of uneven expression of SST2.0, all the average intensity values were normalized by their saturated fluorescent intensities, respectively. The amplitudes of each peak, defined as the increased intensity between the peak point and the initial point, were manually measured by Igor Pro 9 software (WaveMetrics). For quantifications of immunostained samples, we flattened a volumetric image set (10 μm axial) to obtain a 2D image by maximum intensity projection. Cilia were identified using cilia markers (acetylated tubulin, glutamylated tubulin, Arl13b) and regions covering the entire, flattened, cilia were used to determine the enrichment of ciliary receptors relative to the local, non-ciliary, background. Similarly, distances between cilia and δ-cells were also determined using flattened images.

## Segmentation of FIB-SEM datasets

The open-access FIB-SEM dataset of mouse pancreas P7 (https://janelia.figshare.com/articles/dataset/Near-isotropic_reconstructed_volume_electron_microscopy_FIB-SEM_of_P7_mouse_pancreas_jrc_mus-pancreas-4_/23411843) was screened for beta cell cilia in proximity to δ-cells. Smaller volumes were cut out in FIJI and manually segmented with Microscopy Image Browser (Belevich et al, 2016). The segmentation masks were rendered in 3D with ORS Dragonfly.

## Statistical analysis

No power analysis was performed, and this study does not have a predetermined sample size. Sample size was instead determined based on previous experience from similar studies and all experiments were performed a minimum of three times. Samples were not randomized, and blinding was not applied. No specific inclusion or exclusion criteria were used. GraphPad Prism version 9.5.1 for Mac software was used to analyze the data in our study. The data were first tested for normal distribution using the Kolmogorov–Smirnov (KS) test. Nonparametric Wilcoxon signed-rank tests was conducted on non-normally distributed data to evaluate the changes in cAMP and Ca2+ concentrations in control and experimental groups, and to see receptor distribution. Multiple comparisons were performed using the Friedman test or Kruskal–Wallis test with Dunn–Šidák's multiple comparisons post hoc test. More details on the statistical analyses can be found in the figure legends.

# Data availability

This study includes no data deposited in external repositories.

The source data of this paper are collected in the following database record: biostudies:S-SCDT-10_1038-S44318-025-00383-7.

# Peer review information

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

## Acknowledgements

The authors express our sincere gratitude to Anders Tengholm, Erik Gylfe, and all the members of Olof Idevall-Hagren lab for their valuable input during this study, to Mrs. Helene Dansk for help with islet isolation, and to Mrs. Finja Hildebrand for assistance with immunostainings during the early stages of the project. The authors also acknowledge the staff at the BioVis facility at Uppsala University for their outstanding assistance with STED microscopy imaging and for use of the Leica Stellaris 5 LSCM. This work was supported by grants from the Swedish Research Council, the Novo-Nordisk Foundation, the Swedish Diabetes Foundation, the Family Ernfors Foundation, The European Foundation for the Study of Diabetes/Lilly, Diabetes Wellness Sweden, and Exodiab.

## Author contributions

**Ceren Incedal Nilsson**: Conceptualization; Formal analysis; Supervision; Investigation; Visualization; Methodology; Writing—original draft; Writing—review and editing. **Özge Dumral**: Formal analysis; Investigation; Methodology; Writing—review and editing. **Gonzalo Sanchez**: Conceptualization; Formal analysis; Supervision; Investigation; Methodology; Writing—original draft. **Beichen Xie**: Formal analysis; Investigation; Methodology. **Andreas Müller**: Formal analysis; Investigation; Visualization; Methodology; Writing—review and editing. **Michele Solimena**: Supervision; Funding acquisition; Writing—review and editing. **Huixia Ren**: Funding acquisition; Writing—review and editing. **Olof Idevall-Hagren**: Conceptualization; Formal analysis; Supervision; Funding acquisition; Investigation; Visualization; Writing—original draft; Project administration; Writing—review and editing.

Source data underlying figure panels in this paper may have individual authorship assigned. Where available, figure panel/source data authorship is listed in the following database record: biostudies:S-SCDT-10_1038-S44318-025-00383-7.

## Funding

## Disclosure and competing interests statement

The authors declare no competing interests.

# Expanded View Figures

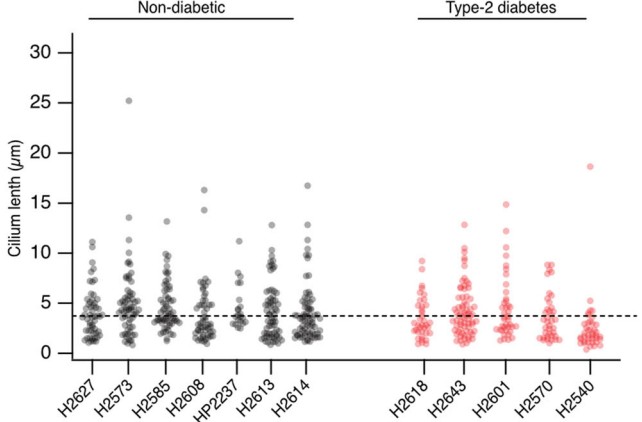

**Figure EV1.　Cilia length in human islets from non-diabetic and type-2 diabetic donors.**

(A) Quantification of cilium length in insulin-positive β-cells in islets from 7 non-diabetic (black; $n = 94, 66, 57, 56, 45, 92$ and 54 cilia) and 5 type-2 diabetic (red; $n = 57, 42, 53, 33$ and 71 cilia) human organ donors. Dashed line shows average for all non-diabetic donors. (B) Quantification of cilium length in insulin-negative cells in islets from 7 non-diabetic (black; $n = 54, 65, 63, 54, 22, 66$ and 74 cilia) and 5 type-2 diabetic (red; $n = 37, 70, 40, 38$ and 40 cilia) human organ donors. Dashed line shows average for all non-diabetic donors. Source data are available online for this figure.

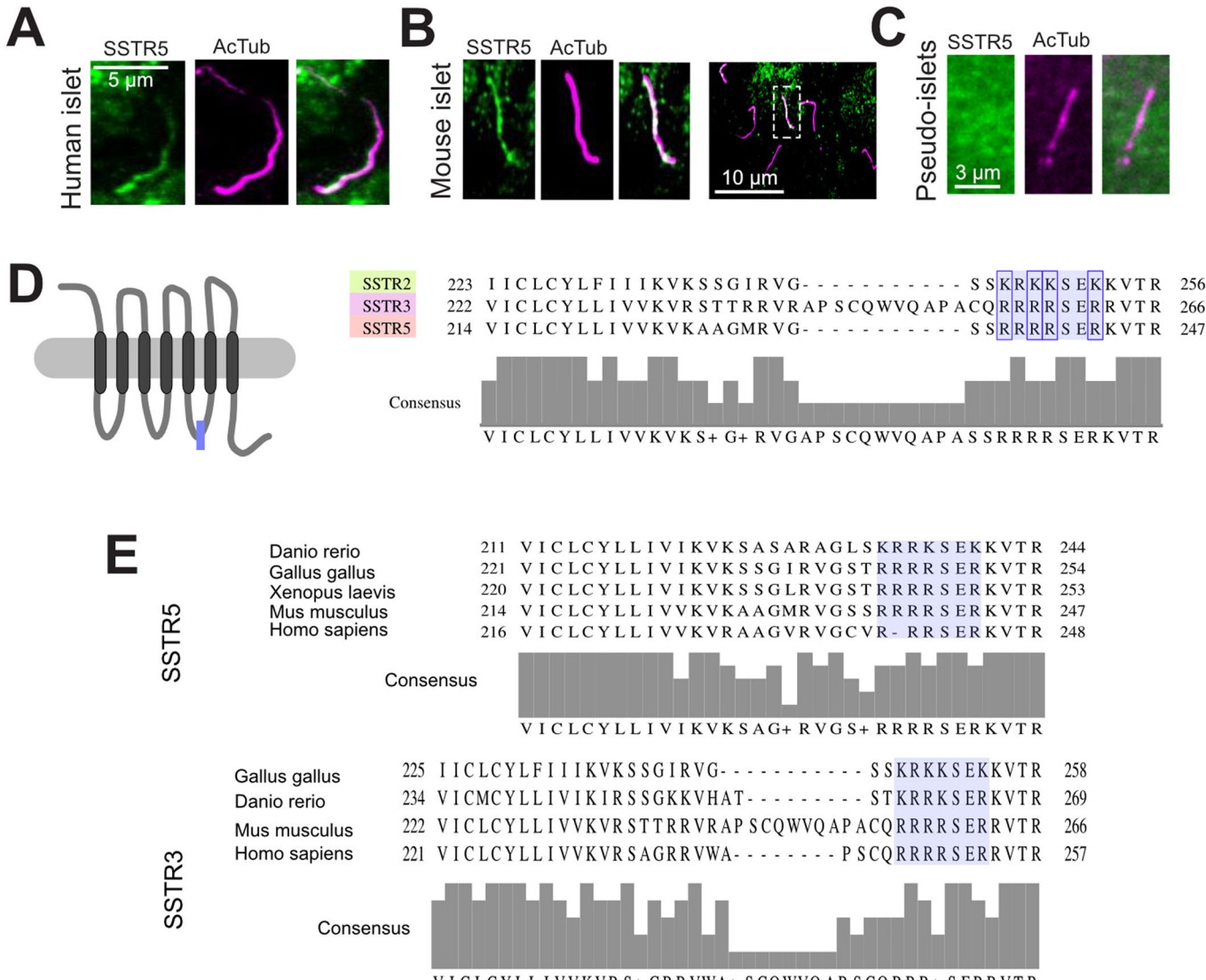

**Figure EV2. SSTR5 localized to primary cilia.**

(A) Confocal microscopy image of a primary cilium (acetylated tubulin; magenta) in a human islet that expresses SSTR5 (green). (B) Confocal microscopy image of primary cilia (acetylated tubulin; magenta) in a mouse islet that expresses SSTR5 (green). (C) Confocal microscopy image of a primary cilium (acetylated tubulin; magenta) in a MIN6 pseudoislet that lack expression of SSTR5 (green). (D) An illustration shows the membrane localization of SSTRs, with IC3 loop indicated by the purple box. Sequences of mSSTR2, mSSTR3 and mSSTR5. Conserved motif RxRxxR is highlighted. (E) Sequences of SSTR5 and SSTR3 showing the evolutionary conservation of a stretch of amino acids in purple. Source data are available online for this figure.

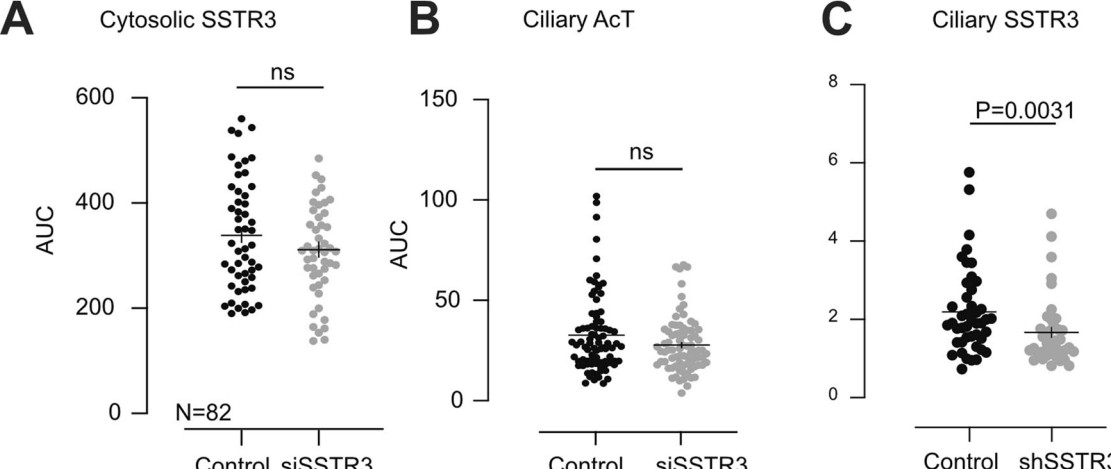

**Figure EV3. SSTR3 is a ciliary somatostatin receptor.**

(A) Quantification of cytosolic area selected in MIN6 pseudoislets. Fluorescence intensity is from control (black) and SSTR3 KD (gray) cells immunostained for SSTR3. (means ± SEM; $n_{ctrl} = 50$ and $n_{KD} = 48$, 3 different preparations, no statistical difference by Mann–Whitney $U$ test, unpaired.). (B) Quantifications of line profiles drawn along cilia of MIN6 pseudoislets for acetylated tubulin. Acetylated tubulin signal is unaffected on the left. (means ± SEM; $n_{ctrl} = 83$ and $n_{KD} = 86$, three different preparations, no statistical difference by Mann–hitney $U$ test, unpaired.). (C) Quantifications of line profiles drawn along cilia of MIN6 pseudoislets positive for SSTR3. Ciliary SSTR3 signal is significantly reduced in shSSTR3 expressing cells (means ± SEM; $n_{ctrl} = 44$ and $n_{KD} = 40$ cilia, 1 preparation; Mann–Whitney $U$ test, unpaired). Source data are available online for this figure.

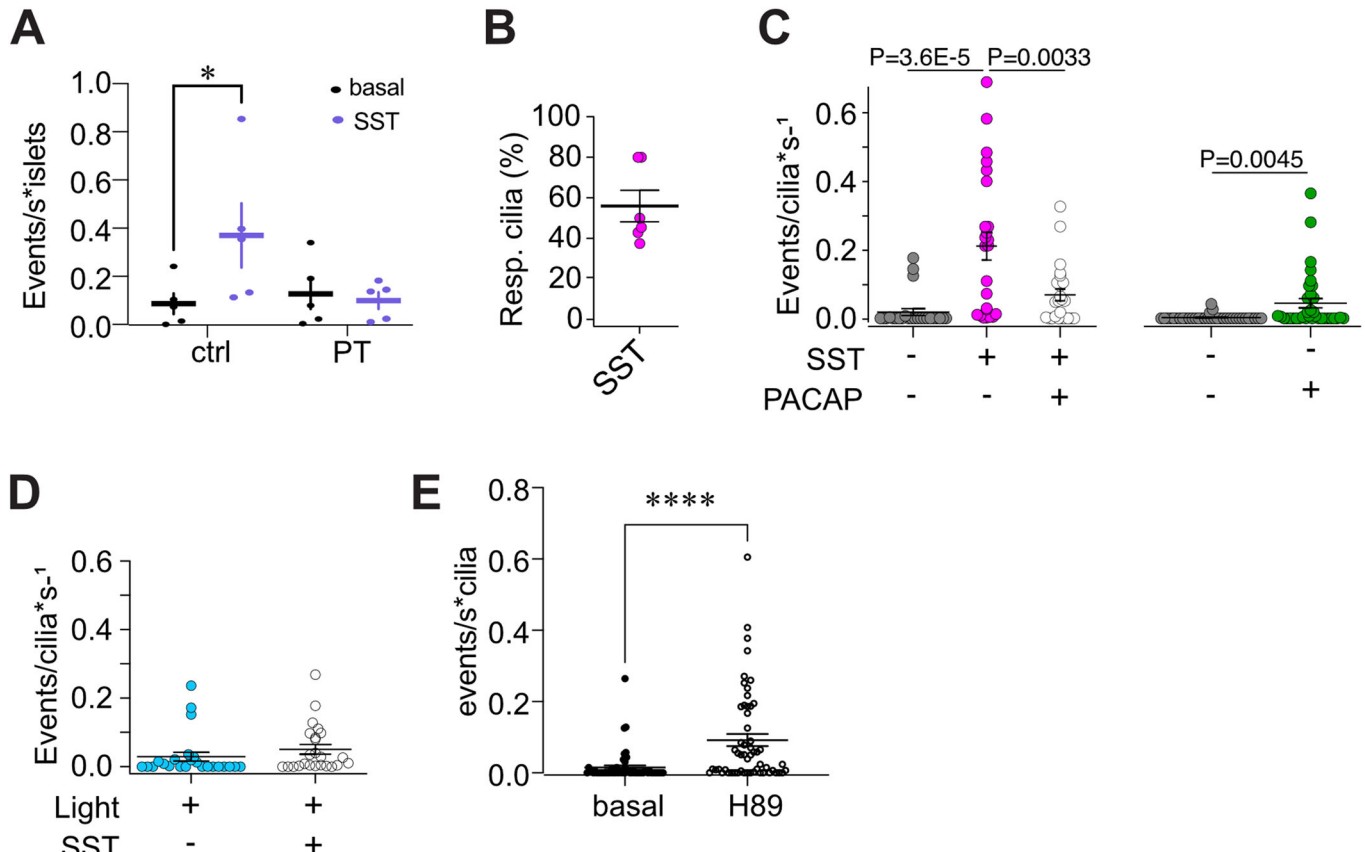

**Figure EV4.   Elevation of cAMP counteracts somatostatin-induced ciliary Ca²⁺ signaling.**

(A) MIN6 pseudoislets were cultured under control condition (black) or in the presence of pertussis toxin (PT; purple) for 18 h. Quantifications of the Ca$^{2+}$ responses to 100 nM somatostatin are shown to the right (means ± SEM; $n_{ctrl} = 5$ islets, $n_{PT} = 5$ islets; from three different preparations, SST response in control $P = 0.0425$, SST response in PT not significant, assessed by Sidak´s multiple comparison). (B) Fraction of MIN6 cell cilia per pseudoislet that exhibit somatostatin-induced ciliary Ca$^{2+}$ signaling (means ± SEM; $n = 6$ islets). (C) Ciliary Ca$^{2+}$ responses under basal conditions (gray) or in the presence of 100 nM somatostatin (magenta), a combination of somatostatin and 100 nM PACAP (white) or 100 nM PACAP alone (green) in MIN6 pseudoislets (means ± SEM; left panel: $n = 25$ cilia from 6 islets; right panel: $n = 36$ cilia from 4 islets; Student's paired $t$ test). (D) Ciliary Ca$^{2+}$ responses to 100 nM somatostatin in MIN6 pseudoislets co-expressing cytosolic bPac. bPac activity was continuously stimulated during the experiment by 491-nm illumination (Light). (mean s± SEM; $n = 24$ cilia from 3 islets; Student's paired $t$ test). (E) Event count of ciliary Ca$^{2+}$ changes from mouse islets in response to 10 μM H89 (means ± SEM; 7 islets; 56 cilia; 3 different preparations. ****$P < 0.0001$, Wilcoxon-matched pair $t$ test). Source data are available online for this figure.

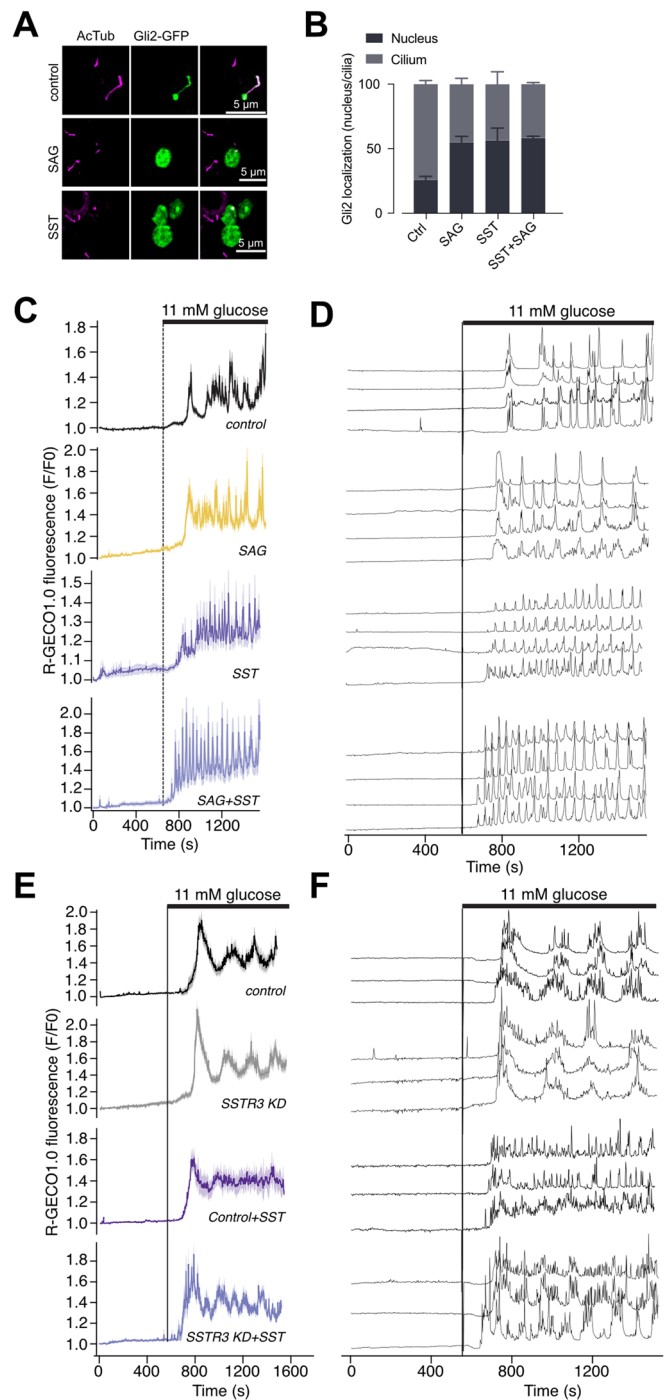

**Figure EV5.   Long-term somatostatin stimulation alters beta cell Ca²⁺ response to glucose.**

(A) Confocal microscopy images of MIN6 cells expressing Gli2-GFP (green) and immunostained against acetylated tubulin (magenta). 18 h treatment with 100 nM SAG and 100 nM SST induced translocation of Gli2 to the nucleus. (B) Means ± SEM for the nucleus/cilia Gli2 ratio change in MIN6 cells ($P_{SAG} = 0.0345$, $P_{SST} = 0.0266$, $P_{SAG+SST} = 0.0186$ all compared to control and assessed by Sidak´s multiple comparison test). (C, D) Islet averages (C) and example recordings (D) of glucose-induced R-GECO1 fluorescence changes in MIN6 pseudoislets treated for 18 h with DMSO (control), 100 nM SAG, 100 nM somatostatin or 100 nM SAG in combination with 100 nM somatostatin ($n = 3$ replicates). (E, F) Islet averages (E) and example recordings (F) of glucose-induced R-GECO1 fluorescence changes in control and SSTR3 knockdown MIN6 pseudoislets treated for 18 h with DMSO (control) or 100 nM somatostatin ($n = 3$ replicates). Source data are available online for this figure.

