## [Peer Review File · The EMBO Journal]

Somatostatin triggers local cAMP and Ca²⁺ signaling in primary cilia to modulate pancreatic β -cell function

Tugce Incedal, Özge Dumral, Gonzalo Sanchez, Beichen Xie, Andreas Mueller, Michele Solimena, Huixia Ren, and Olof Idevall-Hagren

Corresponding author: Olof Idevall-Hagren (olof.idevall@mcb.uu.se)

Review Timeline:

Submission Date:	24th Jun 24
Editorial Decision:	31st Jul 24
Revision Received:	25th Nov 24
Editorial Decision:	20th Dec 24
Revision Received:	15th Jan 25
Accepted:	29th Jan 25

Editor: Ieva Gailite

Transaction Report:

Dear Dr. Idevall-Hagren,

Thank you for submitting your manuscript for consideration by the EMBO Journal. We have now received comments from three reviewers, which are included below for your information.

As you can see, all reviewers are generally positive in their assessment and find the proposed role of ciliary somatostatin signalling in regulation of beta cell function of interest. However, they also raise a range of concerns that would need to be addressed before they can support publication. From my side, I find the raised points generally reasonable. I would therefore invite you to address these remaining comments in a revised manuscript. I think that it would be useful to discuss the revision in more detail via email or phone/videoconferencing - please let me know which option you prefer.

We generally allow three months as standard revision time, which can be extended to six months in the case of major revisions. Should you foresee a problem in meeting this deadline, please let us know in advance to discuss an extension. As a matter of policy, competing manuscripts published during this period will not negatively impact on our assessment of the conceptual advance presented by your study. However, please contact me as soon as possible upon publication of any related work to discuss the appropriate course of action.

When preparing your letter of response to the referees' comments, please bear in mind that this will form part of the Review Process File and will therefore be available online to the community. For more details on our Transparent Editorial Process, please visit our website: <https://www.embopress.org/page/journal/14602075/authorguide#transparentprocess>. Please also see the attached instructions for further guidelines on preparation of the revised manuscript.

Please feel free to contact me if have any further questions regarding the revision. Thank you for the opportunity to consider your work for publication, and I look forward to discussing your revision with you.

With best regards,

Ieva

We realize that it is difficult to revise to a specific deadline. In the interest of protecting the conceptual advance provided by the work, we recommend a revision within 3 months (29th Oct 2024). Please discuss the revision progress ahead of this time with the editor if you require more time to complete the revisions.

Referee #1:

In the manuscript "Somatostatin triggers local cAMP and Ca²⁺ signaling in primary cilia to modulate beta-cell function" Incedal Nilsson et al suggest that islet beta-cell primary cilia constitute an important target of somatostatin action and thereby enabling regulation of islet cell function beyond inhibition of insulin release.

Overall this is an interesting manuscript addressing the potential role of islet cell primary cilia in the context of islet cell signal-transduction and thereby function. The manuscript demonstrates that somatostatin secreting delta-cells are positioned close to cilia from insulin-secreting beta-cells within the pancreatic islet. Released somatostatin directly activates receptors on the beta-cell cilia resulting in a lowering of cilia cAMP levels and thereby nuclear translocation of the cilia-dependent transcription factor GLI2. The manuscript also shows that primary cilia length is reduced in human beta-cells obtained from donors with T2DM and that this is associated with reduction in cilia delta-cell interactions. Nevertheless, there are a number of things that need clarification, a few of them listed below.

Due to the complex paracrine signaling network within the pancreatic islet it would be important also to understand how cilia on the alpha-cells are affected under these conditions. With regard to the MIN6 cells, is the expression of somatostatin receptors on the primary cilia affected by the glucose sensitivity of these cells, i.e. when the cells dedifferentiate and lose their glucose sensitivity what happens with the cilia from a functional and morphological standpoint? How can it be mechanistically explained that neither the number of somatostatin receptors on the primary cilia nor their sensitivity to the hormone are affected after 18 h of stimulation? What physiological implications are there for islet hormone release in general upon the shortening of islet cell primary cilia and thereby impaired connectivity with delta-cells in T2DM? What is the molecular mechanism underlying this shortening? Is there a metabolic explanation and if yes is this process reversible upon normalization of blood glucose concentration? Is cAMP also diffusing freely between the cytosol and the cilium under T2DM conditions? The relationship between somatostatin, cAMP and Ca²⁺ signaling in the cilia is interesting but is not clear. What is going on mechanistically and what does it mean in terms of signaling in the primary cilia? Why is exposure to SAG associated with a slight lengthening of cilia and somatostatin exposure associated with a small reduction in cilia length?

Referee #2:

This manuscript evaluates regulation of somatostatin (SST) signaling in pancreatic islet cells. The results demonstrate that SST, which is released from filopodia-like extensions of δ -cells, signals to SST receptor 3 (SSTR3), a G protein coupled receptor (GPCR) that localizes to primary cilia of β -cells. The authors demonstrate that SSTR3, which couples to G α i, leads to cAMP reduction in SST responding cells. Consistent with a previous report (doi.org/10.1080/19382014.2023.2252855), the study demonstrates that activation of primary cilium localized SSTR3-G α i signaling leads to changes in Ca²⁺ flux. This manuscript demonstrates that increased Ca²⁺ in the primary cilium occurs downstream of SSTR3. These signals are proposed to facilitate downstream effector responses, including changes in transcription regulation through noncanonical activation of the Hedgehog transcriptional effector GLI2. These results suggest crosstalk between SSTR and Hedgehog signaling pathways occurs at the primary cilium. The manuscript provides high resolution imaging of δ -cell contact with β -cell primary cilia, makes use of some novel tools to track ciliary cAMP, and provides an interesting model for GPCR signal crosstalk in the primary cilium. The work will likely be of broad interest, pending some additional experimentation.

Key Questions

It is noted in Figures 2E, I, J, and 8C that primary cilia of β -cells shorten following SSTR3 activation, but implications of ciliary

SST-induced shortening are not discussed. Is this a result of activation of G α i coupled SSTR3 in the primary cilium? Other studies demonstrate that changes in ciliary cAMP can influence ciliary length and that these length changes can impact signal output (doi: 10.1038/ncb3029; doi.org/10.1083/jcb.202306002; doi.org/10.1016/j.cub.2009.11.072).

Figure 3: How does SSTR3-responsive shortening seen in Figure 2 relate to the ciliary shortening that is observed in β -cells from type II diabetes samples?

Figure 4: It is noted that the SST1.0 sensor that is used to report plasma membrane and primary cilium SSTR3 activation shows a higher response frequency in primary cilia compared to the plasma membrane. Please provide a more detailed explanation of what this means in the results section discussing Figure 4K, L. It is stated that these data show that "...somatostatin secretion preferentially occurs towards primary cilia". How? Could it be that the increased response frequency in cilia is due to a higher local concentration of receptor in the small primary cilium compared to more sparsely localized receptor on the plasma membrane?

Figure 6: If an antibody is available, please provide a western blot to demonstrate the level of SSTR3 protein reduction following knockdown (Figure 6D).

Figure 7 shows that Ca $^{2+}$ responses are dependent on SSTR3 because the response is reduced following SSTR3 knockdown. The authors previously showed that GABAB1-G α i signaling controls Ca $^{2+}$ signaling in β -cells (DOI: 10.1083/jcb.202108101). Is GABAB1 involved in this response? GABAB1 should be knocked down to determine whether there is synergy between SSTR3 and GABAB1 - or if the Ca $^{2+}$ response occurs directly through SSTR3-G α i signaling. If it does involve synergy with GABAB1 - why is this necessary? Why can't SSTR3-G α i signaling induce a Ca $^{2+}$ response?

Figure 8A, 8I: Please use DAPI to mark nuclei. Why is no GLI2-GFP evident at the tip of the primary cilium following SAG stimulation in 8A? GLI2 levels typically increase at the tip of the primary cilium in response to SHH pathway activation.

Figure 8C shows that SAG treatment increases primary cilium length, but this result is not discussed. What is going on here?

Figure 8J: Why does SAG reduce SSTR3 expression?

Figure 8A and K-L: The authors use GLI2-GFP nuclear accumulation as a surrogate for transcriptional activation. Please assess activation of SHH target genes like Ptch1 and Gli1 to determine whether nuclear GLI2 induces a transcriptional response. GLI2 and GLI3 are not transcriptional targets, and the expression changes shown in panels K and L are very modest, so are not good indicators of a SHH/GLI transcriptional response.

Minor Suggestion

The authors refer to GLI2 "processing" in the results and discussion. GLI2/3 "processing" typically refers to proteolytic processing of GLI2/3 that removes the transcription activation domain to generate GLI repressors. If you mean "activation", please write "activation".

The discussion states that "...Ca $^{2+}$ has been shown to be involved in the Hedgehog response..." and references the Belgacem & Borodinsky, 2011 PNAS paper (DOI: 10.1073/pnas.1018217108). The SMO-G α i response described in the 2011 paper describes a non-canonical GLI transcription independent SHH signal mediated by IP3 and Ca $^{2+}$. This should be clarified in the discussion because, as written, the authors' statement might be interpreted as suggesting that Ca $^{2+}$ has been demonstrated to influence canonical GLI activity. This has not been shown in the literature.

The authors state in the discussion and indicate in their model (Figure 8P) that SSTR3-mediated G α i activation leads to reduced PKA activity. It doesn't look like PKA activity was directly tested in this study. Please clarify.

Referee #3:

General Summary:

In this manuscript the authors create a compelling case for the unique role that ciliary SSTR signaling plays in maintaining proper beta cell function through paracrine interactions with the delta cell. By combining EM and fluorescent microscopy techniques, the authors functionally map out the presence and signaling contributions of ciliary SSTR3 using established sensors and techniques. The data presented are fundamentally similar to the approaches and findings of ciliary signaling importance in recent papers focused on neurons (Sheu 2022, Reiter 2021). Overall, I find several aspects of this manuscript very exciting, specifically, the differences between SSTR activation in the cilia (calcium increase) and cytosol, coupled with the relationship between PKA activity and this novel calcium pathway. This manuscript will serve as a launching point for future studies of this pathway and lays out the importance of understanding the relationship between the cilia and metabolic disease

as others have before (Hughes 2020). Finally, it is appreciated the length with which the authors went to place their system in the context of known islet biology, with the use of ghrelin to stimulate delta cells in figure 7 as an example. Overall, this manuscript was an interesting and fun read, but also one that provoked several questions:

There are however two areas where the manuscript would need some careful reconsideration. The first is that the purported contribution of SSTR5 in beta cell cilia and the second is the coupling of FFAR4 to cAMP in cilia. In both examples, the authors cite some of the prior work that aligns with this narrative, but do not cite and discuss the work of others that is less easily reconciled with this model. Ultimately, these parts of the manuscript distract from an otherwise exciting manuscript that represents a significant amount of innovative experimental work that is well presented. Finally, the connection with Gli2 towards the conclusion of the paper is established only using an overexpression model, while these data could be strengthened considerably with evidence of endogenous Gli2 playing a comparable role.

There are a series of other comments and suggestions that would be readily addressable that are provided to the authors for their consideration.

specific major concerns essential to be addressed to support the conclusions

The authors make their case for SSTR5 expression on the primary cilium of beta cells, showing immunohistochemistry that shows a ciliary pattern in mouse tissue. This staining is not shown in human tissue and is absent in Min6 cells (mouse derived). These findings contradict previous results from multiple labs using SSTR5 as a ciliary negative control (e.g. Berbari 2008). The authors cite a few papers in support of the expression of SSTR5 in islets, but fail to acknowledge strong evidence of no SSTR5 expression. Specifically, bulk transcriptome data from purified alpha, beta, and delta cells from several groups - cited elsewhere in the manuscript - demonstrates no detectable SSTR5 in mouse islet cells, with one of these papers even dedicating a figure to the expression of a select set of GPCRs including the SSTRs. Once the paper reaches the actual characterization of the model using ciliary sensors, it appears as though SSTR3 dictates downstream ciliary processes, indicating that it is the main/only SST receptor in mouse beta cell cilia. This again calls into question any role of SSTR5. Unless the antibody result is backed up by proper validation in SSTR5 null beta cells, these findings largely distract from your story. It appears that this line of inquiry flowed from the SST sensor, that is based on SSTR5 and in the hands of these investigators ends up in cilia when overexpressed. Others using the same sensor have not seen the same expression pattern. The remaining data on SSTR5 expression are based solely on antibody staining, with evidence of no functional contribution of SSTR5. My recommendation is to simply leave the SSTR5 observations out. I think your story is much more compelling without them.

This question again arises with the use of an SSTR5 based SST GRAB sensor which appears to localize preferentially to the cilia in this manuscript, but shows no similar cilia localized phenotype in islets within the 2023 paper it is cited from. Again, this is unsettled science and should be explained or addressed in the discussion at the least.

The other main comment relates to FFAR4. It is presented as a cilia GPCR in beta cells that couples to Gas and therefore - activates cilia cAMP. However, there is a substantial body of work by multiple groups that indicates that FFAR4 signals via Gαq by default (see eg. <https://doi.org/10.1016/j.celrep.2024.114509>). The manuscript also ignores a body of literature that describes that primary mouse beta cells do not express FFAR4 mRNA, but that this is expressed instead by delta cells (e.g. PMID: 33484949). Perhaps closer attention should be paid to the models used. The authors switch between mouse islets and min6 cells regularly. While this is common and often quite reasonable as beta cell lines offer experimental benefits over primary islet experiments, in this case min6 cells likely do not model the signaling in primary beta cells, with min6 cells express FFAR4 on their cilia, while primary beta cells do not. The experiments where Tug891 and SST are tested together (Figure 6) are then useful to interrogate the cilia-specific mechanisms of signaling, but likely do not reflect the actual signaling dynamics in primary beta cells where the stimulation of insulin secretion by FFAR4 agonists is mediated by the inhibition of delta cells: delta cell-specific deletion of FFAR4 abolishes the effects of FFAR4 agonism on insulin secretion. None of this is considered or discussed. The results observing translocation of GLI2-GFP in min6 cells are quite striking. While this experiment determines that GLI2 will translocate to the nucleus, it does not demonstrate that endogenous GLI2 does the same. How does the overexpression of GLI2 compare to endogenous protein and mRNA levels, which are low to not detected. Is there compelling data that link SST stimulation of primary beta cells to the activation of GLI2-dependent gene expression pathways?

minor concerns that should be addressed

The citation of SSTR2 as the main receptor in human alpha and beta cells does not reflect the known expression data in mouse islet endocrine cells, where there is evidence of no expression of SSTR2 in beta cells. Since the data in this manuscript are primarily obtained in cells of mouse origin, introducing the expression pattern in mouse would seem appropriate.

The suggestion that cilia serve as antennas needs to contend with the fact that nutrients and paracrines would by simple diffusion travel the length of the cilia within a very short amount of time. Instead, the view of cilia as isolated signaling compartments where signaling can take place independent from the cytosol (see PMID: 37072495, PMID: 38366037) is more in line with the data presented in this manuscript.

In that same statement, the authors speak of 'enrichment of receptors' in the cilia. This statement requires qualification, a select number of GPCRs that is listed by the authors is of course expressed by cilia. These are the receptors that are 'enriched in the cilia' and for which signals can be sensed selectively at that location. However, a beta cells expresses upwards of 150 GPCRs, and most are not expressed selectively by cilia. Therefore, for the class of GPCRs, speaking of 'enrichment' in the cilia is not true.

The cilia length measurements appear to be reduced to a single average per donor (Figure 3G, H). I assume that many cilia per donor were measured. The authors should explore how these intra-donor replicates can be better reflected in the data and statistical analysis.

The experiments in Figure 4 that measure cilia signaling to SST in delta cell adjacent cilia are elegant and nicely done!

The observation that SST stimulates calcium in primary cilia, but not the cytosol of the same cells is surprising. What is the source of this calcium?

The use of ghrelin to elicit the local release of SST in figure 7 is a compelling piece of experimental data that takes full advantage of previously characterized local crosstalk within the islet.

The authors claim in the discussion that it is not known if SST secretion is changed in T2D. PMID: 26076035 show that SST secretion in obob islets in response to glucose is impaired. PMID: 32446876 show comparable findings with regards to SST secretion in a HFD model.

The model in figure 8 is not helpful. It tries to reconcile observations from mouse and human primary islet cells and mouse cell lines. Specifically, SSTR3 is at the cilia, and not the PM. SSTR2 is not expressed by mouse beta cells (human beta cells are perhaps different, but almost all data in this paper originate in mouse). As discussed earlier, there is strong evidence of no expression of SSTR5.

The legend for figure 7 does not include a description of panel K and L.

Response to reviewers comments

We thank the reviewers for the positive comments and for suggestions on how to improve the manuscript. Below are point-by-point answers to all comments. All changes to the manuscript text are indicated by red color.

Referee #1:

In the manuscript "Somatostatin triggers local cAMP and Ca²⁺ signaling in primary cilia to modulate beta-cell function" Incedal Nilsson et al suggest that islet beta-cell primary cilia constitute an important target of somatostatin action and thereby enabling regulation of islet cell function beyond inhibition of insulin release.

Overall this is an interesting manuscript addressing the potential role of islet cell primary cilia in the context of islet cell signal-transduction and thereby function. The manuscript demonstrates that somatostatin secreting delta-cells are positioned close to cilia from insulin-secreting beta-cells within the pancreatic islet. Released somatostatin directly activates receptors on the beta-cell cilia resulting in a lowering of cilia cAMP levels and thereby nuclear translocation of the cilia-dependent transcription factor GLI2. The manuscript also shows that primary cilia length is reduced in human beta-cells obtained from donors with T2DM and that this is associated with reduction in cilia delta-cell interactions. Nevertheless, there are a number of things that need clarification, a few of them listed below.

1) Due to the complex paracrine signaling network within the pancreatic islet it would be important also to understand how cilia on the alpha-cells are affected under these conditions.

Reply: We agree with the reviewer that it will be important to understand the role of primary cilia in all cell types of the islet. Given that the expression of plasma membrane receptors differs between the cell types of the islet, it is likely that the same is true for cilia-localized receptors (as also shown in PMID: 34385262). In figure 3, we also show that cilia of all islet cell types (including alpha cells) are shortened in islets from type-2 diabetic donors. However, exploring the role of alpha cell cilia was never the aim of this study, which instead focused on further understanding how beta cell cilia sense the islet environment and how this influence beta cell function. To extend these studies to also include alpha cells is a large undertaking that we feel is beyond the scope of this study. We hope that the reviewer understands this.

2) With regard to the MIN6 cells, is the expression of somatostatin receptors on the primary cilia affected by the glucose sensitivity of these cells, i.e. when the cells dedifferentiate and lose their glucose sensitivity what happens with the cilia from a functional and morphological standpoint?

Reply: We use low passage number MIN6 cells (18-40) and make sure that they are glucose responsive at all times. Shown below are results from insulin ELISA measurements of GSIS from MIN6 cells passages 18-40 (note: only one replicate). As can be seen, the cells retain strong glucose-stimulated insulin secretion between these passages. We do not use cells above passage 40, and have therefore not investigated what happens with ciliary signaling when these cells eventually lose glucose responsiveness.

3) How can it be mechanistically explained that neither the number of somatostatin receptors on the primary cilia nor their sensitivity to the hormone are affected after 18 h of stimulation?

Reply: These are very good questions. When it comes to the number of receptors, we don't believe that our analysis method, where we look at different cell populations at two time points (15 min and 24h somatostatin treatment), is sensitive enough to determine the exact number of receptors in the cilium. The point we want to make is that SSTR3 is still present in the cilium even after 18h incubation with somatostatin. This is also consistent with the GLI2 nuclear translocation data, where we see that GLI2 is maintained in the nucleus 18h after somatostatin addition. When it comes to receptor sensitivity to somatostatin, it is important to note that the sensitivity (determined by recordings of ciliary Ca²⁺) was assessed in cells that had been allowed to recover in somatostatin-free medium for one hour prior to stimulation. We can therefore not rule out that desensitization occurs, although the lack of receptor removal from the cilium and the sustained GLI2 nuclear translocation speaks against this. Other studies have shown that SSTR3 exits the cilium upon agonist binding through a mechanism dependent on β -arrestin 2 and ubiquitination (e.g. <https://doi.org/10.1128/MCB.00765-15> and <https://doi.org/10.1083/jcb.202003020>). While these studies show reduced accumulation of SSTR3 in the cilium after 30-60 min of somatostatin stimulation, the receptor is not completely gone from the cilium and it is not tested whether the remaining pool of ciliary receptors can engage in signaling. In addition, these studies look at overexpressed SSTR3 that has been modified to be compatible with live cell fluorescence microscopy whereas we perform immunostainings of endogenous SSTR3. It is possible that overexpression and/or modifications to the receptor also influence receptor removal from the cilium.

4) What physiological implications are there for islet hormone release in general upon the shortening of islet cell primary cilia and thereby impaired connectivity with delta-cells in T2DM?

Reply: We do not believe that ciliary sensing of islet-derived signals (e.g. somatostatin) has an acute effect on secretion. Instead, we believe that the cilia sense the islet microenvironment, integrate these signals as cilia-intrinsic second messenger changes that in turn lead to transcription factor activation and long-term adaptation, and that somatostatin is one of the molecules sensed by the cilium. We show here that GLI2 activity is regulated by somatostatin, but recent studies in other cell types have also indicated that other pathways may be under ciliary control, including CREB and chromatin accessibility (PMID: 36055200; PMID: 35695071). The islet microenvironment is changed in T2DM, and cilia may facilitate cellular adaptation to these changes. However, shortening of cilia will negatively impact the cilia's ability to sense and transmit signals (doi: [10.1016/j.tcb.2023.05.005](https://doi.org/10.1016/j.tcb.2023.05.005)). As we show now in new Suppl. Fig. 7, 18h treatment with somatostatin resulted in cilia shortening and in impaired glucose-induced Ca²⁺ signaling. Ca²⁺ is the main trigger of insulin granule exocytosis and loss of regular glucose-induced insulin pulses are seen in T2DM (<https://doi.org/10.2337/diab.30.5.435>). It remains unclear if it is the altered connectivity with delta cells that drive cilia shortening in T2DM, although based on our observations this would more likely lead to the opposite. We now also show that culture of human islets from non-diabetic donors in a palmitate-containing diabetogenic medium also resulted in cilia shortening (new Fig. 3I), indicating that environmental factors contribute to cilia

shortening in T2DM. To better understand the molecular mechanism leading to cilia shortening is an important future research goal for us.

5) What is the molecular mechanism underlying this shortening? Is there a metabolic explanation and if yes is this process reversible upon normalization of blood glucose concentration?

Reply: This is possible but not easy to test. Human islets are isolated and kept in culture for days before experiments, in a medium containing 5.5 mM glucose. Studies in other cell types have shown that glucose deprivation causes ciliogenesis and shortening of cilia (DOI: [10.1242/jcs.208769](https://doi.org/10.1242/jcs.208769)). It has also been shown that elevated levels of FFA (specifically palmitic acid) leads to cilia shortening in hypothalamic neuronal cilia through inhibition of autophagy (<https://www.nature.com/articles/s41419-022-05109-9>). Induction of short cilia in the hypothalamus of adult mice has also been shown to increased food intake and decreased energy expenditure, leading to a positive energy balance. Mice with short hypothalamic cilia also exhibited attenuated anorectic responses to leptin, insulin, and glucose, which indicates that leptin-induced cilia assembly is essential for sensing these satiety signals by hypothalamic neurons (PMID: 24667636). Culture in palmitate-containing medium is commonly used as an in vitro model of type-2 diabetes (doi: [10.3389/fendo.2023.1275835](https://doi.org/10.3389/fendo.2023.1275835)), and inhibition of autophagy has been observed in islets from T2DM, although the cilium was not investigated in these studies. It is therefore plausible that environmental changes in T2DM drives changes in primary cilia length and function. Consistent with this, we now show that culture of human islets from non-diabetic donors in a diabetogenic environment (7 days in the presence of 0.5 mM palmitic acid) leads to cilia shortening, indicating that the shortening may be an adaptation and/or consequence of the diabetogenic environment (new Fig. 3I). In future studies, we want to identify transcriptional changes that involve cilia signaling and use these changes as readout for islet cell function. This will enable intervention studies similar to the one proposed by the reviewer.

6) Is cAMP also diffusing freely between the cytosol and the cilium under T2DM conditions?

Reply: We have not yet tested this, mostly because we did not receive islets from T2DM donors during the revision period. Live material from these donors is very rare, and we typically only get material from 1-2 donors per year. Given the small size of cAMP, we find it unlikely that its trafficking will be influenced by changes in the expression of e.g. genes encoding for proteins in the ciliary transition zone. However, it is possible that changes in the expression of e.g. ciliary adenylate cyclase or phosphodiesterase in T2DM could influence cAMP dynamics in the cilium. In fact, polymorphisms in AC5, which is a ciliary protein in beta cells (see STED microscopy image below from a mouse islet beta cell), leads to reduced gene expression and is associated with increased risk of type-2 diabetes (PMID: 28684635). It would be interesting in the future to further explore cAMP dynamics in T2DM or experimental diabetes models.

7) The relationship between somatostatin, cAMP and Ca²⁺ signaling in the cilia is interesting but is not clear. What is going on mechanistically and what does it mean in terms of signaling in the primary cilia?

Reply: We show that both cAMP lowering and inhibition of PKA induce Ca²⁺ influx in the cilium. We have now further substantiated these findings by new experiments (Suppl. Fig. 6) showing that

somatostatin-induced Ca^{2+} signaling is counteracted by simultaneous elevation of cAMP through either G α s-coupled receptor activation or light-driven cAMP formation. So far, we have been unable to determine the identity of the ion channels responsible to the influx. This is an important future aim that will also help us understand how PKA controls Ca^{2+} signaling in the cilium. Ca^{2+} increases, in turn, appears to be a permissive signal for GLI2 activation. This mechanism is also unclear but may involve local modulation of cAMP levels by direct effects on adenylyl cyclase and phosphodiesterases. In addition to the local crosstalk in the cilium, lowering of ciliary cAMP long-term modulate whole cell cAMP responses to the insulin secretagogue GLP-1 (Fig. 8M-O). We now also show that the same conditions alter beta cell Ca^{2+} signaling (new suppl. Fig. 7). Under control conditions, glucose induce regular, slow Ca^{2+} oscillations that are known to drive pulsatile insulin secretion, but 18h treatment with somatostatin transforms these oscillations to more rapid and irregular Ca^{2+} changes in a SST3-dependent manner. Importantly, the hedgehog agonist SAG only modestly altered Ca^{2+} signaling, indicating that somatostatin is not simply a modulator of hedgehog signaling.

8) Why is exposure to SAG associated with a slight lengthening of cilia and somatostatin exposure associated with a small reduction in cilia length?

Reply: We thank the reviewer for raising this question. Going over the source data for this figure panel, we realized that there had been a mix-up of data when designing the scatter plot, where in the previous version the control group was the SAG-treated group and vice versa. In the new, corrected, panel 8C we show that both SAG and somatostatin causes shortening of the cilium. We have also corrected the text in results to “Exposure to both SAG and somatostatin was associated with a slight shortening of cilia (Fig. 8C).” We sincerely apologize for this.

Referee #2:

This manuscript evaluates regulation of somatostatin (SST) signaling in pancreatic islet cells. The results demonstrate that SST, which is released from filopodia-like extensions of δ -cells, signals to SSTR3 (SSTR3), a G protein coupled receptor (GPCR) that localizes to primary cilia of β -cells. The authors demonstrate that SSTR3, which couples to Gai, leads to cAMP reduction in SST responding cells. Consistent with a previous report (doi.org/10.1080/19382014.2023.2252855), the study demonstrates that activation of primary cilium localized SSTR3-Gai signaling leads to changes in Ca²⁺ flux. This manuscript demonstrates that increased Ca²⁺ in the primary cilium occurs downstream of SSTR3. These signals are proposed to facilitate downstream effector responses, including changes in transcription regulation through noncanonical activation of the Hedgehog transcriptional effector GLI2. These results suggest crosstalk between SSTR and Hedgehog signaling pathways occurs at the primary cilium. The manuscript provides high resolution imaging of δ -cell contact with β -cell primary cilia, makes use of some novel tools to track ciliary cAMP, and provides an interesting model for GPCR signal crosstalk in the primary cilium. The work will likely be of broad interest, pending some additional experimentation.

Key Questions

1) It is noted in Figures 2E, I, J, and 8C that primary cilia of β -cells shorten following SSTR3 activation, but implications of ciliary SST-induced shortening are not discussed. Is this a result of activation of Gai coupled SSTR3 in the primary cilium? Other studies demonstrate that changes in ciliary cAMP can influence ciliary length and that these length changes can impact signal output ([doi: 10.1038/ncb3029](https://doi.org/10.1038/ncb3029); doi.org/10.1083/jcb.202306002; doi.org/10.1016/j.cub.2009.11.072).

Reply: Yes, this is very likely the case, but it does not seem to be the case for all stimuli since we previously showed that GABA is without effect on cilia length. Serotonin (Gas-coupled) increase cilia length in neurons (within hours) [PubMed: 28931427], while dopamine can either cause cilia lengthening (D1R \rightarrow Gas) or shortening (D2R \rightarrow Gai) [PubMed: 34665407; 24830745]. Changes in circadian rhythm genes (as has been shown in T2D) may also cause changes in cilia length [[doi: 10.1101/2022.01.26.477948](https://doi.org/10.1101/2022.01.26.477948)]. Ca²⁺ may also play a role here. We show that cAMP lowering or inhibition of PKA triggers ciliary Ca²⁺ signaling, which has been shown to cause cilia shortening in other cell types (PMID: 20096584). In addition, we have now included data (Fig. 3I) showing that culture of human islets from non-diabetic donors in a diabetogenic environment (0.5 mM palmitic acid) induce cilia shortening to the same extent as that observed in islets from type-2 diabetic donors. This indicate that pathologic changes in the islet environment can drive cilia shortening. In the future, we hope to further determine the mechanistic coupling between ciliary cAMP and Ca²⁺ changes to understand how these two second messengers cooperate to control ciliary signaling.

2) Figure 3: How does SSTR3-responsive shortening seen in Figure 2 relate to the ciliary shortening that is observed in β -cells from type II diabetes samples?

Reply: First, we do not observe any apparent difference in the localization of SSTR3 to primary cilia when comparing islets from non-diabetic and T2D donors (see Fig. 3O). As discussed above, it is most likely that the shortening of the cilium observed in T2D will have an impact on the cilium's ability to detect signals and initiate signaling. We observe that the number of delta cells is reduced in T2D, which is similar to what has been recently reported (PMID: [36834860](https://pubmed.ncbi.nlm.nih.gov/36834860/)) and also in line with measurements of somatostatin secretion from obese and high fat diet-fed mice (PMID: 26076035; PMID: 32446876). This hyposecretion of somatostatin would not result in reduced cilia length. We therefore believe that the shortening is unrelated to somatostatin secretion, but may impact somatostatins' ability to initiate signaling in cilia.

3) Figure 4: It is noted that the SST1.0 sensor that is used to report plasma membrane and primary cilium SSTR3 activation shows a higher response frequency in primary cilia compared to the plasma membrane. Please provide a more detailed explanation of what this means in the results section discussing Figure 4K, L. It is stated that these data show that "...somatostatin secretion preferentially occurs towards primary cilia". How? Could it be that the increased response frequency in cilia is due to a higher local concentration of receptor in the small

primary cilium compared to more sparsely localized receptor on the plasma membrane?

Reply: Yes, that is possible. However, the enrichment in the cilium compared to the plasma membrane is modest and the sensor response to 1 μ M exogenous somatostatin is also identical in cilia and plasma membrane (Fig. 4J), indicating that the sensor response is not sensitive to cellular localization. Our interpretation is that cilia are positioned closer to the somatostatin release sites on delta cells, and therefore more easily detect the signaling peptide. It is important to keep in mind that the SST1.0 is an engineered receptor and that its binding to somatostatin may not accurately reflect that of endogenous somatostatin receptors in the islet cells. Our main point here is not to say that cilia exclusively detect somatostatin, but rather that they also detect somatostatin. We have now rephrased the text to “*The SST1.0 response amplitude was not different between primary cilia and cilia-adjacent plasma membrane when matched for distance to δ -cells (average distance between primary cilium and δ -cell was $4.3 \pm 0.9 \mu\text{m}$ and between plasma membrane and δ -cell $3 \pm 0.7 \mu\text{m}$; NS), but the response frequency was 1.8-fold higher in primary cilia (Fig. 4K, L). Quantifications showed that the responses in both plasma membrane and primary cilia inversely correlated with the distance to δ -cells, indicating that the sensor reports local somatostatin release (Fig. 4M). Importantly, the addition of 1 μM exogenous somatostatin resulted in identical SST1.0 fluorescence changes in both plasma membrane and primary cilia, showing that the sensitivity of SST1.0 is not location-dependent (Fig. 4J). Together, these results show that primary cilia are exposed to endogenously released somatostatin, and also indicate that somatostatin secretion is at least partially directed towards primary cilia.*”

4) Figure 6: If an antibody is available, please provide a western blot to demonstrate the level of SSTR3 protein reduction following knockdown (Figure 6D).

Reply: We show in Supplementary Figure 5 that the SSTR3 immunoreactivity in the primary cilium is specifically reduced in SSTR3 KD cells compared to control cells.

5) Figure 7 shows that Ca²⁺ responses are dependent on SSTR3 because the response is reduced following SSTR3 knockdown. The authors previously showed that GABAB1-Gai signaling controls Ca²⁺ signaling in β -cells (DOI: 10.1083/jcb.202108101). Is GABAB1 involved in this response? GABAB1 should be knocked down to determine whether there is synergy between SSTR3 and GABAB1 - or if the Ca²⁺ response occurs directly through SSTR3-Gai signaling. If it does involve synergy with GABAB1 - why is this necessary? Why can't SSTR3-Gai signaling induce a Ca²⁺ response?

Reply: This is an interesting point. In the case of GABA-GABBR1, this response does not seem to require Gai-signaling since we could not block it by pertussis toxin treatment (PMID: 36350286). This is different from the response triggered by somatostatin-SSTR3, which was strongly impaired by pertussis toxin treatment. These two processes therefore appear to operate through distinct mechanisms that converge on Ca²⁺ influx. The question about cross-talk is relevant, because both GABA and somatostatin are released in a manner that is not very strongly dependent on glucose, and therefore likely sensed by the islet cell cilia at the same time. We have now performed experiments in MIN6 pseudoislets with reduced GABBR1 expression and show that cilia in these islets also exhibit somatostatin-induced Ca²⁺ signaling. We also show that the effects of SST and Baclofen (GABBR1 agonist) were not additive, indicating a common source for Ca²⁺ influx into the cilium. Because this study focuses on somatostatin, we prefer not to include this data in the manuscript but show it below here for the reviewer to evaluate.

6) Figure 8A, 8I: Please use DAPI to mark nuclei. Why is no GLI2-GFP evident at the tip of the primary cilium following SAG stimulation in 8A? GLI2 levels typically increase at the tip of the primary cilium in response to SHH pathway activation.

Reply: We now show in new Figure 8A immunostainings of mouse islets expressing GLI2-Halo^{JFX650}, where it is clearly shown that both SAG and somatostatin induce nuclear accumulation of GLI2 (nuclei shown by Dapi staining). Also unstimulated islets to some extent contain cells with GLI2 present in both the cilium and nucleus, perhaps indicating endogenous release of Hedgehog or somatostatin. Below are shown examples where it is clear that GLI2-Halo localizes to both the nucleus and cilium. This is not as obvious in the low-magnification images shown in 8A.

7) Figure 8C shows that SAG treatment increases primary cilium length, but this result is not discussed. What is going on here?

Reply: We thank the reviewer for raising this question. Going over the source data for this figure panel, we realized that there had been a mix-up of data when designing the scatter plot, where in the previous version the control group was the SAG-treated group and vice versa. In the new, corrected, panel 8C we show that both SAG and somatostatin causes shortening of the cilium. We have also corrected the text in results to “Exposure to both SAG and somatostatin was associated with a slight shortening of cilia (Fig. 8C).” We sincerely apologize for this.

8) Figure 8J: Why does SAG reduce SSTR3 expression?

Reply: We don't know, but speculate that this may be part of a homeostatic response that prevents excessive lowering of ciliary cAMP levels. In follow-up work we will more broadly explore how different ciliary signaling pathways cross-talk and what the outcome of this cross-talk is.

9) Figure 8A and K-L: The authors use GLI2-GFP nuclear accumulation as a surrogate for transcriptional activation. Please assess activation of SHH target genes like Ptch1 and Gli1 to determine whether nuclear GLI2 induces a transcriptional response. GLI2 and GLI3 are not transcriptional targets, and the expression changes shown in panels K and L are very modest, so are not good indicators of a SHH/GLI transcriptional response.

Reply: We have now performed measurements of GLI1 and PTCH1 expression in control or SSTR3 KD MIN6 cells treated or not with 100 nM SAG or 100 nM somatostatin for 18h. The results of these experiments are shown below. As can be seen, both SAG and SST modestly increase GLI1 expression in control cells but not in SSTR3 KD cells while there are no changes in PTCH1 expression under any of the tested conditions. A complicating factor with these measurements is that the Hh pathways appears to be constitutively active in beta cells (e.g. we see localization of SMO to the cilium in untreated cells and GLI2 is both ciliary and nuclear in around 25% of cells under the same conditions). We initially performed experiments looking at GLI2 nuclear translocation because it is the most robust and well-established readout for ciliary Hh signaling. However, whether this pathway is the most important ciliary pathway in beta cells remain unclear. Addressing this would require a more broad and unbiased approach than determining the expression of individual, selected genes. In the future, we will attempt to answer these questions using e.g. ATAC-seq. We hope that the reviewer agrees that such experiments are beyond the scope of the present study.

Minor Suggestion

10) The authors refer to GLI2 "processing" in the results and discussion. GLI2/3 "processing" typically refers to proteolytic processing of GLI2/3 that removes the transcription activation domain to generate GLI repressors. If you mean "activation", please write "activation".

Reply: Thank you for pointing this out. We have now corrected it to activation.

11) The discussion states that "...Ca²⁺ has been shown to be involved in the Hedgehog response...." and references the Belgacem & Borodinsky, 2011 PNAS paper (DOI: 10.1073/pnas.1018217108). The SMO-Gai response described in the 2011 paper describes a non-canonical GLI transcription independent SHH signal mediated by IP3 and Ca²⁺. This should be clarified in the discussion because, as written, the authors' statement might be interpreted as suggesting that Ca²⁺ has been demonstrated to influence canonical GLI activity. This has not been shown in the literature.

Reply: Thanks for clarifying this. We have now rephrased the discussion to: "*Activation of the Hedgehog pathway also leads to increase in ciliary Ca²⁺ (Delling et al., 2013)(Belgacem & Borodinsky, 2011; Teperino et al., 2012), but it is not known if this increase is required for downstream GLI-mediated transcriptional responses. Similarly, both Ca²⁺ increases and cAMP lowering have been shown to promote cilia shortening, but whether the two messengers operate through a common pathway is not known (DOI 10.1016/j.cub.2009.11.072).*"

12) The authors state in the discussion and indicate in their model (Figure 8P) that SSTR3-mediated Gai activation leads to reduced PKA activity. It doesn't look like PKA activity was directly tested in this study. Please clarify.

Reply: Correct, we didn't show it. However, we show that inhibition of PKA induced Ca²⁺ signaling in the cilium (Fig. 7G, H), and that Ca²⁺ increases are required for somatostatin-mediated GLI2 nuclear translocation (Fig. 8I). It has also previously been shown that inhibition of ciliary PKA by expression of a cilia-localized PKA-inhibitor increase the expression of GLI target genes (PMID: 33932338). However, since we don't directly show that addition of somatostatin reduces PKA activity, we have added "?" to the arrow in our model pointing from SSTR3 to PKA.

Referee #3:

General Summary:

In this manuscript the authors create a compelling case for the unique role that ciliary SSTR signaling plays in maintaining proper beta cell function through paracrine interactions with the delta cell. By combining EM and fluorescent microscopy techniques, the authors functionally map out the presence and signaling contributions of ciliary SSTR3 using established sensors and techniques. The data presented are fundamentally similar to the approaches and findings of ciliary signaling importance in recent papers focused on neurons (Sheu 2022, Reiter 2021). Overall, I find several aspects of this manuscript very exciting, specifically, the differences between SSTR activation in the cilia (calcium increase) and cytosol, coupled with the relationship between PKA activity and this novel calcium pathway. This manuscript will serve as a launching point for future studies of this pathway and lays out the importance of understanding the relationship between the cilia and metabolic disease as others have before (Hughes 2020). Finally, it is appreciated the length with which the authors went to place their system in the context of known islet biology, with the use of ghrelin to stimulate delta cells in figure 7 as an example. Overall, this manuscript was an interesting and fun read, but also one that provoked several questions:

There are however two areas where the manuscript would need some careful reconsideration. The first is that the purported contribution of SSTR5 in beta cell cilia and the second is the coupling of FFAR4 to cAMP in cilia. In both examples, the authors cite some of the prior work that aligns with this narrative, but do not cite and discuss the work of others that is less easily reconciled with this model. Ultimately, these parts of the manuscript distract from an otherwise exciting manuscript that represents a significant amount of innovative experimental work that is well presented. Finally, the connection with Gli2 towards the conclusion of the paper is established only using an overexpression model, while these data could be strengthened considerably with evidence of endogenous Gli2 playing a comparable role.

There are a series of other comments and suggestions that would be readily addressable that are provided to the authors for their consideration.

specific major concerns essential to be addressed to support the conclusions

1) The authors make their case for SSTR5 expression on the primary cilium of beta cells, showing immunohistochemistry that shows a ciliary pattern in mouse tissue. This staining is not shown in human tissue and is absent in Min6 cells (mouse derived). These findings contradict previous results from multiple labs using SSTR5 as a ciliary negative control (e.g. Berbari 2008). The authors cite a few papers in support of the expression of SSTR5 in islets, but fail to acknowledge strong evidence of no SSTR5 expression. Specifically, bulk transcriptome data from purified alpha, beta, and delta cells from several groups - cited elsewhere in the manuscript - demonstrates no detectable SSTR5 in mouse islet cells, with one of these papers even dedicating a figure to the expression of a select set of GPCRs including the SSTRs. Once the paper reaches the actual characterization of the model using ciliary sensors, it appears as though SSTR3 dictates downstream ciliary processes, indicating that it is the main/only SST receptor in mouse beta cell cilia. This again calls into question any role of SSTR5. Unless the antibody result is backed up by proper validation in SSTR5 null beta cells, these findings largely distract from your story. It appears that this line of inquiry flowed from the SST sensor, that is based on SSTR5 and in the hands of these investigators ends up in cilia when overexpressed. Others using the same sensor have not seen the same expression pattern. The remaining data on SSTR5 expression are based solely on antibody staining, with evidence of no functional contribution of SSTR5. My recommendation is to simply leave the SSTR5 observations out. I think your story is much more compelling without them.

Reply: The reviewer is correct in that we first observed localization of the somatostatin sensor SST1.0, which is based on SSTR5, to primary cilia. This prompted us to look into the distribution of endogenous SSTR5 in islets cells. Indeed, using two different antibodies we see SSTR5 positive cilia, although neither the fraction of cilia positive for SSTR5 nor the extent of receptor enrichment reaches the levels seen for SSTR3. Whether these receptors are in fact functional is unknown, and at least in clonal MIN6 beta cells, SSTR3 appears to be the only ciliary SSTR. We therefore agree with the reviewer that the best strategy is to remove the data on SSTR5 from the main figure. The immunostainings against SSTR5 are now only used to support the observation that SST1.0 is ciliary with the following text: "*Immunostaining against the ciliary axoneme confirmed the localization of SST1.0 to the ciliary membrane (Fig. 4H), and immunostainings in mouse and human islets confirmed the ciliary localization of SSTR5 (Suppl. Fig. 1).*"

2) This question again arises with the use of an SSTR5 based SST GRAB sensor which appears to localize preferentially to the cilia in this manuscript, but shows no similar cilia localized phenotype in islets within the 2023 paper it is cited from. Again, this is unsettled science and should be explained or addressed in the discussion at the least.

Reply: We observed significant localization of the SST1.0 sensor both in cilia and at the plasma membrane. The study where SST1.0 was first described ([0.1126/science.abq8173](https://doi.org/10.1126/science.abq8173)) did not emphasize the cilia-localization. We have contacted Dr. Huan Wang, the first author of the original study, who confirmed that they also observed both plasma membrane and cilia-like localization. However, their focus was on demonstrating that the SST1.0 sensor can respond to endogenous SST secretion, and they did not further investigate whether the cilia-like structures were indeed cilia. Moreover, based on our experience, cilia are more readily observed on the surface or relatively shallow optical planes of the islets. The compact nature of the islets means that cilia located in the central areas are often squeezed into narrow spaces between cells, appearing as dots in confocal images (similar to the "dots" visible in [0.1126/science.abq8173](https://doi.org/10.1126/science.abq8173), fig. 3). Sometimes, cilia extend along the gaps between cells and may not be distinguishable from the cell membrane under confocal microscopy examination, unless specifically labeled. Secondly, we observed variability in cilia-localization of the SST1.0 sensor; in most islets, we detected clear cilia-localization, but this was not consistent across all samples. In summary, our results indicate that the SST1.0 sensor localizes to cilia of islet cells, though this localization is not exclusive. Shown below are example images of mouse islets expressing SST1.0 delivered by adenoviral transduction (top) or transgenic expression (bottom).

Adenovirus SST1.0

Transgenic SST1.0

3) The other main comment relates to FFAR4. It is presented as a cilia GPCR in beta cells that couples to Gas and therefore - activates cilia cAMP. However, there is a substantial body of work by multiple groups that indicates that FFAR4 signals via Gq by default (see eg. <https://doi.org/10.1016/j.celrep.2024.114509>).

The manuscript also ignores a body of literature that describes that primary mouse beta cells do not express FFAR4 mRNA, but that this is expressed instead by delta cells (e.g. PMID: 33484949). Perhaps closer attention should be paid to the models used. The authors switch between mouse islets and min6 cells regularly. While this is common and often quite reasonable as beta cell lines offer experimental benefits over primary islet experiments, in this case min6 cells likely do not model the signaling in primary beta cells, with min6 cells express FFAR4 on their cilia, while primary beta cells do not. The experiments where Tug891 and SST are tested together (Figure 6) are then useful to interrogate the cilia-specific mechanisms of signaling, but likely do not reflect the actual signaling dynamics in primary beta cells where the stimulation of insulin secretion by FFAR4 agonists is mediated by the inhibition of delta cells: delta cell-specific deletion of FFAR4 abolishes the effects of FFAR4 agonism on insulin secretion. None of this is considered or discussed.

Reply: We thank the reviewer for raising these points, which we think stems largely from us not explaining clearly enough the rationale for the experiments using FFAR4 agonists. First, we agree with the reviewer that plasma membrane-localized FFAR4 primarily couples via Gq. However, when in the cilium this receptor is clearly Gas-coupled (see e.g. [10.1016/j.cell.2019.11.005](https://doi.org/10.1016/j.cell.2019.11.005)), which is also consistent with the results shown in Figure 6C and 7G. We agree that the results presented here may lack physiological relevance. However, our point was never to suggest that free fatty acids, via FFAR4, may modulate somatostatin signaling in the cilium under physiological conditions, but we only used it as a means to trigger cAMP increases in the cilium downstream of a receptor (instead of using adenylate cyclase activators). The purpose was to show that the cilium is an integrator, and that it is the combined activation of Gas and Gai that control output from this organelle. In addition to this, we have now also included data from MIN6 pseudoislets where we show that somatostatin-induced ciliary Ca²⁺ signaling is counteracted by the addition of the Gas-agonist PACAP (new suppl. Fig. 6), and that light-induced cAMP synthesis in the cytosol also counteract somatostatin-induced ciliary Ca²⁺

signaling (new suppl. Fig. 6). Altogether, we believe these results show that ciliary cAMP and Ca²⁺ signaling can be influenced by cytosolic cAMP changes. The new text reads:

“These results indicate a connection between cAMP lowering and the initiation of Ca²⁺ signaling in the primary cilium. To test this more directly, we exposed mouse islets expressing 5HT₆-GGECO1 to 100 nM somatostatin, followed by the addition of 100 μM of the FFAR4 agonist TUG-891 or 100 nM of the Gas-coupled PAK1 receptor agonist PACAP (pituitary adenylate cyclase activating peptide). Consistent with an inverse relationship between cAMP and Ca²⁺, we found that the somatostatin-induced Ca²⁺ signaling was suppressed when cAMP was elevated (Fig. 7G and Suppl. Fig. 6). Similar suppression was also seen when cytosolic cAMP levels were elevated by blue light-induced activation of bPAC (REF, Suppl. Fig. 6).”

4) The results observing translocation of GLI2-GFP in min6 cells are quite striking. While this experiment determines that GLI2 will translocate to the nucleus, it does not demonstrate that endogenous GLI2 does the same. How does the overexpression of GLI2 compare to endogenous protein and mRNA levels, which are low to not detected. Is there compelling data that link SST stimulation of primary beta cells to the activation of GLI2-dependent gene expression pathways?

Reply: To visualize GLI2 subcellular localization in mouse islets, we have now generated and expressed an adenoviral vector-based GLI2-Halo construct. Under resting conditions (5 mM glucose), GLI2 was primarily cytosolic, but around 25% of the islet β-cells also had GLI2 in the nucleus. Stimulation for 18h with either 100 nM somatostatin or 100 nM SAG resulted in a doubling of the fraction of nuclei positive for GLI2, often accompanied by GLI2 accumulation in the primary cilia. These results are very similar to those obtained in MIN6 cells and are now shown in Fig. 8A and B and in supplementary figure 7. We have attempted to immunostain endogenous GLI2 using two different antibodies but have not consistently seen either cilia localization nor nuclear translocation in response to SAG. Whether this reflects lack of antibody specificity or low endogenous expression of GLI2 is unclear. Based on public repositories (e.g. humanislets.com) GLI2 is expressed in adult islets, although at a lower level than during development. In the future, we will explore where somatostatin-induced lowering of ciliary cAMP may involve other transcriptional regulators, e.g. CREB.

minor concerns that should be addressed

5) The citation of SSTR2 as the main receptor in human alpha and beta cells does not reflect the known expression data in mouse islet endocrine cells, where there is evidence of no expression of SSTR2 in beta cells. Since the data in this manuscript are primarily obtained in cells of mouse origin, introducing the expression pattern in mouse would seem appropriate.

Reply: We thank the reviewer for bringing our attention to this. We have now rephrased it to: *“SSTR2 is considered the predominant somatostatin receptor in human islet β-cells (Braun, 2014; Kailey et al., 2012), while SSTR3 appears to be the major isoform in mouse β-cells, at least at the mRNA level (DiGrucchio et al., 2016)”*

6) The suggestion that cilia serve as antennas needs to contend with the fact that nutrients and paracines would by simple diffusion travel the length of the cilia within a very short amount of time. Instead, the view of cilia as isolated signaling compartments where signaling can take place independent from the cytosol (see PMID: 37072495, PMID: 38366037) is more in line with the data presented in this manuscript.

Reply: We believe that one does not exclude the other. We think that the cilium can function as an antenna for sensing local changes in e.g. receptor ligands, but that in addition to this it also functions as an isolated signaling compartment (so not all signals detected may be “directed” towards the cilium). We have now reformulated this in the text to *“The presence of receptors in the primary cilium enables these structures to function as antennas that sense changes in the local cell environment, and downstream signaling is often restricted to the cilium, which therefore forms an isolated signaling compartment (Delling et al., 2013; Hansen et al., 2022; Sanchez et al., 2023; Truong et al., 2021).”*

7) In that same statement, the authors speak of 'enrichment of receptors' in the cilia. This

statement requires qualification, a select number of GPCRs that is listed by the authors is of course expressed by cilia. These are the receptors that are 'enriched in the cilia' and for which signals can be sensed selectively at that location. However, a beta cells expresses upwards of 150 GPCRs, and most are not expressed selectively by cilia. Therefore, for the class of GPCRs, speaking of 'enrichment' in the cilia is not true.

Reply: We understand the point raised by the reviewer. We are also of the opinion that few receptors are exclusively in the cilium, some receptors are enriched in the cilium, but the vast majority of ciliary receptors are just present. Due to the dimensions of the cilium, with a diameter of around 200 nm, we will irrespective of imaging technique (TIRF or confocal) likely overestimate the extent of receptor accumulation since were collecting fluorescence from more membrane per volume than is the case for e.g. the plasma membrane (we have made calculations on this in a previous publication: PMID: 36350286). We have now rephrased the sentence in the introduction to “*The presence of receptors in the primary cilium enables these structures to function as antennas that sense changes in the local cell environment, and downstream signaling is often restricted to the cilium, which therefore forms an isolated signaling compartment (Delling et al., 2013; Hansen et al., 2022; Sanchez et al., 2023; Truong et al., 2021).*”.

8) The cilia length measurements appear to be reduced to a single average per donor (Figure 3G, H). I assume that many cilia per donor were measured. The authors should explore how these intra-donor replicates can be better reflected in the data and statistical analysis.

Reply: In suppl. fig. 2A and 2B, we show scatter plots of all cilia measured for all of the donors in both non-diabetic and T2D donors. As can be seen from these distributions, there is a clear reduction in cilia length in T2D. Organ donors are heterogenic, and for this reason we decided to plot the data in the main figure based on donors (to emphasize that all T2D donors have shorter cilia than all non-diabetic donors). The statistical test was also performed on the number of donors and not number of islets or cilia (although these numbers are given in the main figure legend).

9) The experiments in Figure 4 that measure cilia signaling to SST in delta cell adjacent cilia are elegant and nicely done! The observation that SST stimulates calcium in primary cilia, but not the cytosol of the same cells is surprising. What is the source of this calcium?

Reply: We have not specifically investigated the source of Ca²⁺ in response to somatostatin, but shown in a previous publication (PMID: 36350286) that cilia Ca²⁺ signaling depends on influx and not release from intracellular stores. We therefore believe that the source in this case is extracellular.

10 The use of ghrelin to elicit the local release of SST in figure 7 is a compelling piece of experimental data that takes full advantage of previously characterized local crosstalk within the islet.

Reply: Thanks!

11) The authors claim in the discussion that it is not known if SST secretion is changed in T2D. PMID: 26076035 show that SST secretion in obob islets in response to glucose is impaired. PMID: 32446876 show comparable findings with regards to SST secretion in a HFD model.

Reply: We thank the reviewer for making us aware of these studies. We now cite them, and have rephrased this part of the discussion to: “*Somatostatin secretion has also been shown to be reduced in both diabetic ob/ob and high fat diet-fed mice (PMID: 26076035; PMID: 32446876), and a recent study showed reduced numbers of ghrelin-secreting cells and reduced circulating levels of ghrelin in human type-2 diabetes (Lindqvist et al., 2020), which may explain the impaired somatostatin release*”

12) The model in figure 8 is not helpful. It tries to reconcile observations from mouse and human primary islet cells and mouse cell lines. Specifically, SSTR3 is at the cilia, and not the PM. SSTR2 is not expressed by mouse beta cells (human beta cells are perhaps different, but almost all data in this paper originate in mouse). As discussed earlier, there is strong evidence of no expression of SSTR5.

Reply: We have now revised the figure to only depict SSTR3 in the cilium and not mentioning a specific SSTR isoform from plasma membrane localized receptors.

13) The legend for figure 7 does not include a description of panel K and L.

Reply: We are sorry about that. This has now been corrected.

Dear Olof,

Thank you for submitting a revised version of your manuscript. We have now received input from one of the original reviewers, who finds that their main concerns have been addressed satisfactorily and recommends acceptance of the manuscript after a textual revision in response to their remaining concerns. I have also gone through your responses to the concerns by the other two reviewers and I find them reasonable.

Additionally, there remain a few editorial points that need addressing before I can extend official acceptance of the manuscript:

1. Please check that the funding information is correct and identical both in the manuscript and our online system.
2. Please make sure that the order of the sections in the manuscript is as follows: Abstract / Keywords / Introduction / Results / Discussion / Methods / Acknowledgments / Disclosure and competing interests statement / References / Figure legends / Tables and their legends / Expanded View Figure legends.
3. Please remove the Reagents and Tools Table from the manuscript text file and include the information provided in it and the similarly labelled Excel file in a single table using our Reagents and Tools Table template (.docx), which you can find in our author guidelines: <https://www.embopress.org/page/journal/14602075/authorguide#structuredmethods>
When submitting your revised manuscript, please do not include the Reagents and Tools Table in the Methods section of the manuscript but upload it as a separate file choosing the file type "Reagent Table".
An example of a Method paper with Structured Methods can be found here:
<https://www.embopress.org/doi/10.15252/msb.20178071>.
4. We require a Data Availability Section at the end of Materials and Methods. As far as I can see, no data deposition in external databases is needed for this paper. If I am correct, then please state in this section: This study includes no data deposited in external repositories. Further information can be found at
<https://www.embopress.org/page/journal/14602075/authorguide#dataavailability>
5. Please rename "Disclosure statement and competing interests" section into "Disclosure and competing interests statement" (further info: <https://www.embopress.org/page/journal/14602075/authorguide#conflictsofinterest>).
6. CRediT has replaced the traditional author contributions section because it offers a systematic, machine-readable author contributions format that allows for more effective research assessment. Please remove the Authors Contributions from the manuscript and use the free text boxes beneath each contributing author's name in our online submission system to add specific details on the author's contribution. More information is available in our guide to authors.
7. Please remove DOIs from references to already published articles. According to The EMBO Journal style, DOIs should be used only for preprints and datasets. Please see further information here:
<https://www.embopress.org/page/journal/14602075/authorguide#referencesformat>
8. Please add callouts to individual EV figure panels in the manuscript text.
9. In the Appendix, please add the manuscript title in the front page and include page numbers in the table of contents.
10. Our data editors have flagged the following issues in figure legends that need correcting:
 - Please provide information on the number and nature of replicates in the legends of figures EV 1a-b.
 - Please note that $n=2$ in figure 8j. In such cases, application of a statistical comparison is not appropriate.
 - Please define the error bars in the legends of figures 2e; 3b-f, i, k-l; 4k-l; EV 5b, i; EV 6e, g-h; 7c-i; 8b-d, h-l; EV 1a-b; EV 3b-c; EV 4a-e.
 - Please provide the exact p values in the legends of figures 6b-c; EV 6g-h; 7g-i; 8j-m, o; EV 4e.
 - Please note that for the figures 4b, d, p-values and statistical tests are indicated in the legends. However, the indicated label "****" has not been included in the figures. Please check and correct as appropriate.
 - Please note that scale bar and its definition are missing for figure EV 2a.
11. Papers published in The EMBO Journal are accompanied online by a 'Synopsis' to enhance discoverability of the manuscript. It consists of A) a short (1-2 sentences) summary of the findings and their significance, B) 3-4 bullet points highlighting key results and C) a synopsis image that is 550x300-600 pixels large (width x height, jpeg or png format). You can either show a model or key data in the synopsis image. Please note that the image size is rather small and that text needs to be readable at the final size. Please send us this information together with the revised manuscript.

Thank you again for giving us the chance to consider your manuscript for The EMBO Journal. I look forward to receiving the final version. Please note that I will be away from the office until January 8th, so there is no need to rush to get all adjustments done over the holiday period.

With best wishes,

leva

We realize that it is difficult to revise to a specific deadline. In the interest of protecting the conceptual advance provided by the work, we recommend a revision within 3 months (20th Mar 2025). Please discuss the revision progress ahead of this time with the editor if you require more time to complete the revisions.

Referee #2:

The authors were responsive to questions raised during the first round of review, but some points regarding the effects of somatostatin on Sonic Hedgehog (SHH)/Smoothed (SMO) signaling still need clarification. A few of the observations appear to be inconsistent with the published literature, so need to be addressed prior to publication. The remaining questions can be addressed by text revisions, so do not necessitate additional experimentation.

1. Figure 8D: SAG, a direct agonist of SMO, was used to activate SMO signaling and ciliary PTCH levels were examined as a readout for pathway induction. The manuscript states: "SAG-dependent activation of GLI2 involves exit of the Hedgehog receptor Patched from the cilium." This is not strictly correct because PTCH is regulated upstream of SMO. Its ciliary exit and degradation are regulated by SHH binding. Thus, one would not expect PTCH ciliary localization to be impacted by a direct SMO agonist. This needs clarification.

2. The observations for the effect of Ca²⁺ on GLI2 nuclear localization are not consistent with what has been reported for Ca²⁺ signaling downstream of SMO. The referenced Belgacem & Borodinsky publication shows that Ca²⁺ signaling impacts noncanonical, GLI independent SMO signaling, but the authors seem indicate that GLI is impacted by Ca²⁺ in their system:

"There was a slightly increased fraction of cells with nuclear GLI2 under resting conditions, but this fraction did not increase further in the presence of somatostatin, indicating that the response is Ca²⁺-dependent (Fig. 8I). Quantitative RT-PCR showed that SAG reduced the expression of SSTR3 and GLI3 and increased the expression of GLI2, while somatostatin had no effect on GLI2 expression but increased GLI3 expression in a SSTR3-dependent manner (Fig. 8J-L). These results show that ciliary Hedgehog and somatostatin signaling to some extent involve distinct transcriptional changes."

The authors tried to address this point during the first round of review, but the way the results are presented are still confusing and read as though the effects on Ca²⁺ are driving the transcriptional changes.

3. Why do you think SAG is affecting GLI2 and GLI3 transcriptional regulation? Gli1 is a transcriptional target, but GLI2/3 are not. Their activity is regulated by SHH at the protein level (i.e. the PKA-regulated degradation of GLI2 and 3 into truncated repressors is blocked by SHH signaling).

Minor point: Figure 8C. The text states: "Exposure to both SAG and somatostatin was associated with a slight shortening of cilia (Fig. 8C)". This reads as though cells were treated with ligands together, but the figure shows they were treated individually. Please adjust the wording to ...exposure to SAG or somatostatin...

Referee #2:

The authors were responsive to questions raised during the first round of review, but some points regarding the effects of somatostatin on Sonic Hedgehog (SHH)/Smoothed (SMO) signaling still need clarification. A few of the observations appear to be inconsistent with the published literature, so need to be addressed prior to publication. The remaining questions can be addressed by text revisions, so do not necessitate additional experimentation.

1. Figure 8D: SAG, a direct agonist of SMO, was used to activate SMO signaling and ciliary PTCH levels were examined as a readout for pathway induction. The manuscript states: "SAG-dependent activation of GLI2 involves exit of the Hedgehog receptor Patched from the cilium." This is not strictly correct because PTCH is regulated upstream of SMO. Its ciliary exit and degradation are regulated by SHH binding. Thus, one would not expect PTCH ciliary localization to be impacted by a direct SMO agonist. This needs clarification.

We have previously observed the same in mouse islets, i.e. that SAG triggers PTCH exit from the cilium (doi: 10.1083/jcb.202108101). We agree with the reviewer that SAG binding to SMO occurs downstream of PTCH and would not be expected to influence PTCH localization. However, we are unaware of studies specifically investigating this under native conditions. It has however been shown, at least under conditions of PTCH overexpression on a PTCH null background, that SAG can reduce the affinity of PTCH binding to SMO (<https://doi.org/10.1073/pnas.2006800118>). We speculate that this may lead to release of PTCH which may exit the cilium, although the mechanism may be different from that triggered by direct Hh binding. Another possibility is that SAG treatment has an impact on Hh secretion from the cells we study, as has been shown for neurons (<https://doi.org/10.1016/j.brainresbull.2023.01.014>). Pancreatic beta cells are endogenous hedgehog producers, and SMO is to some extent present in cilia even under resting conditions, although the receptor gets enriched following SAG treatment (doi: 10.1083/jcb.202108101). It would be interesting in the future to measure SHh (which is the most abundantly expressed Hh molecule in these cells) secretion under these conditions.

2. The observations for the effect of Ca²⁺ on GLI2 nuclear localization are not consistent with what has been reported for Ca²⁺ signaling downstream of SMO. The referenced Belgacem & Borodinsky publication shows that Ca²⁺ signaling impacts noncanonical, GLI independent SMO signaling, but the authors seem indicate that GLI is impacted by Ca²⁺ in their system:

"There was a slightly increased fraction of cells with nuclear GLI2 under resting conditions, but this fraction did not increase further in the presence of somatostatin, indicating that the response is Ca²⁺-dependent (Fig. 8I). Quantitative RT-PCR showed that SAG reduced the expression of SSTR3 and GLI3 and increased the expression of GLI2, while somatostatin had no effect on GLI2 expression but increased GLI3 expression in a SSTR3-dependent manner (Fig. 8J-L). These results show that ciliary Hedgehog and somatostatin signaling to some extent involve distinct transcriptional changes."

The authors tried to address this point during the first round of review, but the way the results are presented are still confusing and read as though the effects on Ca²⁺ are driving the transcriptional changes.

We thank the reviewer for pointing out the difference between canonical and non-canonical Hh signaling. We have now tried to better explain how we think canonical Hh pathway activation

depends on ciliary Ca²⁺. Shh has been shown to trigger Ca²⁺ influx in neuronal cells and to control cilia-dependent transcription (<https://www.pnas.org/doi/epub/10.1073/pnas.2220037120>). It is proposed that this signaling causes a switch in Hh signaling from proliferation to differentiation. Our observation here would be consistent with a switch from canonical to non-canonical signaling, but we have not specifically addressed this. Although it may be non-canonical it still involves PKA, consistent with our observation that PKA inhibition was sufficient to induce cilia Ca²⁺ signaling. It has also been shown that canonical, Hh-dependent inhibition of PKA and induction of GLI1 is Ca²⁺ sensitive and inhibited by the Ca²⁺ channel blocker Gd³⁺ (<https://www.pnas.org/doi/epdf/10.1073/pnas.1602393113>). In this study, Hh-induced GLI2 accumulation at the cilium tip was completely abolished by preventing Ca²⁺ influx, likely via inhibition of cilia-localized AC5/6. However, it was not shown that this specifically involved cilia Ca²⁺. Based on this, we have now revised the text in the manuscript to:

RESULTS: “Hedgehog pathway activation has been shown to stimulate cilia Ca²⁺ influx through a cAMP-dependent mechanism, and blocking this influx attenuated GLI2 processing and downstream transcription (<https://www.pnas.org/doi/epdf/10.1073/pnas.1602393113>).”

DISCUSSION: “Somatostatin-induced GLI2 nuclear entry depended on Ca²⁺, since expression of a cilia-localized Ca²⁺ chelator completely prevented somatostatin-induced GLI2 nuclear entry. Hedgehog stimulation has also been shown to result in ciliary Ca²⁺ increases, but whether this increase is required for canonical Hedgehog signaling appears to be cell type or context dependent (Belgacem & Borodinsky, 2011; Teperino et al., 2012; Moore et al., 2016). The mechanism likely involves PKA, since acute inhibition of this enzyme also triggered ciliary Ca²⁺ signaling, and Ca²⁺ in turn may inhibit AC5/6, as has been shown in other cell types (Moore et al., 2016), in a positive feedback loop. Consistently, both Ca²⁺ increases and cAMP lowering have been shown to promote cilia shortening, but whether the two messengers operate through a common pathway is not known (Besschetnova et al., 2010).”

3. Why do you think SAG is affecting GLI2 and GLI3 transcriptional regulation? Gli1 is a transcriptional target, but GLI2/3 are not. Their activity is regulated by SHH at the protein level (i.e. the PKA-regulated degradation of GLI2 and 3 into truncated repressors is blocked by SHH signaling).

We do not understand this and think that this must be further investigated using a more unbiased approach, such as RNAseq or ATTACseq. We think that the results could be confounded by the endogenous secretion of Hedgehog molecules from these cells, and that application of a SMO antagonist together with somatostatin inhibition would be one condition that may better allow us to distinguish between Hh effects and somatostatin effects. We would also like to point out that it is only GLI2, and not GLI3, that is changed with SAG treatment.

Minor point: Figure 8C. The text states: "Exposure to both SAG and somatostatin was associated with a slight shortening of cilia (Fig. 8C)". This reads as though cells were treated with ligands together, but the figure shows they were treated individually. Please adjust the wording to ...exposure to SAG or somatostatin...

Thanks for pointing this out. The sentence has now been corrected to: “Exposure to SAG or somatostatin was associated with a slight shortening of cilia (Fig. 8C).”

Dear Dr. Idevall-Hagren,

Thank you for submitting the final revised version and addressing the remaining editorial points. I sincerely apologise for the delay in communicating the decision - I was away from the office due to an illness in the family. I am now pleased to inform you that your manuscript has been accepted for publication. Congratulations on a nice study!

Before we forward your manuscript to our publishers, there are a couple of minor points that need your attention:

1. I have adjusted the legend for the figure 8j to remove the statistical information, as this is not applicable to n=2 comparisons.
2. I would like to propose some edits in the manuscript title, abstract and synopsis (please see below and the attached manuscript text file). I have also written a short blurb that will accompany the title of your manuscript in our online table of contents. Please take a look and let me know if any corrections are needed.

Title:

Somatostatin triggers local cAMP and Ca²⁺ signaling in primary cilia to modulate pancreatic β -cell function

Blurb:

Pancreatic δ -cell-released somatostatin is perceived by its receptor SSTR3 expressed on cilia of nearby β -cells, a process that is disrupted in type-2 diabetes.

Synopsis

Pancreatic δ -cells secrete somatostatin, a paracrine factor that inhibits insulin and glucagon secretion from the islet β -cells. This study shows that somatostatin is locally detected by primary cilia on nearby β -cells, where it initiates ciliary cAMP and Ca²⁺ signaling, with long-term effects on β -cell function.

- β -cell primary cilia are positioned close to δ -cells and directly detect released somatostatin.
- Somatostatin binding by ciliary SSTR lowers cAMP and triggers Ca²⁺ signaling in the cilia, thus inducing nuclear translocation of the transcription factor GLI2.
- Long-term somatostatin treatment alters β -cell Ca²⁺ and cAMP signaling through mechanisms that depend on ciliary SSTR3.
- Type-2 diabetes is associated with shortening of β -cell primary cilia and leads to reduced interactions between δ -cells and β -cell primary cilia.

If you have any questions, please do not hesitate to contact the Editorial Office. Thank you for your contribution to The EMBO Journal!

With best wishes,

Ieva
